JCB Journal of Cell Biology

# Actomyosin contractility is a potent suppressor of mesoderm induction by human pluripotent stem cells

Loic Fort[1], Vaishna Vamadevan[1], Wenjun Wang[2], and Ian G. Macara[1]

**Activation of WNT signaling in human pluripotent stem cells efficiently drives lateral mesoderm specification and subsequent cardiomyocyte differentiation. Stabilization of the WNT effector β-catenin induces mesodermal genes such as TBXT (Brachyury) and triggers an epithelial–mesenchymal transition (EMT). Although mechanical forces are essential for embryonic development, the role of actomyosin contractility during human mesoderm specification remains unclear. We show that increasing contractility through constitutively active Rho kinase or myosin light-chain kinase unexpectedly blocks β-catenin–dependent mesoderm induction and prevents EMT. In contrast, pharmacological or genetic suppression of contractility enhances Brachyury expression and advances EMT onset by 24 h. While β-catenin signaling alone promotes colony-level contractility, we find that contractility must be reduced prior to WNT activation to promote mesoderm specification, indicating a sensitization effect at the pluripotent state. Mechanistically, reduced tension decreases junctional β-catenin and increases nuclear active β-catenin, identifying actomyosin contractility as a key regulator of lineage commitment following WNT pathway activation.**

## Introduction

Mechanical forces play major roles at many stages of animal development. Starting as early as compaction at the eight-cell stage (Firmin et al., 2024), mechanics plays a central role in self-organization to shape the embryo (Lecuit and Lenne, 2007; Goodwin and Nelson, 2021; Caldarelli et al., 2024; Mundhe et al., 2025). In vitro, the differentiation of pluripotent stem cells (PSCs) into germ layers is governed by dynamic interplay between mechanical forces, signaling cues, and transcriptional regulators. Internal forces are largely generated by actomyosin contractility, and in epithelia are transmitted across cell boundaries by intercellular junctions, enabling collective organization (Behrndt et al., 2012; Shyer et al., 2017; Hall et al., 2019; Khalilgharibi et al., 2019; De Belly et al., 2022; Yang et al., 2022). But how actomyosin contractility is coupled to cell fate specification through target gene expression has remained unclear.

Human embryonic stem cell (hESC) and induced pluripotent stem cell (hiPSC) colonies can be induced to select the mesoderm lineage and eventually to differentiate into functional cardiomyocytes (Lian et al., 2012; Lian et al., 2013; Zhao et al., 2019). Both types of PSCs are strongly epithelial, and lineage specification relies on an epithelial-to-mesenchymal transition (EMT) that is reminiscent of some aspects of gastrulation (Przybyla et al., 2016; Muncie et al., 2020; Fort et al., 2022).

Mesoderm can also be generated from hPSCs by direct WNT pathway activation using the small molecule GSK3β inhibitor Chiron 99021 (hereafter CHIR), which protects cytosolic β-catenin from degradation through a similar mechanism to WNT ligand signaling. The stabilized β-catenin then enters the nucleus and binds to TCF transcription factors at target gene promoters. A separate pool of β-catenin in epithelial cells is tightly associated with E-cadherin at the adherens junctions (AJs), but whether this pool contributes to WNT signaling and mesoderm induction by hPSCs, or participates in WNT responses in other situations, remains controversial (van der Wal and van Amerongen, 2020). For example, increased junctional β-catenin has been reported to correlate with increased nuclear β-catenin, though the mechanism for this phenomenon was not explained (Przybyla et al., 2016). To the contrary, in *Drosophila* imaginal disks, actomyosin contractility can promote E-cadherin accumulation at AJs, which reduces β-catenin–mediated gene expression (Hall et al., 2019), while deletion of E-cadherin promotes WNT-responsive gene expression (Orsulic et al., 1999).

[1]Department of Cell and Developmental Biology, Vanderbilt University School of Medicine, Nashville, TN, USA; [2]Department of Biomedical Engineering, Vanderbilt University, Nashville, TN, USA.

Correspondence to Loic Fort: loic.fort@uta.edu

L. Fort's current affiliation is The University of Texas at Arlington, Dept of Biology, Arlington TX, USA.

These studies in *Drosophila* suggest that cytoplasmic β-catenin and β-catenin at the AJs are dynamically linked, but whether a similar titration system applies in human cells, and particularly in hPSCs, remains unknown. Although there are multiple studies on the impact of extracellular matrix stiffness (Przybyla et al., 2016; Muncie et al., 2020), tissue confinement (Pukhlyakova et al., 2018), and EMT (Gayrard et al., 2018) on the E-cadherin/β-catenin interaction and its downstream consequences, there has been to our knowledge no direct interrogation of how actomyosin contractility impacts mesoderm commitment by hPSCs.

From prior work on BMP4-driven mesoderm in hESCs (Przybyla et al., 2016; Muncie et al., 2020), we expected that inhibiting actomyosin activity would block induction. Surprisingly, however, we obtained the opposite result: increasing contractility was sufficient to completely block differentiation and EMT, whereas suppressing myosin light chain (MLC) phosphorylation using small molecule inhibitors or genetic tools substantially promoted mesoderm induction. These data were also puzzling because induction of differentiation alone promotes contractility of stem cell colonies. However, we discovered that the effects of elevated or suppressed actomyosin contractility have a temporal dependence, since their effects on differentiation depend on sensitization of the pluripotent state, not on events after induction. Mechanistically, we found that promoting cell relaxation led, following the WNT pathway, to loss of β-catenin from cell junctions and increased active β-catenin in the nucleus, suggesting that the β-catenin pools are dynamically coupled through actomyosin contractility to control mesoderm specification.

## Results

### Contractility is a potent inhibitor of mesoderm identity

We initially hypothesized that actomyosin contractility might support mesoderm specification, given prior reports implicating mechanical forces in lineage commitment (Przybyla et al., 2016; Muncie et al., 2020; Caldarelli et al., 2024). To promote contractility, we initially treated wild-type (WT) hiPSCs before and during differentiation with RhoA activator II (CN03), a highly specific bacterial toxin derivative that converts RhoA glutamine 63 to glutamate, locking RhoA in a constitutively active form (Flatau et al., 1997; Schmidt et al., 1997), which will then specifically activate ROCK to phosphorylate MLC (Fig. S1 A). Staining for phospho-T18/S19 Myosin Light Chain 2 (ppMLC2) confirmed that CN03 treatment led to higher MLC phosphorylation by 24 h after CHIR addition (Fig. S1 B). Strikingly, CN03-treated cells failed to express any detectable Brachyury (encoded by the *TBXT* gene, also known as the *T* gene), a mesoderm marker (Fig. S1, C and D). However, CN03 treatment caused cell toxicity past 48 h, preventing further analysis.

For this reason, we turned to genetic approaches by the lentiviral expression of activated ROCK2 or myosin light-chain kinase (MLCK) (ROCK2$^{CA}$ and MLCK$^{CA}$) under a doxycycline (Dox)-inducible promoter, with constitutive expression of a Venus reporter as a transduction marker. To provide an internal control, we added WT cells at a 1:1 ratio. We first confirmed that

Venus$^{Pos}$ cells treated with 1 μg/ml Dox showed signs of higher contractility. In cocultures containing ROCK2$^{CA}$/WT or MLCK$^{CA}$/WT cells treated with Dox, Venus$^{Pos}$ clusters showed strong cortical F-actin and higher ppMLC2 staining organized as fibers, compared with WT Venus$^{Neg}$ cells, which show more diffuse staining (Fig. S1, E and G). These data support the idea that Dox-treated Venus$^{Pos}$ cells experience higher contractility, allowing us to test the impact on cell fate. To this end, cocultures were pretreated with Dox (or vehicle) to turn on ROCK2$^{CA}$ or MLCK$^{CA}$ before starting the differentiation protocol. We prolonged the differentiation to 72 h in this coculture system to better assess the impact of enhanced contractility (Fig. 1 A). Venus-positive and Venus-negative areas were outlined based on Venus expression across clusters (Fig. S1, F and H), and percentages of cells expressing mesoderm and EMT markers were measured. Strikingly, following 72 h of differentiation in the presence of Dox, Venus$^{Pos}$ clusters of ROCK2$^{CA}$ cells showed a dramatic decrease in mesoderm (Fig. 1, B and C) and primitive streak gene expression (Fig. 1, D and E) compared with Venus$^{Neg}$ cells or vehicle-treated cells. Venus$^{Pos}$ cells also failed to undergo EMT, which usually occurs 50–52 h after differentiation, as observed by the absence of Slug expression and the persistence of the epithelial marker ZO-1 (Fig. 1, F–H).

In addition to ROCK2, MLC2 phosphorylation was previously shown to be spatially regulated by MLCK in polarized and nonpolarized cells (Totsukawa et al., 2000; Shi et al., 2024, *Preprint*). Therefore, we also probed the effect of MLCK-driven contractility on cell identity by expressing an active mutant. Consistent with the effects of ROCK2$^{CA}$, MLCK$^{CA}$-positive clusters failed to commit to the primitive streak and mesoderm lineage and to trigger the EMT program required for mesoderm identity (Fig. S1, I–K). These data support the surprising conclusion that activation of ROCK/MLCK-mediated contractility is sufficient to counteract the effect of WNT signaling activated by CHIR and completely prevents mesoderm specification and EMT.

### Genetic suppression of actomyosin contractility promotes stem cell conversion to the mesoderm lineage

This unexpected finding prompted us to investigate whether inhibition of contractility might conversely promote mesoderm differentiation. We designed a genetic approach to suppress contractility in human cells. Dephosphorylation of ppMLC is mediated by MLC phosphatase, a heterotrimer composed of a catalytic subunit PP1cβ, a myosin phosphatase–targeting regulatory subunit (MYPT1), and a protein of unknown function M20 (Matsumura et al., 2011; Kiss et al., 2019) (Fig. 2, A and B). Based on previous work describing an optogenetic construct to control actomyosin contractility (Yamamoto et al., 2021), we created a truncated MYPT1, fused with a nuclear export sequence and a mNeonGreen (mNG) reporter (Fig. 2 B), which is expected to constitutively activate PP1 cβ and promote MLC2 dephosphorylation.

To validate the activity of this new tool, we cloned the construct (referred to as MYPT1$^{CA}$-NES-mNG) under a Dox-inducible promoter and established a stable hiPSC line. Global reduction of ppMLC2 was observed by immunoblot and

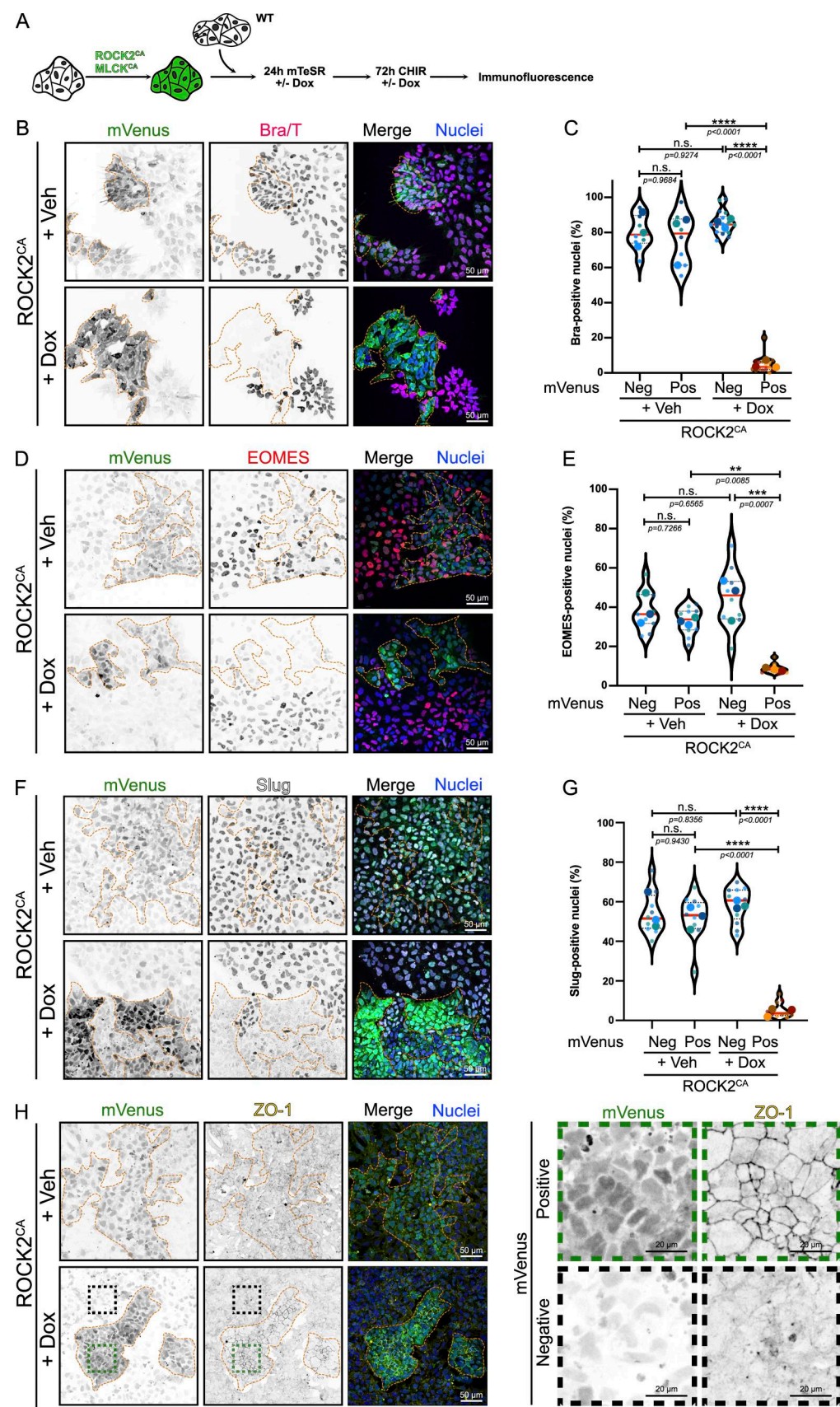

Figure 1. **Increased actomyosin contractility is sufficient to block mesoderm commitment and EMT. (A)** Experimental design for coculture experiment. hiPSCs were transduced with lentivector expressing ROCK2CA or MLCKCA. mVenus positive–transduced cells were mixed with mVenus-negative WT hiPSCs.

ROCK2$^{CA}$/MLCK$^{CA}$ expression is induced by the addition of Dox 24 h prior to initiating mesoderm differentiation. Dox induction was maintained during the 72 h of differentiation using CHIR before fixing and staining. Note that mVenus expression is constitutive and used as a marker for transduction, while the expression of ROCK2/MLCK is Dox-inducible. **(B–H)** Representative MaxIP immunofluorescence for Brachyury (B), EOMES (D), Slug (F), and ZO-1 (H) in Veh and Dox-induced ROCK2$^{CA}$ coculture, treated for 72 h with CHIR. mVenus-positive cell clusters are highlighted by an orange dotted line. Individual channels are presented as inverted LUT. Scale bar = 50 μm. For H, magnified mVenus-positive and mVenus-negative areas are shown as insets. Scale bar = 20 μm. Quantification of Brachyury (C), EOMES (E), and Slug-positive cells (G) is reported as a violin plot, comparing mVenus-positive *vs* mVenus-negative clusters in the presence or absence of Dox. Median (plain red line) and quartiles (dotted black lines) are displayed. For Brachyury, n = 7 (+ Veh) and n = 15 (+ Dox) across N = 3 independent biological repeats. For EOMES, n = 9 across N = 3 independent biological repeats. For Slug, n = 9 (+ Veh) and n = 10 (+ Dox) across N = 3 independent biological repeats. One-way ANOVA with Tukey's multiple comparisons posttest was performed. Veh, vehicle. **** P < 0.0001, *** P < 0.001, ** P < 0.01.

immunofluorescence after Dox addition at 1 μg/ml (Fig. 2, C–E). In addition, F-actin organization was strongly affected, specifically the loss of cortical actin (Fig. 2 F). Strikingly, MYPT1$^{CA}$ cells pretreated overnight with Dox and followed by differentiation displayed enhanced Brachyury expression (Fig. 2, G–L) and earlier induction of the EMT genes Snail and Slug (Fig. 2 H) following Dox induction, compared with controls. Because apoptosis is crucial to enable mesoderm fate specification (Fort et al., 2022), we probed for PARP cleavage as an apoptosis marker but did not detect any difference (Fig. 2 J). We conclude that reducing MLC2-driven contractility significantly accelerates mesoderm specification.

**Promotion of hiPSC and hESC conversion to the mesoderm lineage by actomyosin relaxation is time-dependent**

We next used small molecule inhibitors as an orthogonal method to validate our previous findings. This approach allowed us to expand our investigation to hESCs, while also providing better control over the timing of inhibition.

First, hiPSCs were pretreated overnight with the ROCK inhibitor, Y-27632, to suppress actomyosin contractility at the pluripotent state, then differentiated by the addition of CHIR in the continued presence of Y-27632 (Fig. 3 A). We confirmed that MLC2 phosphorylation was reduced by ROCK inhibition (Fig. S2, A and B). Similar to the genetic approach, pharmaceutical blockage of contractility strongly promoted Brachyury expression (Fig. 3, B and C). The expression of other key mesoderm (*MESP1* and *TBX6*) and EMT (*SNAI1* and *SNAI2*) genes was also upregulated following Y-27632 treatment, demonstrating that contractility directly affects gene expression rather than protein stability (Fig. 3, D–G). To rule out an effect of ROCK inhibition on global transcription, we examined expression levels of genes unrelated to mesodermal fate and did not detect any response to decreased contractility, suggesting a specific role of actomyosin contractility in mesoderm specification (Fig. S2 C).

At the cellular level, Brachyury expression is initially restricted to the colony edges at 24 h, before broader expression across the colony at later time points (Fig. 3, H–I). However, Y-27632 treatment resulted in more widespread Brachyury expression (Fig. 3, H–I), markedly accelerated EMT as shown by the loss of the epithelial marker ZO-1 from tight junctions (Fig. 3, H and J), and increased expression of the mesenchymal marker Slug (Fig. S2 D). To validate these data, we used a more specific ROCK inhibitor, H-1152, which caused similar decreases in ppMLC2 (Fig. S2, E and F) and recapitulated the early EMT and higher induction of mesoderm markers (Fig. S2, G–M).

Apoptosis was not affected following Y-27632 (Fig. S2 N) or H-1152 treatment (Fig. S2 I). Importantly, these effects of ROCK inhibition were conserved in hESCs expressing a knock-in T-mNG fusion (Muncie et al., 2020) (Fig. 3, K and L) and in WT H9 hESCs treated with CHIR (Fig. S2, N–O), demonstrating that the response is not unique to our iPSCs. Similar to hiPSCs, there was no effect of reduced contractility on apoptosis initiation by the hESCs (Fig. S2, N and P) or on exit from the pluripotency state (Fig. S2, N and Q). Interestingly, using BMP4 plus bFGF as an alternative differentiation protocol (Muncie et al., 2020), we could detect Brachyury expression by 48 h, and Y-27632-induced relaxation promoted expression (Fig. S2 R). This result suggests that the effects of contractility on mesoderm differentiation are not unique to the use of small molecule–based β-catenin stabilization but are broadly applicable to morphogen-dependent protocols.

Because MLCK activation was sufficient to block cell commitment (Fig. S1, I–K), we tested the effect of ML-7, a potent MLCK inhibitor (Makishima et al., 1991). ML-7 led to efficient reduction in ppMLC2 (Fig. 3, M and N) and strongly enhanced Brachyury expression at 48 h, without affecting cell death (Fig. 3, O and P). Together, these data show that pharmacological blockade of MLC2-mediated force generation enhances mesoderm commitment quantitatively and accelerates the EMT required for conversion to cardiac mesoderm.

These results were particularly surprising, because hiPSC colonies show rapid retraction of their edges following CHIR addition (Fig. S3 A and Video 1), consistent with observations from other groups (Hookway et al., 2024, *Preprint*). Having previously reported an essential and permissive role of apoptosis during mesoderm specification (Fort et al., 2022), we tested whether colony retraction was related to the wave of cell death that follows CHIR addition. However, cotreatment with a pan-caspase inhibitor (Q-VD-OPH) did not prevent retraction of colony edges (Fig. S3 A), ruling out effects of cell density and cell death.

Commitment to the mesoderm lineage also resulted in thicker epithelial layers, containing taller cells (Fig. S3, B–D), suggesting morphological changes that are highly dependent on the cellular contractile state. Consistent with the role of nuclei as a mechanosensor (Lomakin et al., 2020; de Leeuw et al., 2024), differentiating cells exhibit significantly smaller nuclear area after 48 h of CHIR treatment (Fig. S3, E and F). To address the variation of cell density, we correlated each of our nuclear area measurements with the actual cell number (Fig. S3 G). As expected, the nuclear area negatively correlates with cell density

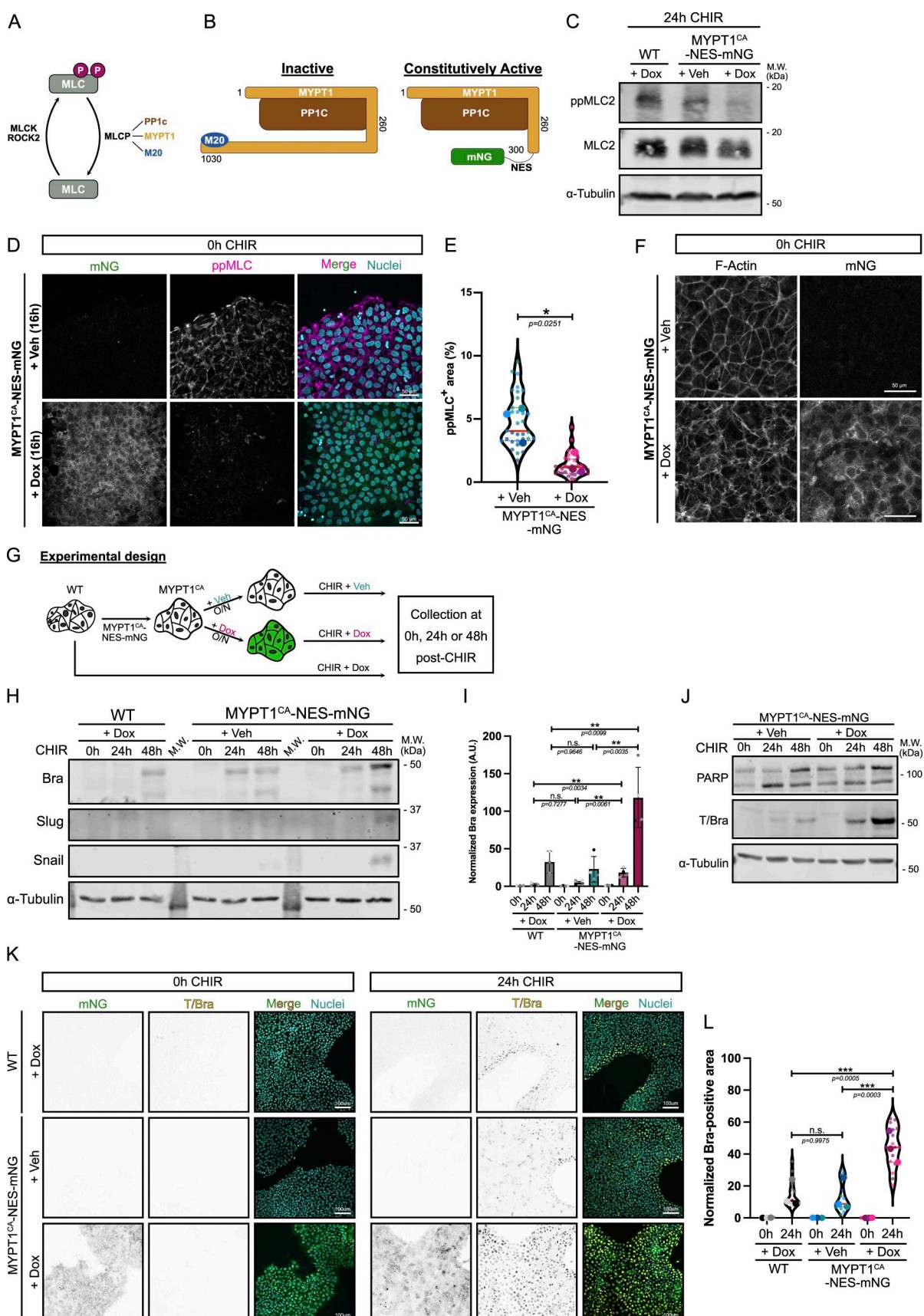

Figure 2. **Genetic inhibition of actomyosin contractility promotes hiPSC conversion to the mesoderm lineage. (A)** Summary of the main regulators of MLC phosphorylation. **(B)** Schematic representation of full-length human MYPT1 (1030 aa), interacting with PP1c phosphatase and a protein of unknown

function M20. Truncated MYPT1 Nterm 1–300 (containing the MLC-binding domain) leads to constitutive recruitment to and activation of PP1C. Truncated MYPT1 was fused with NES-mNG as a marker, and is referred to as MYPT1$^{CA}$-NES-mNG. **(C)** Immunoblot of ppMLC2 and total MLC2 in WT hiPSCs treated with Dox or hiPSCs expressing MYPT1$^{CA}$-NES-mNG in the presence or absence of 1 µg/ml Dox. α-Tubulin was used as a loading control. M.W. are displayed on the right side. **(D and E)** Representative MaxIP immunofluorescence of hiPSCs expressing MYPT1$^{CA}$-NES-mNG and treated or not with Dox overnight. Cells were fixed and stained for ppMLC2 (magenta), DNA (blue), and tight junction marker ZO-1 (orange). Scale bar = 50 µm (D). Fraction of the cellular area positive for ppMLC2 following Dox induction of MYPT1$^{CA}$-NES-mNG hiPSC and reported as violin plots. Median (plain red line) and quartiles (dotted black lines) are displayed. $n$ = 32 technical repeats across $N$ = 3 independent biological repeats. A two-tailed unpaired $t$ test was performed on the biological repeats (E). **(F)** Representative MaxIP immunofluorescence of hiPSCs expressing MYPT1$^{CA}$-NES-mNG and treated or not with Dox overnight. Cells were stained for F-actin using phalloidin. **(G)** Experimental design. hiPSCs were transduced with pInducer20-MYPT1$^{CA}$-NES-mNG, and stable population was selected with puromycin. MYPT1$^{CA}$ cells were treated or not with Dox for 16 h and treated with CHIR supplemented or not with Dox. Parental WT line was differentiated with Dox. **(H and I)** Representative immunoblot for Brachyury (mesoderm marker), and Snail and Slug (EMT markers) using WT and MYPT1$^{CA}$-NES-mNG hiPSCs, following CHIR treatment (0–48 h) in the presence or absence of Dox, as shown in G. M.W. are displayed on the right side (H). Brachyury expression was quantified by densitometry and normalized to α-tubulin as a loading control across $N$ = 3–4 independent biological repeats. Mean and SD are displayed. A one-way ANOVA with Tukey's multiple comparisons posttest was performed (I). **(J)** Immunoblot of PARP (cell death marker) and Brachyury (mesoderm marker) during mesoderm commitment (0-h to 48-h CHIR) using MYPT1$^{CA}$-NES-mNG hiPSCs induced +/Dox, as shown in G. M.W. are displayed on the right side. **(K and L)** Representative MaxIP immunofluorescences of WT and MYPT1$^{CA}$-NES-mNG hiPSCs treated with CHIR for 24 h +/− Dox as shown in G. Cells were stained for nuclei (DNA) and mesoderm marker (T/Bra—inverted LUT). Scale bar = 100 µm (K). Quantification of Brachyury expression is presented as violin plots. Median (plain red line) and quartiles (dotted black lines) are displayed. $n$ = 15–16 technical repeats across $N$ = 3 independent biological repeats. One-way ANOVA with Šidák's multiple comparisons posttest was performed on biological repeats (L). M.W., molecular weights. ***$P < 0.001$, **$P < 0.01$, *$P < 0.05$. Source data are available for this figure: SourceDataF2.

before differentiation. However, we did not find any correlation following CHIR treatment, together suggesting that the smaller nuclear size at 48 h of differentiation is unlikely to be related to cell density.

While these observations were indirect, they point toward an intrinsic change in contractile status during mesoderm commitment. To directly test for higher contractility, differentiating cells were stained for ppMLC2. Immunofluorescence revealed a gradual increase in ppMLC2 intensity over time (Fig. S3, H–I). Finally, we took advantage of the birefringence property of actin fibers to measure optical retardance using quantitative polarization microscopy (QPOL) (Wang et al., 2018). Birefringence changes proportionally to applied strain, providing an orthogonal way to probe for contractile status. Confirming our previous data, retardance values increased during hiPSC-to-mesoderm conversion (Fig. S3 J). Specificity of the retardance value as a readout for contractility was tested by incubating undifferentiated hiPSCs with Y-27632, which reduced retardance. Together, these data suggest that hiPSCs differentiating along the cardiac mesoderm trajectory, in the absence of any external perturbation, experience increasing contractile forces mediated by MLC2 phosphorylation.

How can we reconcile the surprising observation that differentiating cells intrinsically become more contractile, with the inhibitory function of actomyosin contractility on mesodermal specification? We wondered whether the timing of inhibitor treatment was important and therefore staggered the timing such that Y-27632 was added either 12 h prior to CHIR addition (–12 h), at the time of addition (0 h), or 12 h following CHIR addition (+12 h), and analyzed Brachyury expression after 12 h later (Fig. 3 Q). Pretreatment of PSCs with ROCK inhibitor (–12 h) strongly promoted mesoderm differentiation. Cotreatment with ROCK inhibitor and CHIR reduced the effect, and addition of ROCK inhibitor at +12 h had no effect on differentiation (Fig. 3, R and S). We conclude that reduced contractility sensitizes PSCs for mesoderm specification but has no effect once specification has been initiated.

## Hippo pathway and force-dependent WNT ligand secretion do not contribute to enhanced mesoderm specification

Mechanistically, how do changes in contractility impact cell fate specification? A major target of mechanical force in epithelia is the Hippo pathway. YAP, a downstream effector of Hippo signaling, has been previously associated with cell fate patterning in 2D gastruloids (Stronati et al., 2022), and substrate stiffness-driven mesoderm specification (Pagliari et al., 2021). Low tension activates this pathway, resulting in YAP phosphorylation and degradation. Conversely, high tension suppresses Hippo signaling, blocking phosphorylation of YAP. The non-phosphorylated YAP translocates to the nucleus to regulate gene expression. Surprisingly, however, nuclear/cytoplasmic YAP ratios were indistinguishable upon cell relaxation in the MYPT$^{CA}$-NES-mNG cells (Fig. S4, A and B). We confirmed these data by specifically looking at phosphorylated YAP ratios by immunoblot, which were not affected during hiPSC differentiation or by cell relaxation (Fig. S4, C and D). We next directly probed for *CTGF* and *CYR61* expression, two well-described YAP target genes. Gene expression strongly decreased during differentiation, as expected given the reported repressive functions of YAP during mesendoderm specification (Meyer et al., 2023), but their expression level was not affected by the cellular contractile status (Fig. S4 E). Together, these data rule out a significant role of the Hippo pathway in the acceleration of mesodermal differentiation following cell relaxation.

Next, we investigated the positive feedback loop reported for BMP4-driven mesoderm commitment, which relies on tension-dependent secretion of canonical WNT ligands (Muncie et al., 2020). We treated our MYPT1$^{CA}$-NES-mNG cells with Dox to trigger cell relaxation and probed for canonical and noncanonical WNT ligands. As expected, the expression of *TBX6* (a mesoderm marker) increased in low contractile cells. Interestingly, canonical WNT ligand expression (*WNT3A* and *WNT8A*) was strongly upregulated by suppression of contractility, while noncanonical *WNT4* expression was unaffected by CHIR treatment or by relaxation (Fig. S4 F).

Figure 3. **Pharmaceutical inhibition of actomyosin contractility promotes hiPSC and hESC conversion to the mesoderm lineage. (A)** Experimental design. WT hiPSCs or hESCs were pretreated overnight with small molecules to lower contractility, followed by differentiation. **(B)** Immunoblot of Brachyury

during hiPSC-to-mesoderm commitment (0–48 h) ±10 µM ROCK inhibitor Y-27632, as shown in A. M.W. are displayed on the right side. **(C)** Brachyury expression was quantified by densitometry and normalized to α-tubulin as a loading control across $N = 3$ independent biological repeats. Mean and SD are displayed. A two-tailed unpaired $t$ test was performed. **(D–G)** Relative expression of mesoderm markers (*MESP1, TBX6*) (D and E) and EMT markers (*SNAI1, SNAI2*) (F and G) 48 h after CHIR treatment ±10 µM ROCK inhibitor as shown in A. $N = 3$ independent biological repeats except for *MESP1*, $N = 4$ independent biological repeats. Mean and SD are displayed. A two-tailed unpaired $t$ test was performed. **(H)** Representative MaxIP immunofluorescences during hiPSC-to-mesoderm commitment (0–48 h) ±10 µM ROCK inhibitor as shown in A. Cells were stained with a nuclear marker (DNA), EMT marker (ZO-1—inverted LUT), and mesoderm marker (T/Bra—inverted LUT). Scale bar = 50 µm. **(I and J)** Quantification of Brachyury expression (I) and ZO-1 intactness (J) is represented as violin plots with individual measurements (small dots) averaged for each biological repeat (large dots). Median (plain red line) and quartiles (dotted black lines) are displayed. $n = 10$–15 technical repeats across $N = 3$ independent biological repeats. One-way ANOVA with Tukey's multiple comparisons posttest was performed on biological repeats. **(K and L)** Analysis of Brachyury expression by flow cytometry following 24-h CHIR treatment ±10 µM ROCK inhibitor, as shown in A, using Bra-mNG knock-in H9 hESC line (K). Quantification of the percentage of mNG-positive cells from $N = 4$ independent biological repeats. Mean and SD are displayed. A two-tailed unpaired $t$ test was performed (L). **(M and N)** Representative MaxIP immunofluorescence of hiPSCs treated with a MLCK inhibitor (3 µM ML-7) or Veh (DMSO), as shown in A. Cells were stained for nuclei (DNA) and ppMLC2 (inverted LUT). Scale bar = 50 µm. Magnified views of the yellow dotted ROI are shown for ppMLC2. Scale bar = 20 µm (M). Percentage of cellular area positive for ppMLC2 following Veh or ML-7 treatment is reported as violin plots. Median (plain red line) and quartiles (dotted black lines) are displayed. $n = 15$ technical repeats across $N = 3$ independent biological repeats. A two-tailed unpaired $t$ test was performed on the biological repeats (N). **(O and P)** Immunoblot of Brachyury (mesoderm) and PARP (cell death marker) using hiPSCs in the presence (+ ML-7) or absence (+ Veh) of 3 µM of MLCK inhibitor at the basal state (0 h) and during CHIR treatment (24, 48 h), as shown in A. Cells treated with CHIR and Q-VD (caspase inhibitor) for 48 h were used as a control for PARP immunoblot. M.W. are displayed on the right side (O). Brachyury expression was quantified by densitometry and normalized to α-tubulin as a loading control across $N = 3$ independent biological repeats. Mean and SD are displayed. The Mann–Whitney test was performed for the 24-h time points, and a two-tailed unpaired $t$ test was performed for the 48-h time points (P). **(Q)** Experimental design. Effects of Y-27632 were tested at different time points along the mesoderm commitment. 10 µM Y-27632 was added 12 h before differentiation (–12 h), at the time of differentiation (0 h), or 12 h after differentiation (+12 h). All conditions were collected 24 h after differentiation. **(R and S)** Representative immunoblot for Brachyury following the addition of Veh or 10 µM Y-27632 as shown in Q. M.W. are displayed on the right side (R). Brachyury expression was quantified by densitometry and normalized to α-tubulin as a loading control across $N = 3$ independent biological repeats. Mean and SD are displayed. Two-way ANOVA with Šidák's multiple comparisons posttest was performed (S). M.W., molecular weights; Veh, vehicle. ****$P < 0.0001$, ***$P < 0.001$, **$P < 0.01$, *$P < 0.05$. Source data are available for this figure: SourceDataF3.

To test whether increased *WNT3A/8A* expression was responsible for increased mesoderm commitment in low contractile cells, we blocked WNT ligand processing and secretion using IWP-2, a porcupine inhibitor. The addition of IWP-2 did not reverse mesoderm gene expression in Y-27632–treated cells, suggesting that this pathway does not drive accelerated mesoderm specification in response to cell relaxation (Fig. S4 G). Together, these data ruled out the involvement of two major pathways, previously thought to link cell tension to mesoderm commitment (Muncie et al., 2020; Pagliari et al., 2021).

### Intercellular adhesion and AJs mediate cell contractility

Seeking a potential mechanism, we turned our attention to intercellular AJs. These junctions are under tension generated by actomyosin interactions with vinculin/α-catenin, which in turn are coupled to β-catenin and E-cadherin (Pinheiro and Bellaïche, 2018). In addition, tension across AJs is mediated by homotypic and calcium-dependent interactions between E-cadherin molecules on adjacent cells (Koch et al., 1997). As a major signaling hub, we reasoned that AJs might transduce mechanical forces to impact cell fate specification. We first tested whether disrupting AJs using EGTA (a calcium chelator) would affect cell specification. EGTA treatment caused colony decompaction (Fig. 4 A and Fig. S5 A), consistent with previous literature linking AJ disengagement with reduced force transmission (le Duc et al., 2010; Smutny et al., 2010). Strikingly, EGTA treatment triggered a strong enhancement of mesodermal and EMT marker expression (Fig. 4, B–E; and Fig. S5, B and C). Knowing that reduced contractility mimics this phenotype, we tested whether EGTA treatment would alter contractility. Staining for ppMLC2 confirmed that EGTA-treated cells are in a low contractile state (Fig. S5, D and E).

As an orthogonal approach, we sought to uncouple force transmission from the actomyosin network to E-cadherin via vinculin/α-catenin. To this end, we created α-catenin knockdown (*CTNNA1* KD) hiPSCs, with a 50% decrease in protein expression (Fig. S5, F and G). First, because AJs are required for stemness acquisition during fibroblast reprogramming to iPSCs (Bedzhov et al., 2013), we checked that the *CTNNA1* KD cells had not lost pluripotency (Fig. S5, F and H–I). Notably, ppMLC2 levels were 50% lower in the KD cells (Fig. 4, F and G), confirming that α-catenin is a major contributor to cell actomyosin activity. The *CTNNA1* KD hiPSCs exhibited higher and earlier expression of Brachyury. Strikingly, however, these cells failed to further respond to Y-27632 treatment, while control cells responded as expected by increasing Brachyury expression (Fig. 4, H and I). Together, these data demonstrate that AJ disruption reduces actomyosin contractility and that α-catenin–mediated force transmission is a major determinant of mesoderm specification kinetics.

### Junctional β-catenin localization scales with intrinsic actomyosin contractility in undifferentiated hiPSCs

Our discovery of a key role of AJ mechanics in cell fate specification suggested that β-catenin, which in epithelial cells is mostly bound to E-cadherin at AJs, might respond to changes in actomyosin contractility by modulating WNT signaling responsiveness. Release of β-catenin from AJs, for example, might increase the cytoplasmic and nuclear pools to amplify the expression of *TBXT* and other mesoderm genes. Mechanosensitive functions for β-catenin during development and tissue homeostasis have been described (Röper et al., 2018; Muncie et al., 2020) but whether the junction-associated and cytoplasmic β-catenin pools both contribute to WNT signaling is still controversial (van der Wal and van Amerongen, 2020), and other

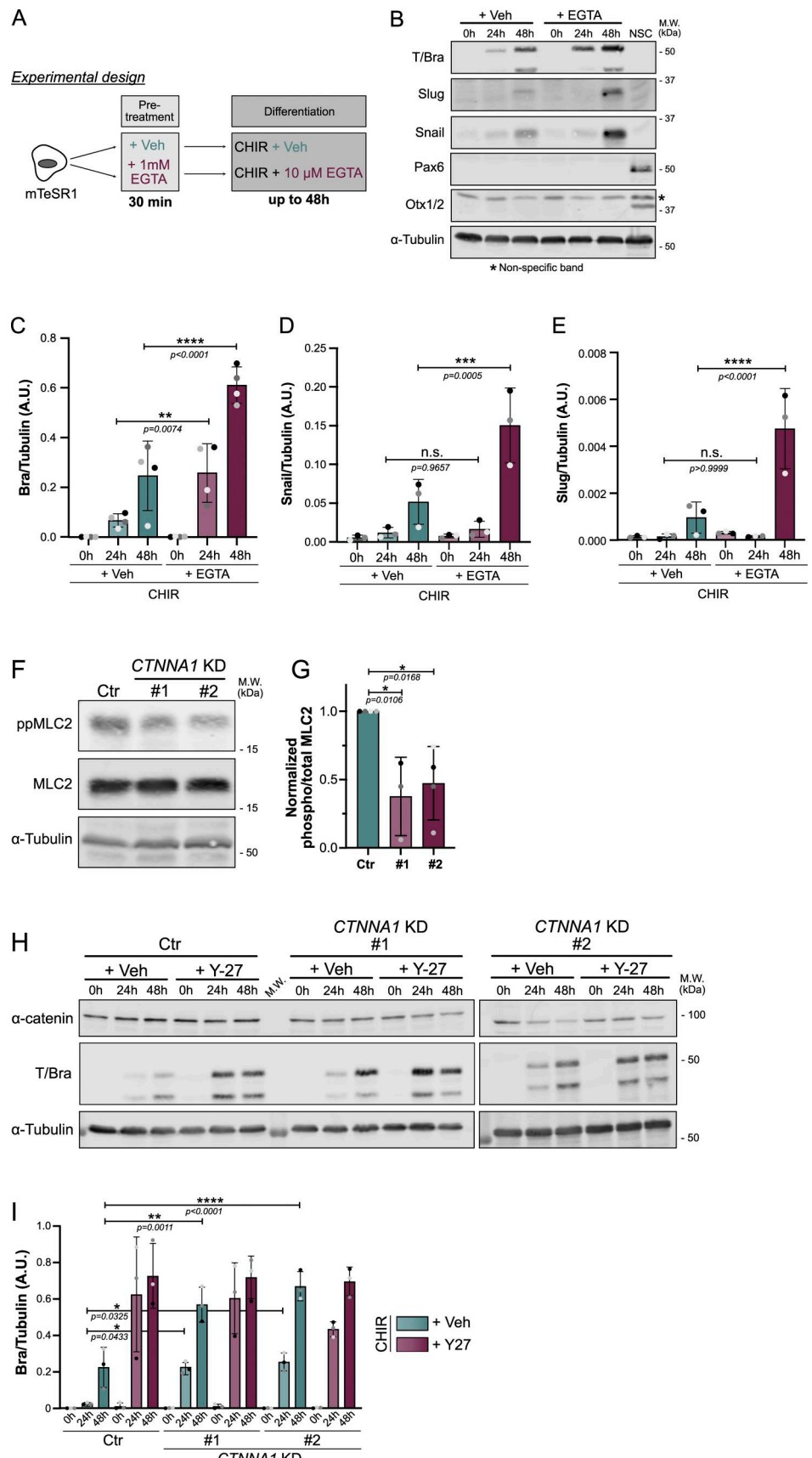

Figure 4. **AJ disengagement promotes mesoderm fate. (A)** Experimental design. hiPSCs were pretreated with 1 µM EGTA to disrupt calcium-dependent E-cadherin junctions. Following pretreatment, hiPSCs were differentiated in the presence of a lower concentration of EGTA. **(B–E)** Representative immunoblot

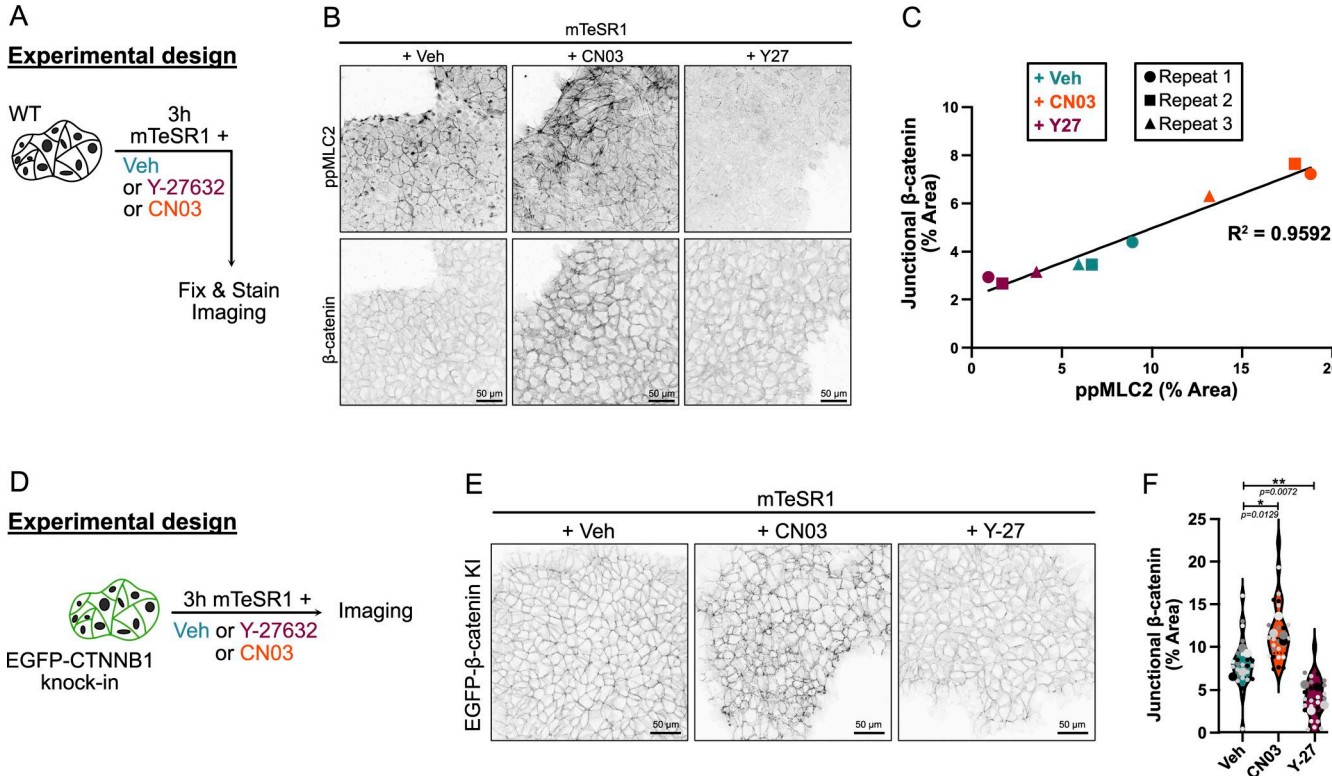

of hiPSCs treated as shown in A and probed for Brachyury (mesoderm marker), Snail and Slug (EMT markers), and Pax6 and Otx1/2 (neuroectoderm markers as negative controls). The NSC lysate was used as a positive control for Pax6 and Otx1/2. The nonspecific band is designated by an asterisk. M.W. are displayed on the right side (B). The expression of Brachyury (C), Snail (D), and Slug (E) was quantified by densitometry and normalized to α-tubulin across $N = 3$ independent biological repeats. Mean and SD are displayed. One-way ANOVA with Šidák's multiple comparisons posttest was performed. **(F and G)** Representative immunoblot of control (Ctr) and *CTNNA1* KD hiPSC lines at the basal state and probed for total and phospho-MLC2. M.W. are displayed on the right side (F). Quantification of active MLC2 (ppMLC2/MLC2) was measured by densitometry across $N = 3$ independent biological repeats. Mean and SD are displayed. One-way ANOVA with Dunnett's multiple comparisons posttest was performed (G). **(H and I)** Representative immunoblot of control (Ctr) and *CTNNA1* KD #1 and #2 hiPSC lines, differentiated for up to 48 h in the presence or absence of 10 μM of ROCK inhibitor Y-27632 and probed for α-catenin, Brachyury, and α-tubulin as a loading control. M.W. are displayed on the right side (H). Normalized Brachyury expression was measured by densitometry across $N = 3$ independent biological repeats. Mean and SD are displayed. One-way ANOVA with Šidák's multiple comparisons posttest was performed (I). M.W., molecular weights; NSC, neural stem cell. ****$P < 0.0001$, ***$P < 0.001$, **$P < 0.01$, *$P < 0.05$. Source data are available for this figure: SourceDataF4.

studies have suggested a connection in which increased junctional β-catenin correlates with increased nuclear β-catenin (Przybyla et al., 2016).

We hypothesized that loss of tension at AJs would release β-catenin to promote WNT-responsive gene expression and conversely that increased tension would promote binding at AJs, thereby titrating out the free cytoplasmic/nuclear β-catenin. To test this mechanism, undifferentiated hiPSCs were treated with either the Rho activator CN03 or ROCK inhibitor Y-27632, causing an increase or decrease in ppMLC2, respectively (Fig. 5, A and B; and Fig. S6 A). Strikingly, under each condition, junctional β-catenin levels correlated closely with the level of active

ppMLC2 in the cells (Fig. 5, B and C; and Fig. S6 B). To rule out staining artifacts, we performed comparable experiments using our EGFP-β-catenin knock-in hiPSC line (Fig. 5 D), with a similar conclusion (Fig. 5, E and F; Fig. S6 C; and Video 2). Importantly, overall β-catenin expression was not affected by the contractile state, suggesting that contractility does not regulate the β-catenin level but rather its localization (Fig. S6, D–F). Finally, fluorescence recovery after photobleaching (FRAP) experiments showed that β-catenin kinetics was not affected by Y-27632 prior to and during differentiation (Fig. S6, G–K). Our interpretation is that contractility does not measurably alter the exchange rate of the junction-engaged β-catenin pool, consistent with a previous

Figure 5. **Contractility differentially regulates β-catenin localization at AJs. (A)** Experimental design. **(B)** Representative immunofluorescence of fixed hiPSCs treated with Veh, 4 μg/ml CN03, or 10 μM Y-27632 for 3 h and stained for ppMLC2 and β-catenin (inverted LUT), as shown in A. Scale bar = 50 μm. **(C)** Linear regression analysis of junctional β-catenin and cellular ppMLC2 across $N = 3$ independent biological repeats (representing $n = 16$ for Veh, $n = 19$ for CN03, and $n = 17$ technical repeats for Y-27632). $R^2$ value is reported. **(D)** Experimental design. **(E and F)** Representative still pictures from mEGFP-β-catenin KI hiPSCs treated with Veh, 4 μg/ml CN03, or 10 μM Y-27632 for 3 h, as shown in D. Scale bar = 50 μm (E). Quantification of junctional β-catenin is reported for the different treatments across $N = 4$ independent biological repeats (representing $n = 25$ technical repeats). Violin plots representing median and quartiles. One-way ANOVA with Dunnett's multiple comparisons posttest was performed on biological repeats (F). KI, knock-in; Veh, vehicle. **$P < 0.01$, *$P < 0.05$.

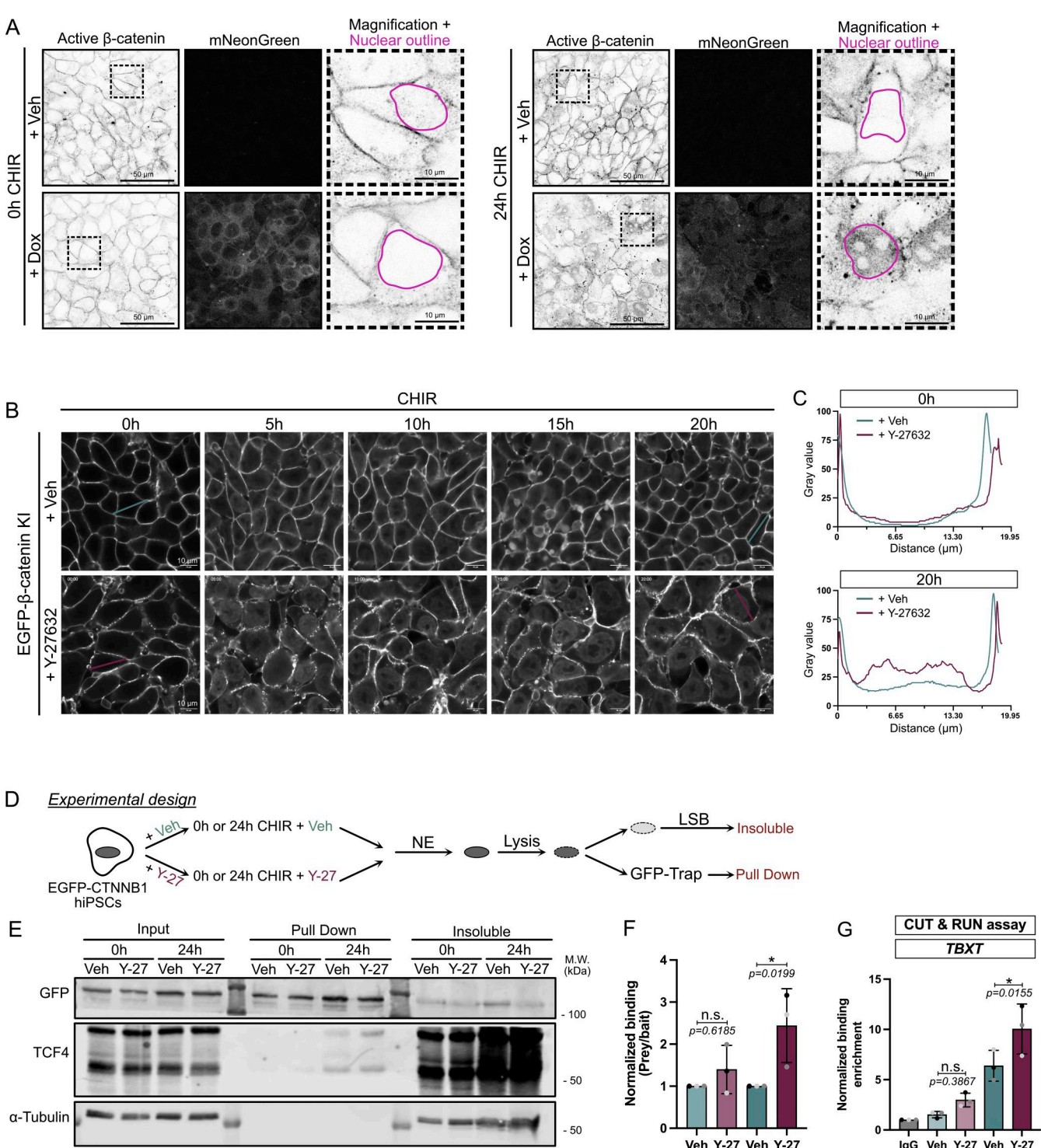

Figure 6. **Reduced cell contractility promotes β-catenin nuclear activity and enhances binding to the mesodermal gene. (A)** Representative immunofluorescence images for active non–phospho-S45 β-catenin (inverted LUT—confocal slice) and mNG (MaxIP). MYPT1^CA-NES-mNG hiPSCs were pretreated with Veh or Dox overnight and fixed at 0 and 24 h after CHIR treatment ± Dox as shown in Fig. S7 A. Scale bar = 50 µm. Insets represent a magnified view of the active non–phospho-S45 β-catenin with an example of the nuclei binary mask overlaid (pink). Scale bar = 10 µm. **(B and C)** Still confocal time-lapse imaging of mEGFP-β-catenin knock-in hiPSCs treated with CHIR ±10 µM Y-26732 (related to Video 3). Time is shown in hours. Scale bar = 10 µm (B). mEGFP-β-catenin intensity is reported as a scan line plot across a cell at 0 and 20 h (C). **(D)** Experiment design for GFP-Trap immunoprecipitation. mEGFP-β-catenin knock-in hiPSCs were pretreated with Veh or 10 µM Y-27632 and differentiated for 24 h. Nuclei were isolated, and nuclear lysates were mixed with GFP-Trap beads. NE buffer = Nuclei Extraction buffer. LSB = lysate sample buffer. **(E and F)** Representative GFP-Trap immunoprecipitation of mEGFP-β-catenin knock-in hiPSCs, probed for TCF7L2/TCF4 and α-tubulin as a loading control. M.W. are displayed on the right side (E). Quantification of TCF4 (Prey) binding to mEGFP-β-catenin (bait) was measured by densitometry across N = 3 independent biological repeats. Mean and SD are displayed. One-way ANOVA with Šidák's multiple

comparisons posttest was performed (F). **(G)** CUT&RUN assay for IgG or β-catenin binding to *TBXT* promoter in WT hiPSCs pretreated with Veh or 10 µM Y-27632 (0-h CHIR) or following 24 h of CHIR treatment in the presence of absence of ROCK inhibitor. N = 3 independent biological repeats. Mean and SD are displayed. One-way ANOVA with Šidák's multiple comparisons posttest was performed. M.W., molecular weights; Veh, vehicle. *P < 0.05. Source data are available for this figure: SourceDataF6.

### Cell relaxation promotes nuclear β-catenin localization and occupancy at the *TBXT* promoter

In the absence of WNT signaling, cytosolic β-catenin is maintained at a low level by association with a destruction complex, where it is phosphorylated at Ser45 by CK1, which primes the protein for subsequent phosphorylation by GSK3β, driving its ubiquitination and degradation. WNT activation disrupts the destruction complex, permitting the accumulation of stable, nonphosphorylated β-catenin, which can enter the nucleus to drive gene expression. This "active" state can be assessed from the detection of nonphosphorylated Ser45 β-catenin. To test for differential accumulation of active β-catenin, we performed immunostaining on the MYPT1^CA-NES-mNG cells as shown in Fig. S7 A. Lowering contractility prior to differentiation did not result in increased nuclear β-catenin (Fig. 6 A), probably because the destruction complex degrades any free β-catenin. However, we observed the nuclear enrichment of active β-catenin (Fig. 6 A; and Fig. S7, B and C) after 24 h of differentiation in low contractile conditions.

We validated this observation by imaging endogenous mEGFP-β-catenin knock-in hiPSCs treated with CHIR in either control or low contractile (+Y-27632) conditions. First, we noticed nuclear accumulation of mEGFP-β-catenin within 5 h of CHIR treatment, regardless of the contractile status of the cells. However, while β-catenin signal decreased over time in vehicle-treated cells, remaining visible after 20 h in only a few nuclei, low contractile cells showed a robust and enhanced nuclear localization (Fig. 6, B and C; and Video 3).

Knowing that β-catenin is enriched in the nucleus upon low contractility, we next focused on the ability of β-catenin to interact with its binding partners and to regulate WNT-responsive genes. Because of the timing of our phenotype, we ruled out an involvement of β-catenin interactions with transcription factors such as SOX family, SMAD, or TBX3 (Funa et al., 2015; Zimmerli et al., 2020), as they are involved later during specification (Mukherjee et al., 2022). In addition, ChIP-seq experiments report that TCFs/β-catenin binding is the main event during pluripotency exit and early specification (Moreira et al., 2017; Mukherjee et al., 2022). Mammalian genomes encode four TCF proteins (TCF7, TCF7L1, TCF7L2, and LEF1), all of which bind to similar DNA sequences (Cadigan and Waterman, 2012), show redundant functions (Moreira et al., 2017; Gerner-Mauro et al., 2020), and co-occupy similar DNA regions (Guo et al., 2021; Blassberg et al., 2022). TCF3 (encoded by *TCF7L1*) is the most highly expressed member in hiPSCs (Fig. S7 D) but is widely reported to be repressive (Pereira et al., 2006; Cole et al., 2008; Yi et al., 2008; Wray et al., 2011). Therefore, we focused on the second most highly expressed factor, TCF4 (encoded by *TCF7L2*). Nuclei isolated from Y-27632–treated EGFP-β-catenin knock-in hiPSCs showed higher binding between β-catenin and TCF4 compared with vehicle-treated cells (Fig. 6, E and F).

Finally, CUT&RUN assays showed that low contractility significantly increased β-catenin recruitment to the *TBXT* promoter during differentiation (Fig. 6 G and Fig. S7 E), supporting a sensitization model. Together, these findings suggest that the promotion of nuclear β-catenin accumulation by cytoskeletal relaxation accelerates the mesoderm program by enhanced binding to its TCF comediator. However, this model can only hold true if the concentration of CHIR used in our study (7.5 µM) does not saturate the β-catenin signaling response. To test this premise, we doubled the concentration of CHIR and probed for expression of WNT target genes as a readout for WNT signaling (Fig. S7 F). *LEF1* and *AXIN2* expression responded in a dose-dependent manner to CHIR. Interestingly, however, at any given CHIR concentration, inhibition of cell contractility further enhanced WNT target gene responses.

Collectively, we demonstrate that actomyosin contractility regulates β-catenin availability and that cell relaxation amplifies active β-catenin translocation to the nucleus synergistically with CHIR to enhance mesoderm commitment (Fig. 7).

## Discussion

In this study, we have identified an inverse relationship between intrinsic actomyosin contractility and mesoderm specification in hPSCs, independent of extracellular matrix properties or external mechanical perturbations. In addition, we identified that hiPSCs can sense contractility at the pluripotent stage, which will affect their ability to differentiate later. Mechanistically, our data support a model in which AJs serve as dynamic reservoirs that sequester β-catenin away from the cytoplasmic signaling pool. We propose that the interdependence of actomyosin contractility and AJs creates a titration mechanism for β-catenin availability, thereby tuning mesoderm specification.

Actomyosin dynamics has previously been reported to control differentiation across several lineages. For example, Engler et al. reported myosin II–dependent fate biasing in adult mesenchymal stem cells (Engler et al., 2006), while others showed actomyosin-mediated endoderm differentiation in PSCs (Toyoda et al., 2017; Jiang et al., 2024). Although in vivo validation is not feasible in human embryos, studies in chick and mouse embryos demonstrate that spatially patterned actomyosin contractility and tension differences are essential for early cell sorting and lineage specification (Rozbicki et al., 2015; Maître et al., 2016). Since these seminal papers, the impact of the cytoskeleton dynamics on early development is now broadly recognized (Lim and Plachta, 2021; Nelson, 2022).

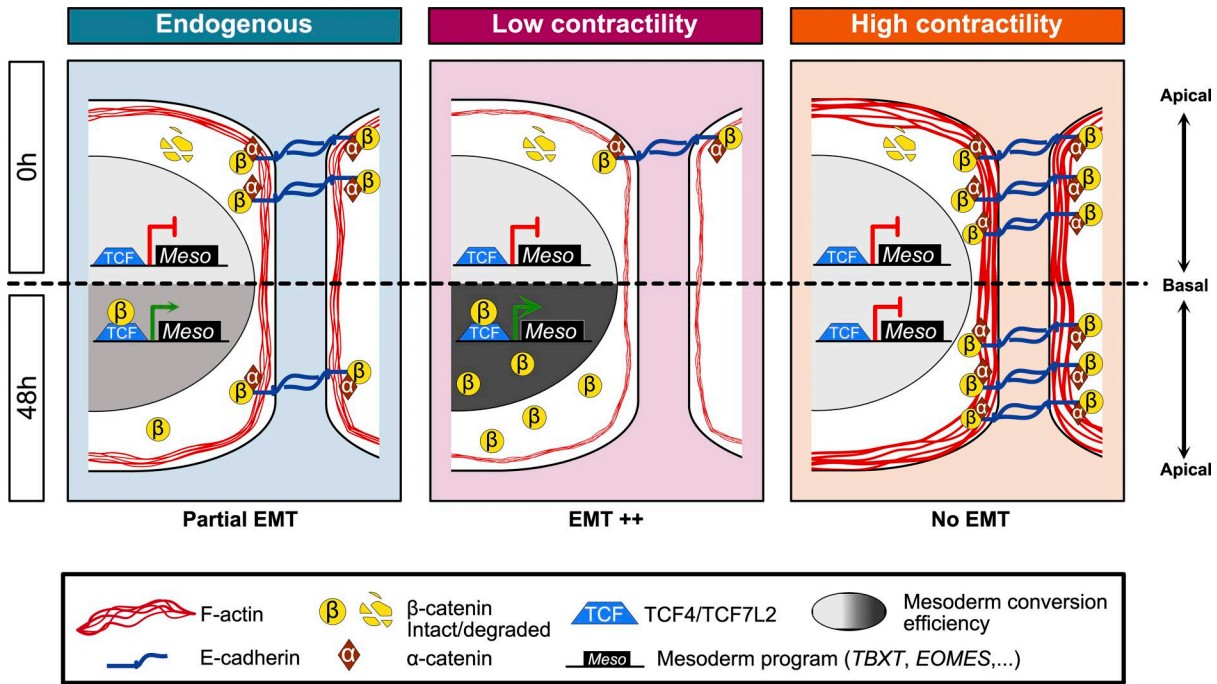

**Figure 7. Working model.** Effect of actomyosin contractility on mesoderm lineage commitment. Each panel represents a specific contractile status at the basal state (0 h—top) and after 48 h (bottom) of CHIR treatment. At the basal state, β-catenin is degraded and localized at AJs. The amount of junctional β-catenin correlates with the level of contractility experienced by hiPSCs. During differentiation, β-catenin accumulates in the nucleus and activates a mesoderm program through its binding with TCF4.

Specific to the mesoderm lineage, previous papers have reported that contrary to our observations, *increased* mechanical tension promotes β-catenin–dependent mesoderm induction. For example, tension has been described to promote Brachyury expression in hESCs (Przybyla et al., 2016), Bilateria (Brunet et al., 2013), and Cnidaria (Pukhlyakova et al., 2018), while in gastrulation-like models of hESCs, Muncie et al. reported that high cell adhesion tension on a compliant substrate can promote mesoderm induction (Muncie et al., 2020). This study proposed a dual function for mechanical tension. First, BMP4-driven Brachyury expression relies on increased mechanical stretch of cell junctions, allowing destabilization of β-catenin from AJs and its nuclear translocation. Following this initiation stage, a positive feedback loop reinforces mesoderm commitment through WNT ligand secretion. Strikingly, on Matrigel-coated plates, we observe the opposite outcome.

When comparing our findings to those of Muncie et al., we identified three key differences in the experimental design. First, we ruled out cell line–specific effects. While Muncie et al. relied exclusively on hESCs, most of our experiments were performed using hiPSCs and major conclusions were validated in H9 hESCs (Fig. S2, N–Q; and Fig. 3, K and L). Importantly, the FACS analyses (Fig. 3, K and L) were conducted using an hESC knock-in reporter line generated by the Weaver laboratory and generously shared with us. A second major difference lies in the mesoderm differentiation strategy. Whereas Muncie et al. used BMP4/bFGF, our primary protocol relies on CHIR-mediated β-catenin stabilization via inhibition of GSK3β (a similar mechanism to that of canonical WNT ligands). To directly address this difference, we extended our analysis to BMP4-driven mesoderm induction and observed a comparable increase in Brachyury expression upon ROCK inhibition. Finally, the two studies were performed on markedly different substrates, which may represent the most likely explanation for the discrepancies between the datasets. Importantly, our unconstrained system directly focuses on contractility rather than tension geometry or substrate properties. Our findings suggest that intrinsic contractility, even in unpatterned settings, is sufficient to guide cell fate decisions.

A question remains regarding the mechanisms by which CHIR induces contractility, also observed by others (Hookway et al., 2024, *Preprint*). While the noncanonical WNT pathway is well known to interface with RhoA/ROCK signaling and to influence myosin II–driven contractility, no direct relationship for the canonical WNT pathway has been established. Although WNT3A has been reported to elevate RhoA activity and ROCK phosphorylation in murine osteoblasts via partial Lrp5/6 dependence (Shi et al., 2021), the mechanism is not clear. GSK3 inhibition has also been shown to remodel focal adhesions (Dobson et al., 2023), raising the possibility that canonical WNT activation could indirectly reshape actomyosin tension.

First, we provide causal and cell-intrinsic evidence that actomyosin-generated forces act upstream of the mesodermal fate, in the pluripotent state, to inhibit cell specification. Second, our protocol directly activates the WNT pathway by inhibition of the destruction complex, as do WNT ligands, in contrast to the indirect effects of BMP4 activation. Importantly, a previous study showed that epithelial cohesiveness and tight junctions

block access by BMP4 to its receptor on the lateral plasma membrane (Vasic et al., 2023). Therefore, BMP4 signaling can only activate Brachyury expression at colony edges and corners (Muncie et al., 2020). However, our protocol uses a cell-permeable small molecule inhibitor as the differentiating cue, circumventing this diffusion barrier. Third, our findings distinguish our model from alternative force-mediated mechanisms such as Hippo/YAP signaling (Pagliari et al., 2021) or tension-driven WNT ligand secretion (Muncie et al., 2020), neither of which accounted for the enhanced mesoderm specification. One explanation might be that PSCs are insensitive to Hippo activation because of their inability to perceive cell–cell interactions, as previously reported (Pagliari et al., 2021). Finally, our system relies on activating or reducing contractility at the pluripotent state by pretreating cells prior to differentiation and maintaining treatment during cell commitment. We observed a timing-dependent response during which enhanced mesoderm identity was only achieved following ROCK inhibition pretreatment of cells in the pluripotent state. Therefore, our study suggests that low contractility sensitizes pluripotent cells to enter the mesoderm differentiation trajectory. Such a licensing mechanism might involve cis-regulatory element-mediated chromatin remodeling that can potentiate gene activity prior to specification events as reported in other contexts (Maytum et al., 2024). Alternatively, we note that actomyosin-dependent epigenome regulation was previously associated with RNA polymerase II activity and lineage commitment in adult stem cells (Le et al., 2016).

We here propose a titration mechanism by actomyosin contractility that allows a synchronization with developmental processes during cell commitment. β-Catenin exists in two pools, one bound to AJs through E-cadherin and one cytosolic pool that is central to WNT signaling. The regulation and exchange of β-catenin molecules between these two pools remain poorly understood. E-cadherin binding to β-catenin prevents its nuclear translocation and transcriptional co-activator function (Orsulic et al., 1999). Src-mediated phosphorylation of β-catenin at residue Y654 has been reported to reduce binding to E-cadherin (Roura et al., 1999), so one possibility might be that Src tyrosine kinase activity is anti-dependent on contractility in PSCs. Alternatively, however, Gayrard et al. (2018) reported an inverse mechanism through which, in migrating cells, Src-dependent phosphorylation of FAK causes actin remodeling and E-cadherin tension relaxation, resulting in the release and nuclear translocation of β-catenin. In our system, reduced tension did not affect junctional β-catenin kinetics (Fig. S6, G–K), reflecting that actomyosin contractility likely regulates AJ organization and reinforcement, as previously reported (Yonemura et al., 2010; Mège and Ishiyama, 2017; Röper et al., 2018; Hall et al., 2019), rather than altering turnover rates of the junctional β-catenin pool. Instead, lowering actomyosin contractility causes loss of β-catenin from AJs, accumulation in the cytosolic compartment, and translocation into the nucleus. It is however possible that actomyosin contractility affects β-catenin nucleocytoplasmic transport dynamics as we noticed a sustained nuclear accumulation in Y-27632–treated cells (Fig. 6, A–C), which would

require further investigation. Finally, in *Xenopus*, decreased ROCK-dependent actomyosin contractility promotes an ectoderm-to-mesoderm transition, suggesting force-dependent lineage plasticity is a conserved mechanism (Kashkooli et al., 2021). While we have not tested for alternative cell fate following CHIR addition, our differentiation protocol was adapted from Lian et al. (2012); Lian et al. (2013) and was not reported to lead to ectodermal identity.

## Materials and methods

### Research compliance and ethical regulation

All experiments using hESCs were performed using the WA09 (H9) cell line under the supervision of the Vanderbilt Institutional Human Pluripotent Cell Research Oversight Committee (protocol IRB no. 160146). No cells were sourced directly from human embryos.

### Reagents

Common laboratory reagents used in this study are listed in Table S1.

### Cell lines, cell culture, and maintenance

Human GM25256 iPSCs were obtained from the Coriell Institute (RRID:CVCL_Y803). The mEGFP-CTNNB1 knock-in GM25256 iPSC line was obtained from the Allen Cell Collection, Coriell Institute (cell ID AICS-0058-067, RRID: CVCL_VK86). The hESC line H9 (WA09) was obtained from the WiCell Research Institute (RRID:CVCL_9773). Cell line authentication was not performed. The T-mNG knock-in hESC H9 was made by the Valerie Weaver lab (UCSF) and kindly provided by Margaret Gardel (University of Chicago, Chicago, IL, USA). The iPSC and ESC H9 lines were cultured on Matrigel-coated six-well plates (Matrigel diluted at 42 µg ml−1 in DMEM/F12 medium) and cultured in mTeSR1 medium. The medium was changed daily until the cells reached 70% confluency. Cells were passaged using gentle cell dissociation reagent for 4 min, resuspended in mTeSR1 medium as small clusters, and replated at 1:7. HEK293T cells were obtained from the ATCC (RRID:CVCL_0063). Cells were grown in 10% FBS/DMEM and passaged at 1:10 every 2–3 days.

All cell lines used in this study were maintained at 37°C under 5% $CO_2$.

Sex is not expected to be a biological variable in this system. In addition, key experiments obtained in hiPSC WTC11 (sex: male) were repeated in hESC H9 (sex: female).

### Cell freezing and thawing

Human iPSCs and ESCs were harvested from culture dishes using gentle cell dissociation reagent and centrifuged at 120 g for 3 min. The pellets were resuspended in mTeSR1 medium supplemented with 10% DMSO and aliquoted into cryovials. Cells were first transferred to −80°C for 24 h before long-term storage in liquid nitrogen. iPSCs and ESCs were slowly thawed using mTeSR1, centrifuged, and resuspended in mTeSR1 supplemented with 10 µM Y-27632 for 24 h.

## Mesoderm differentiation protocol and drug treatments

Differentiation was initiated when cells reached 70–80% confluency. Cells were treated with 7.5 µM CHIR99021 (hereafter CHIR) diluted in RPMI 1640 media supplemented with 1X B27 minus insulin and 100 U/ml penicillin and 100 µg/ml streptomycin (1X). For BMP4-driven differentiation, 50 ng/ml BMP4 and 100 ng bFGF were diluted in RPMI 1640/B27(-Ins) with 1X penicillin/streptomycin.

For pharmaceutical inhibition of contractility, cells (50% confluent) were pretreated overnight or for 3 h with drug or vehicle diluted in mTeSR1. Differentiation was initiated the next day (when cells reached 70–80% confluent) by switching to CHIR medium as described above, supplemented with the drug or vehicle. For MYPT1$^{CA}$, cells were preinduced overnight with 1 µg/ml Dox or vehicle (water) diluted in mTeSR1 before initiating the differentiation protocol in the presence of Dox or water.

For pharmaceutical stimulation of cell contractility, 70–80% confluent cells were pretreated for 3 h with 4 µg/ml CN03 or water in mTeSR1, followed by CHIR treatment supplemented with CN03 or water. For EGTA treatment, 70–80% confluent cells were pretreated for 30 min with 1 mM of EGTA or water in mTeSR1, followed by CHIR treatment supplemented with 10 µM EGTA or water. For concentrations of each drug, please refer to Table S1.

## Lentiviral preparation

HEK293T cells were grown at 40% confluency in 10-cm plates and transfected using calcium phosphate. Briefly, 20 µg of lentiviral plasmid, 15 µg of pSPAX2 (RRID:Addgene_12260), and 6 µg pMD2G (RRID:Addgene_12259) were mixed with 450 µl sterile water and 50 µl of 2.5M CaCl$_2$ solution, previously filter-sterilized. DNA mixture was added dropwise into 500 µl of 2X HeBS (50 mM HEPES, 10 mM KCl, 12 mM dextrose, 280 mM NaCl, and 1.5 mM Na2PO4, pH 7.04) with constant vortexing. This solution was added dropwise on HEK293T cells. After 6 h, the medium was replaced with 8 ml of fresh 10% FBS/DMEM for 48 h. The supernatant was filtered through a 0.22-µm membrane and concentrated using a 100 kDa cutoff Amicon centrifugal unit. Aliquots were frozen at –80°C.

For generation of CRISPR cells, single-guide RNAs were selected using the Benchling design tool (RRID:SCR_013955), and are listed in Table S3. Annealed oligonucleotides were cloned into pLentiCrispRv2-Puro (RRID:Addgene_52961) as described by Sanjana et al. (2014). Lentiviral particles were prepared as described above. hiPSCs were transduced in suspension and selected 48 h after transduction using 1 µg/ml of puromycin.

## SDS-PAGE and western blotting

Cells were washed with ice-cold 1X PBS and lysed with 70 µl RIPA buffer (150 mM NaCl, 10 mM Tris–HCl, pH 7.5, 1 mM EDTA, 1% Triton X-100, 0.1% SDS) supplemented with 1X protease and phosphatase inhibitors. Mechanical disruption was performed by scraping the cells off the plate, transferring the lysate into a 1.5-ml Eppendorf tube, and vortexing for 5 s followed by 5-min incubation on ice. Soluble proteins were collected by centrifugation at 16,000 *g* (13,200 rpm) for 10 min at

4°C. Protein concentration was measured using Precision Red following the manufacturer's instructions. Lysates in sample buffer were boiled for 5 min.

For ppMLC blotting, cells were washed with 1X PBS and directly lysed using 250 µl of boiling 1X sample buffer. Lysates were sonicated 3 times for 10 s with 60-s incubation on ice between each blast. 30 µg of proteins was resolved on 1.5-mm-thick Bis-Tris acrylamide gels and transferred onto a 0.2-µm nitrocellulose membrane using the Bio-Rad wet transfer system (50 V for 2 h). Membranes were blocked for >30 min at room temperature (RT) with 5% BSA in 1X TBS-T (10 mM Tris, pH 8.0, 150 mM NaCl, and 0.5% Tween-20) and incubated overnight at 4°C with gentle rocking with primary antibodies diluted in blocking buffer (Table S2). Membranes were washed three times with 1X TBS-T and incubated for 1 h at RT with Alexa Fluor–conjugated secondary antibodies (Table S2). After 3 additional washes in 1X TBS-T, membranes were scanned using the LI-COR Odyssey CLx system, analyzed, and processed using Image StudioLite v. 5.2 (RRID:SCR_013715).

## Immunofluorescence

Cells were cultured on Matrigel-coated Ø 12-mm coverglass, fixed with 4% PFA for 10 min, permeabilized for 5 min at RT (20 mM glycine, 0.05% Triton X-100 in 1X PBS), and incubated with blocking buffer for >30 min (5% BSA in 1X PBS). Coverslips were transferred into a dark, humidified chamber and incubated with primary and Alexa Fluor–conjugated secondary antibodies (Table S2) diluted in blocking buffer for 2 h. Coverslips were washed in 1X PBS and mounted on glass slides using Fluoromount-G mounting solution.

For ppMLC staining, samples were incubated overnight with the primary antibody.

## Image acquisition, analysis, and processing

Micrographs were obtained using an inverted Nikon A1-R confocal microscope equipped with a ×20 objective (numerical aperture (NA) 0.75), a ×40 oil objective (NA 1.20), or a ×100 oil objective (NA 1.40). Z-stacks covering the entire cell height were acquired. Maximum-intensity projections (MaxIPs) were obtained, and images were analyzed and processed using Fiji (version 2.1.0/1.54f, RRID:SCR_002285). Image processing did not include a denoising step except for Video 3.

Quantification of β-catenin intensity was performed by generating summed intensity images, and applying a fixed Fiji threshold defined from the control condition for each channel (β-catenin and ppMLC), normalized to the cellular area. Cell density was similar across treatments.

## MYPT1$^{CA}$-NES-mNG cloning strategy and cell line generation

A gBlock was obtained from IDT, corresponding to truncated human MYPT1 (aa 1–300) in fusion with a PKI super nuclear export signal and mNG sequence. The gBlock was amplified using fusion polymerase and the following primers: Fwd 5′-CTG CTGACCGGTACCATGGCGGACGCGAAGC-3′ and Rev 5′-CAT CATACGCGTCTACGATCCGCCACCGC-3′. The PCR product and recipient backbone (pInducer10b-HA-KRas G12V, RRID:Addgene_164928) were digested with AgeI-HF and MluI-HF for 1 h at

37°C, gel-purified, and ligated overnight at 16°C using T4 ligase. Ligation reactions were transformed into chemically competent *Escherichia coli* Stbl3 strain and selected on ampicillin plates. Plasmids were isolated from single colonies (QIAprep Spin MiniPrep Kit). The presence of the insert was confirmed by restriction digestion. Positive clones were sequenced by Genewiz Plasmid-EZ service, before plasmid purification using Takara Bio NucleoBond Xtra Midi Kit. WT hiPSCs were transduced as described above, and cells were selected 48 h later with 1 µg/ml of puromycin.

### Generation of ROCK2^CA and MLCK^CA cell lines and coculture setup

ROCK2^CA (RRID:Addgene_84649) and MLCK^CA (RRID:Addgene_84647) constructs, and the empty vector (RRID:Addgene_25734) were obtained from Addgene and used to generate lentiviral particles. hiPSCs were transduced, and clonal populations were obtained. Note that mVenus expression is constitutive and used as a marker for transduction, while the expression of ROCK2^CA and MLCK^CA is inducible. mVenus-positive cells were mixed with WT (mVenus-negative hiPSC) at a 1:1 ratio. Cells were treated with 1 µg/ml Dox or vehicle overnight prior to CHIR treatment in the presence or absence of Dox. To account for a nonspecific background in the WT population, mVenus-positive cells were defined as cells with a green signal covering >15% of their area.

### Quantitative polarization microscopy

The contractility of cells was measured using QPOL as described previously (Wang et al., 2018). Briefly, QPOL was built on an inverted Axiovert microscope equipped with an Axiocam 506 color camera. A motorized linear polarizer (Thorlabs) and a circular polarizer were positioned in the illumination plane above the condenser and in the imaging plane, respectively. Images were captured using a 20×/0.5 NA polarization objective. For each field of view, image sequences were acquired with 10° intervals of the rotating polarizer over the range of 0–180° using Zen software (RRID:SCR_013672). The polarized image sequences were then processed using a custom MATLAB code to generate pixel-by-pixel retardance maps. The obtained retardance images then underwent background subtraction and subsequent analysis with Fiji (RRID:SCR_002285). The retardance signal proportional to cell contractility was quantified as the average value of the background-subtracted retardance over the cell area.

### Flow cytometry analysis

T-mNG knock-in hESCs were transduced with a lentivector expressing Histone 2B-mScarlet (pWPI-H2B-mScarlet). Cells were pretreated overnight with vehicle (water) or 10 µM Y-27632. Differentiation was initiated for 2 h as described above, and cells were harvested as single cells. Briefly, cells were washed and gentle cell dissociation buffer was added for 8 min at 37°C. Cell suspension was centrifuged and resuspended in 500 µl PBS. Cells were passed through a Ø 40-µm strainer directly into 500 µl of 8% PFA (final concentration 4%) and fixed for 10 min on a wheel before sending them for cytometry using a five-laser Fortessa analyzer (70-µm nozzle). Singlets were selected based on forward and side scatter. mScarlet (PE-Texas-RedA) and mNG (GFP-A) population were selected by drawing a gate on double-negative WT hESCs shifted by one order of magnitude. The percentage of mNG/mScarlet double positive was reported using the online flow cytometry analysis resource (https://floreada.io; RRID:SCR_025286).

### RNA isolation and RT-qPCR

Cells were washed, and RNAs were isolated using RNeasy Mini Kit following the manufacturer's instructions. 1 µg of RNA was reverse-transcribed to cDNA using the SuperScript III first-strand synthesis kit. cDNA was diluted 1:5 in water and mixed with Maxima SYBR Green/1 mM of each primer (Table S3). Quantitative PCR was performed on a Bio-Rad CFX96 thermocycler with thermal cycling conditions as follows: initial denaturation: 95°C for 10 min followed by 40 cycles of denaturation at 95°C for 15 s, annealing at 58°C for 30 s, extension at 72°C. The final step was performed for 10 s at 95°C. $C_t$ values from technical triplicates were averaged and normalized to GAPDH using the $\Delta\Delta C_t$ formula.

TCF expression levels (Fig. S7 D) were curated from the transcriptomic data available on the Allen Institute website (https://www.allencell.org/genomics.html).

### Nuclear extraction and CUT&RUN assay
#### Nuclear extraction

500,000 cells (per sample) were harvested using TrypLE, washed with 1X PBS, and centrifuged at 600 × *g* for 3 min at RT. The pellet was resuspended in Nuclear Extraction (NE) buffer (20 mM HEPES, pH 7.9, 10 mM KCl, 0.1% Triton X-100, 20% glycerol, 1 mM MnCl₂, 0.5 mM spermidine, 1X Roche cOmplete Protease Inhibitor EDTA-Free) and incubated for 10 min at 4°C. After incubation, samples were centrifuged at 600 × *g*, and the isolated nuclei were resuspended in 100 µl of NE buffer per sample. Nuclei were incubated with 10 µl of magnetic concanavalin A beads for 10 min at RT to facilitate binding.

#### DNA purification

Bead-bound nuclei were incubated with 3 µl of anti-β-catenin antibody in 50 µl of digitonin buffer (20 mM HEPES, pH 7.5, 150 mM NaCl, 0.5 mM spermidine, 0.01% digitonin, 1X Protease Inhibitor Cocktail) containing 2 mM EDTA overnight at 4°C. Following incubation, samples were washed twice with digitonin buffer and resuspended in 50 µl of the same buffer supplemented with 1.5 µl of pAG-MNase. Samples were incubated for 10 min at RT, followed by two additional washes, and subjected to chromatin digestion at 4°C for 2 h with 2 mM CaCl₂. To quench the reaction, 33 µl of STOP buffer (340 mM NaCl, 20 mM EDTA, 4 mM EGTA, 50 µg/ml RNase A, 50 µg/ml glycogen) was added, followed by incubation at 37°C for 10 min. Beads were collected using a magnetic rack, and the supernatant containing released DNA was transferred to a new tube. DNA was purified using DNA purification buffers and spin columns following the manufacturer's instructions and eluted in 30 µl of buffer.

### qPCR and data analysis

Reactions were prepared using 1 µl of purified DNA from the CUT&RUN assay, with 2 µl forward and reverse primer mix (5 µM each), 5 µl SYBR Green Master Mix (Applied Biosystems), and 2 µl of nuclease-free water. Thermal cycling conditions were set up on a Bio-Rad CFX96 thermocycler as follows: initial denaturation: 95°C for 10 min followed by 40 cycles of denaturation at 95°C for 15 s, annealing at 60°C for 30 s, extension at 72°C. The final step was performed for 10 s at 95°C. Ct values from technical triplicates were averaged, and ΔCt was calculated by subtracting Ct IgG control from target. Fold enrichment was calculated using the ΔΔCt method: fold enrichment = $2^{-\Delta\Delta Ct}$.

### GFP-Trap immunoprecipitation

EGFP-CTNNB1 knock-in hiPSCs were harvested using TrypLE, and nuclei were isolated as described above. For each reaction, two wells were collected. Both 0- and 24-h nuclei were cryopreserved by slowly freezing them in NE buffer using an isopropanol-filled chiller at –80°C. Nuclei were quickly thawed using a 37°C water bath, centrifuged at 600×$g$, and resuspended using 100 µl of lysis buffer (10 mM Tris–HCl, pH 7.5, 150 mM NaCl, 0.5 mM EDTA, 0.5 % NP-40). Nuclear lysates were incubated for 5 min on ice and cleared by centrifugation (16,000×$g$ for 10 min at 4°C). The soluble protein concentration was measured using Precision Red following the manufacturer's instructions, and pellets were mixed with 1X sample buffer (insoluble fraction). Meanwhile, GFP-Trap beads (20 µl/reaction) were equilibrated 3 times with 500 µl of ice-cold dilution buffer (10 mM Tris–HCl, pH 7.5, 150 mM NaCl, 0.5 mM EDTA). 150–200 µg of soluble proteins was prepared in 500 µl of ice-cold dilution buffer and mixed with equilibrated GFP-Trap beads. Reactions were incubated for 2 h at 4°C on a wheel. Beads were then pelleted (2,500×$g$, 3 min at 4°C) and washed 3 times with 500 µl of ice-cold wash buffer (10 mM Tris–HCl, pH 7.5, 150 mM NaCl, 0.05 % NP-40, 0.5 mM EDTA). Proteins were eluted by adding 50 µl of 2X sample buffer and boiled for 5 min. Eluted proteins were resolved by western blot as described above. TCF4 binding was measured by densitometry of the TCF4 band (prey) normalized to the GFP band (bait).

### Fluorescence recovery after photobleaching

Cells were grown on eight-well Lab-Tek II #1.5 chambers coated with Matrigel. Chambers were placed within a stage-top incubator mounted on a Nikon A1R set at 37°C and 5% $CO_2$. A circular region of interest (ROI) corresponding to the location of photobleaching was created. Two frames were acquired prior to photobleaching as an initial reference for fluorescence intensity, using a 40× oil objective (NA 1.20) and a 6× zoom. FRAP ROI was stimulated with a laser corresponding to the fluorophore being bleached for 7 s. Immediately after photobleaching, recovery frames were acquired every 5 s for 2.5 min.

### Reproducibility and statistics

All datasets were analyzed using GraphPad Prism 10 (version 10.0.3). Individual measurements (n) were averaged for each biological repeat (N). Datasets were tested for normality (Shapiro–Wilk test for normality of biological replicates) before applying the appropriate statistical test on the biological repeats (N), except if mentioned differently in the figure legend. Error bars represent the SD except where stated otherwise. For datasets displayed as superplots, individual measurements and biological repeats are represented by small and large dots, respectively. Violin plots display median (thick dotted line) and quartiles (thin dotted lines). Each dataset for a biological repeat is color-coded. Significance levels are given as follows, and exact P values are indicated in each figure: n.s., not significant (P ≥ 0.05); *P < 0.05; **P < 0.01; ***P < 0.001; and ****P < 0.0001.

All experiments, including representative MaxIPs and western blots, were performed at least three times independently as biological repeats, unless stated otherwise in the legends. No data were excluded. No statistical method was used to predetermine the sample size, the experiments were not randomized, and the investigators were not blinded to allocation during experiments and outcome assessment.

### Online supplemental material

Fig. S1 shows that pharmaceutical increases in cell contractility block mesoderm specification and also provide additional controls for the genetic modulation of contractility. Fig. S2 shows effects of actomyosin inhibition on mesoderm and EMT in hiPSCs and hESCs, using CHIR or BMP4/bFGF-based differentiation protocols (related to Fig. 3). Fig. S3 reports that actomyosin-driven contractility increases during mesoderm differentiation. Fig. S4 provides data supporting that key mechanotransduction pathways such as the Hippo pathway and force-dependent WNT ligand secretion are not affected in this study. Fig. S5 shows effect of EGTA treatment on differentiation and validation of *CTNNA1* KD (related to Fig. 4). Fig. S6 shows the effect of contractility on junctional β-catenin localization and dynamics (related to Fig. 5). Fig S7 shows that reduced cell contractility promotes nuclear accumulation of β-catenin and increased WNT pathway activity (related to Fig. 6). Table S1 lists common lab reagents. Table S2 lists all antibodies and dyes used in this study, with their respective application and dilution factor. Table S3 lists all qPCR primers, CUT&RUN assay primers, and CRISPR guide oligos. Video 1 shows bright-field time-lapse imaging of hiPSC colony treated with CHIR supplemented with vehicle (DMSO) or Q-VD-OPH (cell death inhibitor). Video 2 shows time-lapse confocal imaging of mEGFP-β-catenin knock-in hiPSCs treated with drugs affecting cell contractility at the pluripotent state. Video 3 shows time-lapse confocal imaging of mEGFP-β-catenin knock-in hiPSCs treated with drugs affecting cell contractility during CHIR-mediated mesoderm differentiation. SourceDataF2, SourceDataF3, SourceDataF4, and SourceDataF6 show uncropped and unprocessed immunoblots related to Figs. 2, 3, 4, and 6, respectively. SourceDataFS2, SourceDataFS4, SourceDataFS5, and SourceDataFS6 show uncropped and unprocessed immunoblots related to Fig. S2, Fig. S4, Fig. S5, and Fig. S6.

## Data availability

The data are available from the corresponding author upon reasonable request.

## Acknowledgments

We thank Cynthia Reinhart-King for giving us access to the QPOL facility. We thank the Valerie Weaver lab (UCSF, San Francisco, CA, USA) and the Margaret Gardel lab (University of Chicago, Chicago, IL, USA) for sharing the T-mNG knock-in hESC line. We thank the Vivian Gama lab (Vanderbilt University, Nashville, TN, USA) for providing the neural stem cell lysate, and Pax6 and Otx1/2 antibodies. We also thank all members of the Macara lab, especially Dr. Christian de Caestecker, for valuable suggestions and advice.

This work was funded by an American Heart Association postdoctoral fellowship (Award ID 23POST1018347 to L. Fort), a NCI Predoctoral-to-Postdoctoral Fellow Transition Award (F99CA274695 to W. Wang), and a grant from the National Institutes of Health (GM070902 to I.G. Macara). Open Access funding provided by Vanderbilt University.

Author contributions: Loic Fort: conceptualization, data curation, formal analysis, funding acquisition, investigation, methodology, project administration, resources, supervision, validation, visualization, and writing—original draft, review, and editing. Vaishna Vamadevan: investigation and validation. Wenjun Wang: data curation, formal analysis, and validation. Ian G. Macara: conceptualization, funding acquisition, project administration, resources, supervision, and writing—review and editing.

Disclosures: The authors declare no competing interests exist.

Submitted: 15 July 2025

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

# Supplemental material

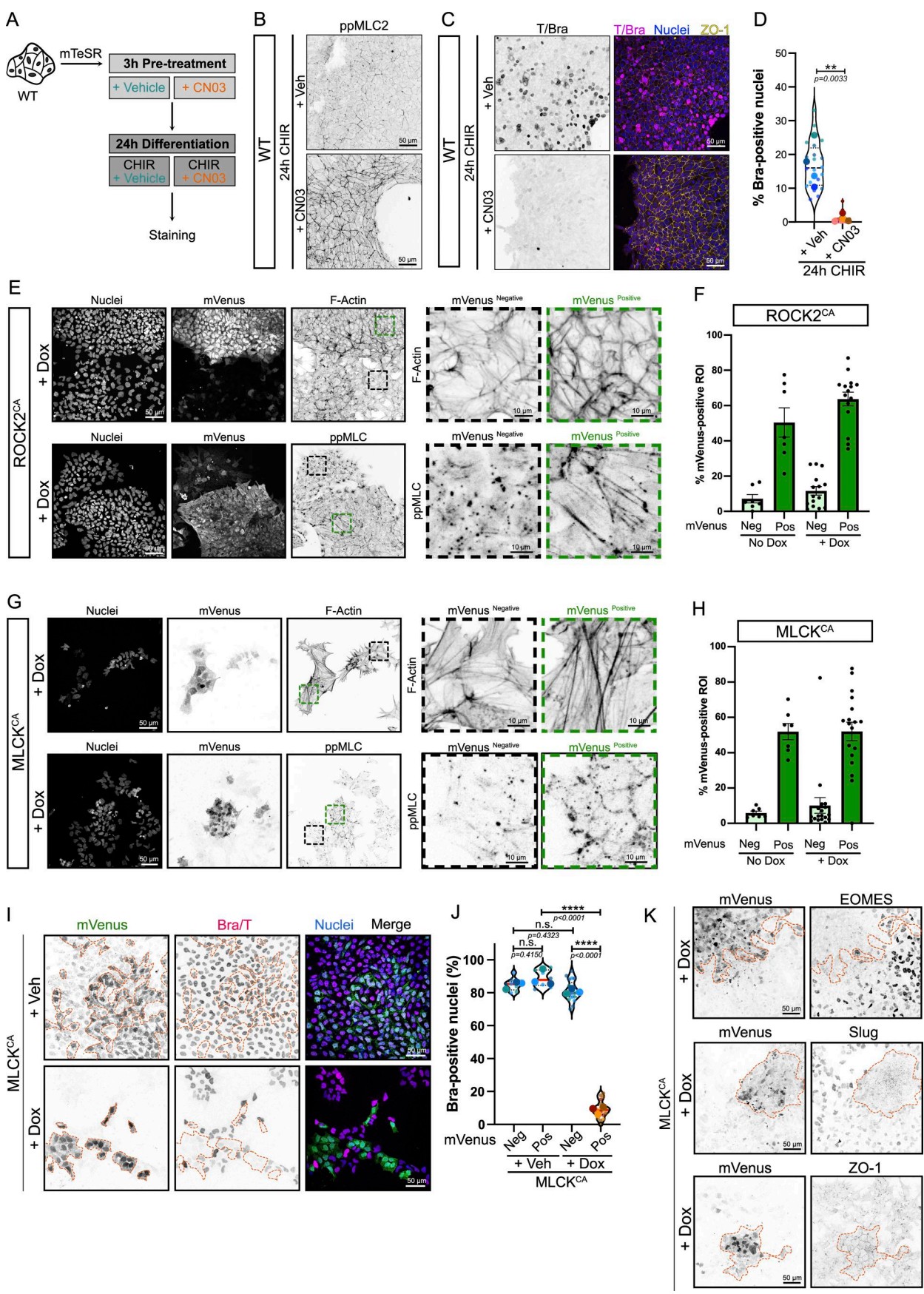

Figure S1. **Increased actomyosin contractility is sufficient to block mesoderm commitment and EMT (related to Fig. 1). (A)** Experimental design. hiPSCs were pretreated with 4 µg/ml CN03 or Veh for 3 h. Differentiation was initiated with CHIR supplemented with Veh or 4 µg/ml CN03 for 24 h before fixation and staining. **(B)** Representative MaxIP immunofluorescence of hiPSCs treated as explained in A and stained for phospho-MLC2 (inverted LUT). Scale bar = 50 µm. **(C and D)** Representative MaxIP immunofluorescence of hiPSCs treated as explained in A and stained for Brachyury (inverted LUT), nuclei, and ZO-1. Scale bar = 50 µm (C). Violin plot (median, quartiles) reporting the percentage of Brachyury-positive nuclei after 24 h of differentiation and quantified across $N$ = 4 independent biological repeats (representing $n$ = 20 technical repeats). An unpaired $t$ test was performed on biological replicates (D). **(E)** Representative MaxIP immunofluorescence image of Dox-induced ROCK2$^{CA}$ hiPSCs (mVenus-positive) mixed with WT hiPSCs (mVenus-negative). Cells were stained for DNA (nuclei), F-actin (top—inverted LUT), and ppMLC2 (bottom—inverted LUT). Scale bar = 50 µm. Magnified views comparing mVenus-positive (green dotted ROI) and mVenus-negative (black dotted ROI) areas are shown for F-actin and ppMLC2. Scale bar = 10 µm. **(F)** Percentage of the ROI area positive for mVenus based on cluster considered as mVenus-negative and mVenus-positive using the ROCK2$^{CA}$ cell line. On average, mVenus-negative clusters have <15% of their area positive for mVenus. Mean and SEM are displayed. $n$ = 7 (no Dox) and $n$ = 15 fields of view (+Dox) across $N$ = 3 independent biological experiments. **(G)** Representative MaxIP immunofluorescence of Dox-induced MLCK$^{CA}$ hiPSCs (mVenus-positive) mixed with WT hiPSCs (mVenus-negative). Cells were stained for DNA (nuclei), F-actin (top—inverted LUT), and ppMLC2 (bottom—inverted LUT). Scale bar = 50 µm. Magnified views comparing mVenus-positive (green dotted ROI) and mVenus-negative (black dotted ROI) areas are shown for F-actin and ppMLC2. Scale bar = 10 µm. **(H)** Percentage of the ROI area positive for mVenus based on clusters considered as mVenus-negative and mVenus-positive using the MLCK$^{CA}$ cell line. On average, mVenus-negative clusters have <15% of their area positive for mVenus. Mean and SEM are displayed. $n$ = 7 (no Dox) and $n$ = 18 fields of view (+Dox) across $N$ = 3 independent biological experiments. **(I and J)** Representative MaxIP immunofluorescence for Brachyury in +Veh and Dox-induced MLCK$^{CA}$ coculture. mVenus-positive cell clusters are highlighted by an orange dotted line. Individual channels are inverted LUT. Scale bar = 50 µm (I). Percentage of Brachyury-positive nuclei is reported as a violin plot, comparing mVenus-positive $vs$ mVenus-negative cluster in the presence or absence of Dox. Median (plain red line) and quartiles (dotted black lines) are displayed. $n$ = 7 (+Veh) and $n$ = 18 (+ Dox) fields of view (small dots) across $N$ = 3 independent biological repeats (large dots). One-way ANOVA with Tukey's multiple comparisons posttest was performed (J). **(K)** Representative MaxIP immunofluorescence for EOMES, Slug, and ZO-1 in Dox-induced MLCK$^{CA}$ coculture. mVenus-positive cell clusters are highlighted by an orange dotted line. Individual channels are inverted LUT. Scale bar = 50 µm. Veh, vehicle. ****$P$ < 0.0001, **$P$ < 0.01.

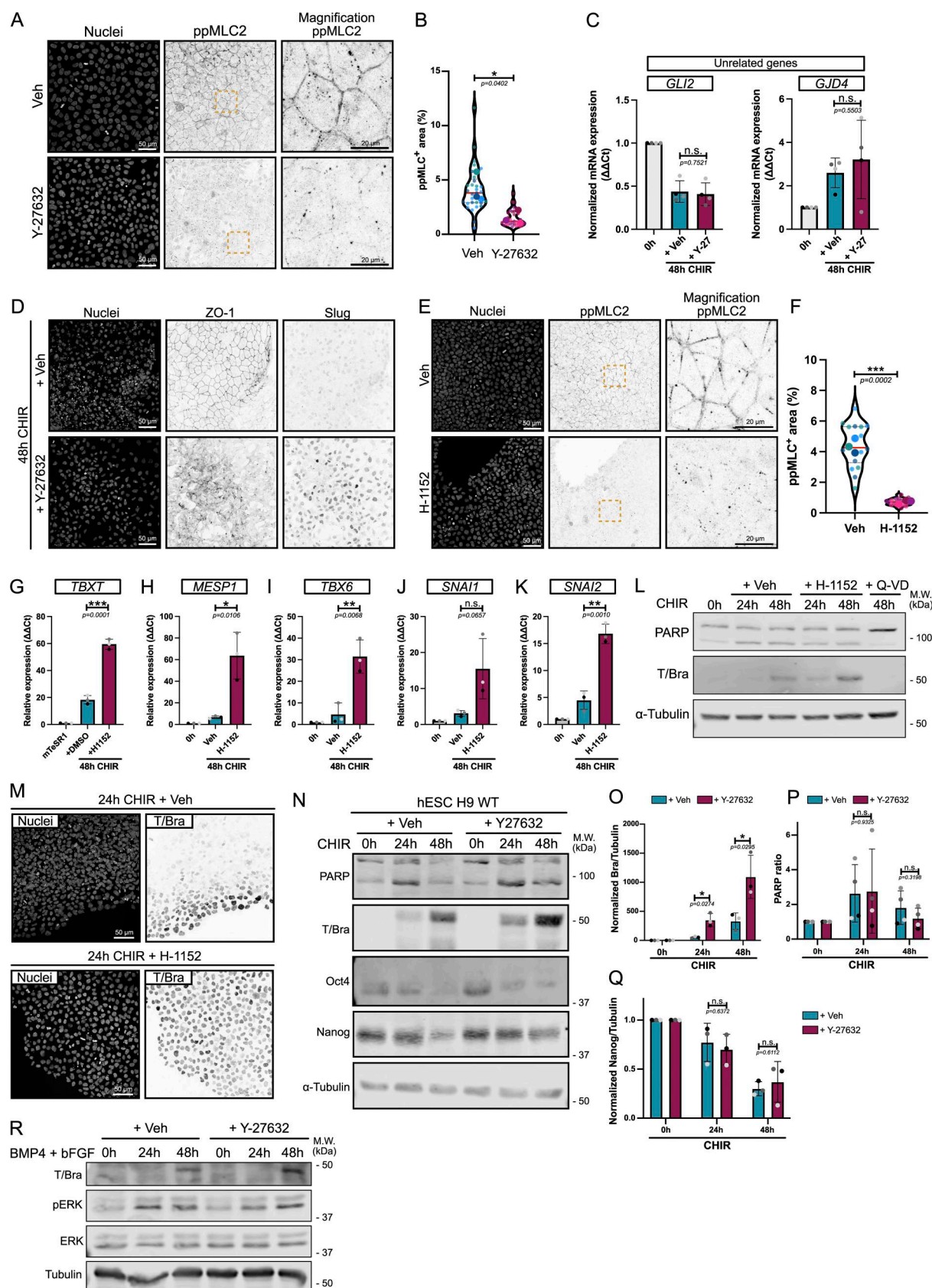

Figure S2.    **Pharmaceutical inhibition of actomyosin contractility promotes hiPSC and hESC conversion to the mesoderm lineage (related to Fig. 3). (A and B)** Representative MaxIP immunofluorescences of hiPSCs treated with a ROCK inhibitor (10 µM Y-27632) or Veh (H$_2$O) and stained for nucleus marker (DNA) and ppMLC2 (inverted LUT). Scale bar = 50 µm. Magnified views of the yellow dotted ROI are shown for ppMLC2. Scale bar = 20 µm (A). Fraction of the

cellular area positive for ppMLC2 following Y-27632 treatment and reported as violin plots. Median (plain red line) and quartiles (dotted black lines) are displayed. $n$ = 30 technical repeats across $N$ = 3 independent biological repeats. A two-tailed unpaired $t$ test was performed on the biological repeat (B). **(C)** Relative expression of genes not directly related to mesoderm commitment at 0 and 48 h after CHIR treatment in the presence (+Y-27) or absence (+Veh) of 10 µM of ROCK inhibitor. $N$ = 4 independent biological repeats. Mean and SD are displayed. A two-tailed unpaired $t$ test was performed. **(D)** Representative MaxIP immunofluorescences of hiPSCs treated 48 h with CHIR in the presence (+ Y-27632) or absence (+ Veh) of 10 µM of ROCK inhibitor. Cells were stained for nuclear marker (DNA), EMT markers (ZO-1 and Slug—inverted LUT). Scale bar = 50 µm. **(E and F)** Representative MaxIP immunofluorescences of hiPSCs treated with a ROCK inhibitor (1 µM H-1152) or Veh ($H_2O$) and stained for nuclear marker (DNA) and ppMLC2 (inverted LUT). Scale bar = 50 µm. Magnified views of the yellow dotted ROI are shown for ppMLC2. Scale bar = 20 µm (E). Fraction of the cellular area positive for ppMLC2 following H-1152 treatment and reported as violin plots. Median (plain red line) and quartiles (dotted black lines) are displayed. $n$ = 15 technical repeats across $N$ = 3 independent biological repeats. A two-tailed unpaired $t$ test was performed on the biological repeats (F). **(G–K)** Relative expression of mesoderm markers (*TBXT, MESP1, TBX6*) (G–I) and EMT markers (*SNAI1, SNAI2*) (J and K) 48 h after CHIR treatment in the presence (+H-1152) or absence (+Veh) of 1 µM of ROCK inhibitor. $N$ = 3 independent biological repeats. Mean and SD are displayed. A two-tailed unpaired $t$ test was performed. **(L)** Immunoblot probing for mesoderm marker expression (T/Bra) and cell death marker (PARP cleavage) during hiPSC-to-mesoderm commitment (0- to 48-h CHIR) in the presence of 1 µM of ROCK inhibitor (+H-1152). The addition of the pan-caspase inhibitor Q-VD-OPH (+ Q-VD) to the differentiation medium totally blocks mesoderm commitment and was used as a control. α-Tubulin was used as a loading control. M.W. are displayed on the right side. **(M)** Representative MaxIP immunofluorescences for mesoderm marker Brachyury (T/Bra—inverted LUT) and nuclei. Cells were fixed 24 h after CHIR treatment in the presence or absence of ROCK inhibitor H-1152. Scale bar = 50 µm. **(N)** Representative immunoblot of PARP (cell death), T/Bra (mesoderm), and Oct-4 and Nanog (pluripotency markers) using hESC H9 in the presence (+ Y-27632) or absence (+ Veh) of 10 µM of ROCK inhibitor, as shown in Fig. 3 A. M.W. are displayed on the right side. **(O–Q)** Brachyury expression (O), PARP cleavage (P), and Nanog expression (Q) were quantified by densitometry and normalized to α-tubulin as a loading control across $N$ = 4 (PARP) and $N$ = 3 (Brachyury and Nanog) independent biological repeats. Mean and SD are displayed. A two-tailed unpaired $t$ test was performed. **(R)** Expression of T/Bra was analyzed by western blot following treatment with a combination of 50 ng/ml BMP4 and 100 ng/ml bFGF, in the presence or absence of 10 µM Y-27632. pERK and total ERK were used as a positive control for FGF pathway activity. M.W. are shown on the right-hand side. M.W., molecular weights; Veh, vehicle. \*\*\*$P < 0.001$, \*\*$P < 0.01$, \*$P < 0.05$. Source data are available for this figure: SourceDataFS2.

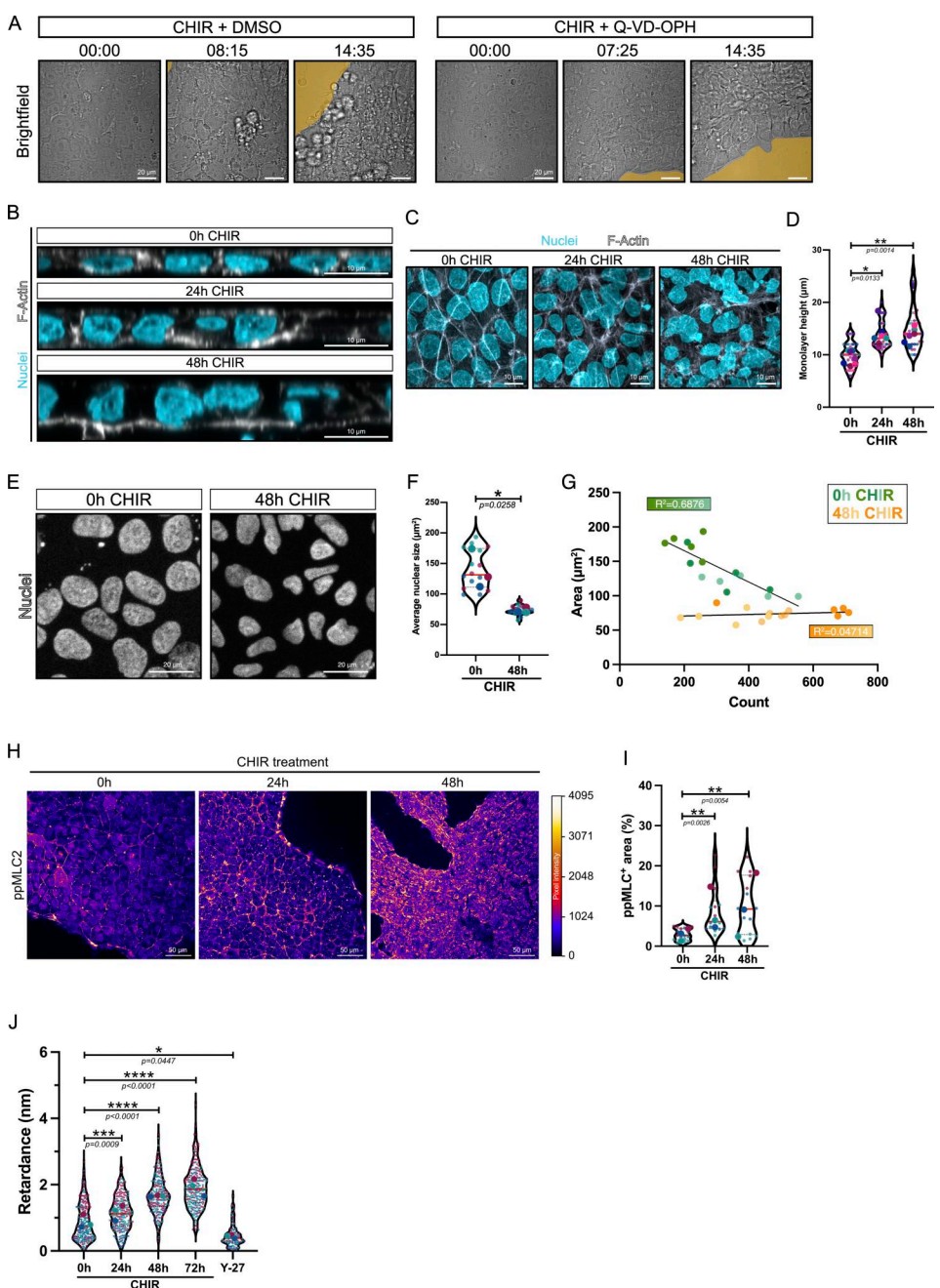

Figure S3.  **hiPSC differentiation along the cardiac mesoderm increases actomyosin-driven contractility. (A)** Bright-field imaging of WT hiPSCs treated with CHIR in the presence or absence of a pan-caspase inhibitor Q-VD-OPH. The retraction area is highlighted in yellow. Time is presented as hh:mm. Scale bar = 20 μm (related to Video 1). **(B and C)** Confocal XZ orthogonal views (B) and MaxIPs (C) of hiPSCs at the basal state (0-h CHIR) or after CHIR treatment (24- and 48-h CHIR), stained for DNA (cyan) and F-actin (gray). Scale bar = 10 μm. **(D)** Quantification of the monolayer height at 0 h, 24, and 48 h after CHIR treatment from B. Violin plots represent individual measurements (small dots) averaged for each biological repeat (large dots). Median (plain red line) and quartiles (dotted black lines) are displayed. $n$ = 32 technical repeats across $N$ = 7 independent biological repeats. A Kruskal–Wallis test with Dunn's multiple comparisons posttest was performed on biological repeats. **(E–G)** Representative MaxIP immunofluorescence of WT hiPSCs during CHIR treatment. Fixed cells were stained for DNA marker (Hoechst). Scale bar = 20 μm (E). Average nuclear size from $n$ = 15 fields of view across $N$ = 3 independent biological repeats. Median (plain red line) and quartiles (dotted black lines) are displayed. A two-tailed unpaired $t$ test was performed (F). Correlation of nuclear area over cell number for each field of view during CHIR treatment. Biological repeats are color-coded. Pearson's correlation coefficient is displayed for each treatment (G). **(H)** Representative MaxIP immunofluorescence of hiPSCs at the basal state (0 h) or after CHIR treatment (24 and 48 h) stained for ppMLC2. Pixels are color-coded by intensity using the fire LUT. Scale bar = 50 μm. **(I)** Percentage of ppMLC2-positive area at 0, 24, and 48 h after CHIR is reported as violin plots. Median (plain red line) and quartiles (dotted black lines) are displayed. $n$ = 15 technical repeats across $N$ = 3 independent biological repeats. A Kruskal–Wallis test with Dunn's multiple comparisons posttest was performed on the technical repeats. **(J)** Retardance measurements obtained from hiPSCs at the basal state (0 h), after CHIR treatment (24, 48, 72 h), or treated with 10 μM Y-27632 at 0 h (Y-27) using QPOL imaging. Median (plain red line) and quartiles (dotted black lines) are displayed. $n$ = 240 technical repeats for 0, 24, 48, and 72 h and $n$ = 120 technical repeats for Y-27 across $N$ = 3 independent biological repeats. A one-way ANOVA with Šidák's multiple comparisons posttest was performed on the biological repeats. ****$P$ < 0.0001, ***$P$ < 0.001, **$P$ < 0.01, *$P$ < 0.05.

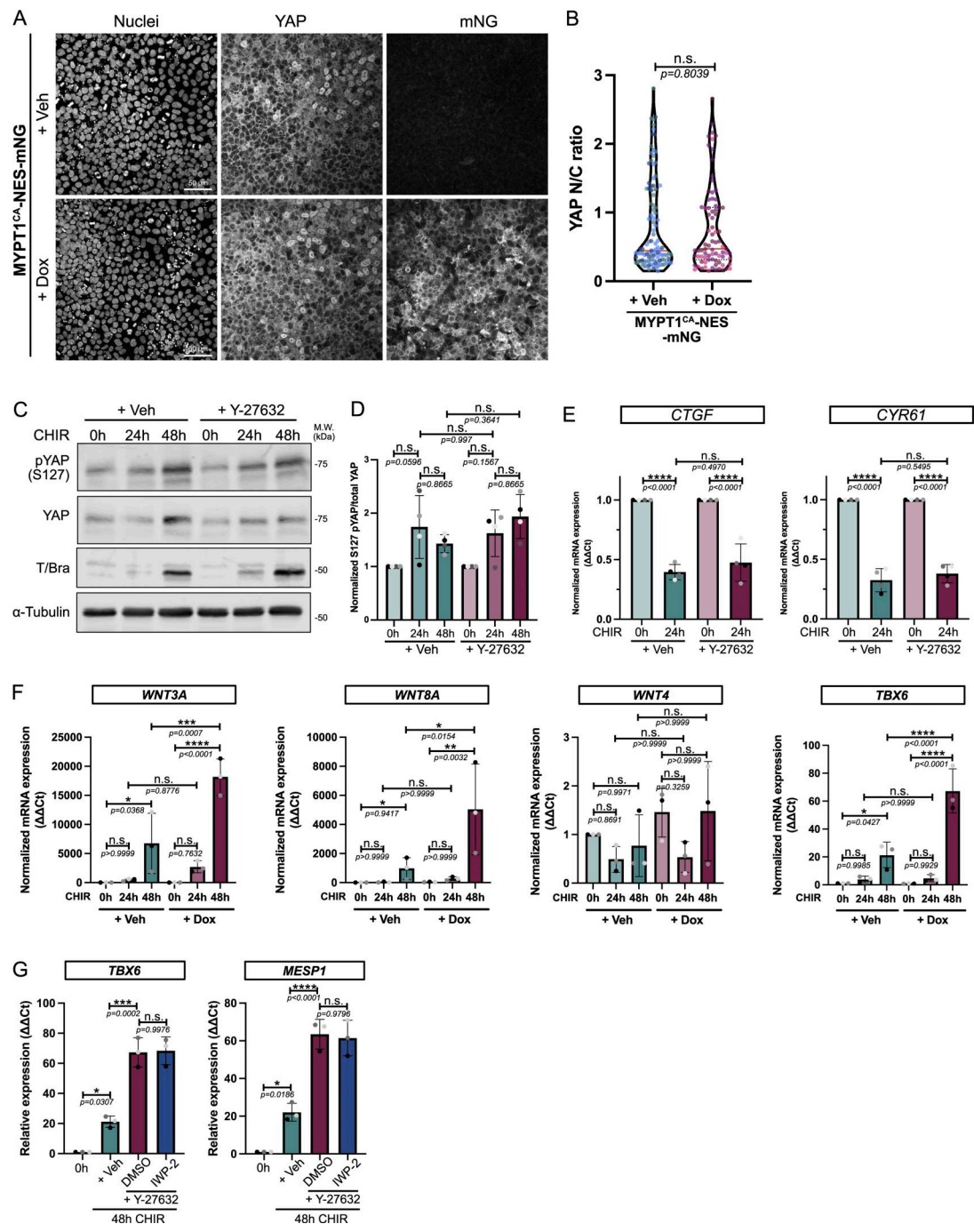

Figure S4. **Key mechanosensitive pathways are not involved in mesoderm specification enhancement following decreased contractility. (A and B)** Immunofluorescence for YAP in MYPT1^CA-NES-mNG cells treated with Dox to promote cell relaxation. Cells were stained for nuclei and YAP. Scale bar = 50 μm (A). N/C YAP ratio was quantified from n = 106 cells (Veh) and n = 80 cells (Dox) across N = 3 independent biological repeats. Mean (red plain line), and first and fourth quartiles (black dotted lines) are displayed on the violin plot. A Mann–Whitney test was performed (B). **(C and D)** Immunoblot for pYAP and total YAP following differentiation (0- to 48-h CHIR) in the presence or absence of 10 μM of ROCK inhibitor Y-27632. Brachyury expression was also probed as an internal positive control, and α-tubulin was used as a loading control. M.W. are displayed on the right side (C). Ratio of phospho/total YAP was quantified across N = 4 independent biological repeats. Mean and SD are displayed. One-way ANOVA with Šidák's multiple comparisons posttest was performed (D). **(E)** Relative expression of YAP target genes *CTGF* and *CYR61* during differentiation ±10 μM Y-27632. N = 4 independent biological repeats. Mean and SD are displayed. One-way ANOVA with Šidák's multiple comparisons posttest was performed. **(F)** Relative expression of canonical WNT (*WNT3A, WNT8A*), noncanonical WNT (*WNT4*), and mesoderm marker (*TBX6*) in MYPT1^CA-NES-mNG cells treated with CHIR (0–48 h) in the presence of Veh or Dox. N = 3 independent biological repeats. Mean and SD are displayed. One-way ANOVA with Šidák's multiple comparisons posttest was performed. **(G)** Relative expression of *TBX6* and *MESP1* (mesoderm markers) following CHIR treatment ±10 μM Y-27632 complemented or not with 7.5 μM porcupine inhibitor (IWP-2 or DMSO, respectively). N = 3 independent biological repeats. Mean and SD are displayed. One-way ANOVA with Tukey's multiple comparisons posttest was performed. M.W., molecular weights; N/C, nuclear-to-cytosolic; Veh, vehicle; pYAP, phospho-S127 YAP. ****P < 0.0001, ***P < 0.001, **P < 0.01, *P < 0.05. Source data are available for this figure: SourceDataFS4.

Fort et al.
Cell contractility suppresses mesoderm induction

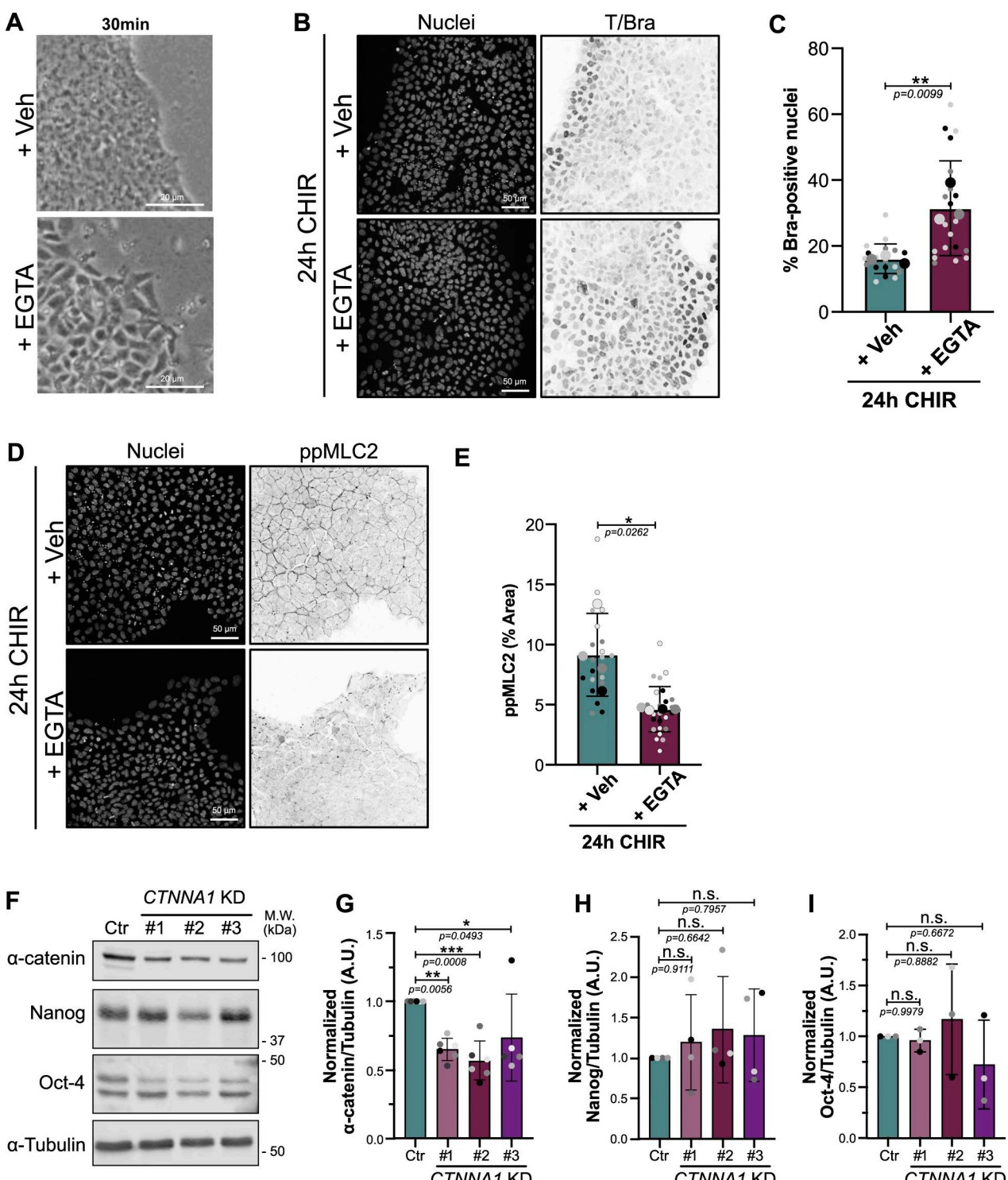

Figure S5. **EGTA treatment and validation *CTNNA1* KD hiPSCs (related to Fig. 4). (A)** Bright-field picture showing the colony morphology following EGTA pretreatment. Colony appears less compact, and individual cell boundaries can be distinguished. Scale bar = 20 µm. **(B and C)** Representative immunofluorescence of hiPSCs treated as shown in Fig. 4 A and stained for Brachyury. Scale bar = 50 µm (B). Quantification of Brachyury-positive nuclei is shown following EGTA treatment. Mean and SD are displayed. *n* = 18 (Veh) and *n* = 20 fields of view across *N* = 3 independent biological repeats. A two-tailed unpaired *t* test was performed on biological repeats (C). **(D and E)** Representative immunofluorescence of hiPSC treated as shown in Fig. 4 A and stained for ppMLC2. Scale bar = 50 µm (D). Quantification of ppMLC2-positive area is shown following EGTA treatment. Mean and SD are displayed. *n* = 20 (Veh) and *n* = 25 fields of view across *N* = 4 independent biological repeats. A two-tailed unpaired *t* test was performed on biological repeats (E). **(F–I)** Representative immunoblot of control (Ctr) or *CTNNA1* KD hiPSCs, probed for α-catenin, Nanog and Oct-4 (pluripotency), and α-tubulin as loading control. M.W. are displayed on the right side (F). Quantification of α-catenin (G), Nanog (H), and Oct-4 (I) expression was obtained by densitometry across *N* = 5–6 (α-catenin), *N* = 4 (Nanog), and *N* = 3 (Oct-4) independent biological repeats. Mean and SD are displayed. One-way ANOVA with Dunnett's multiple comparisons posttest was performed. M.W., molecular weights; Veh, vehicle. ***P < 0.001, **P < 0.01, *P < 0.05. Source data are available for this figure: SourceDataFS5.

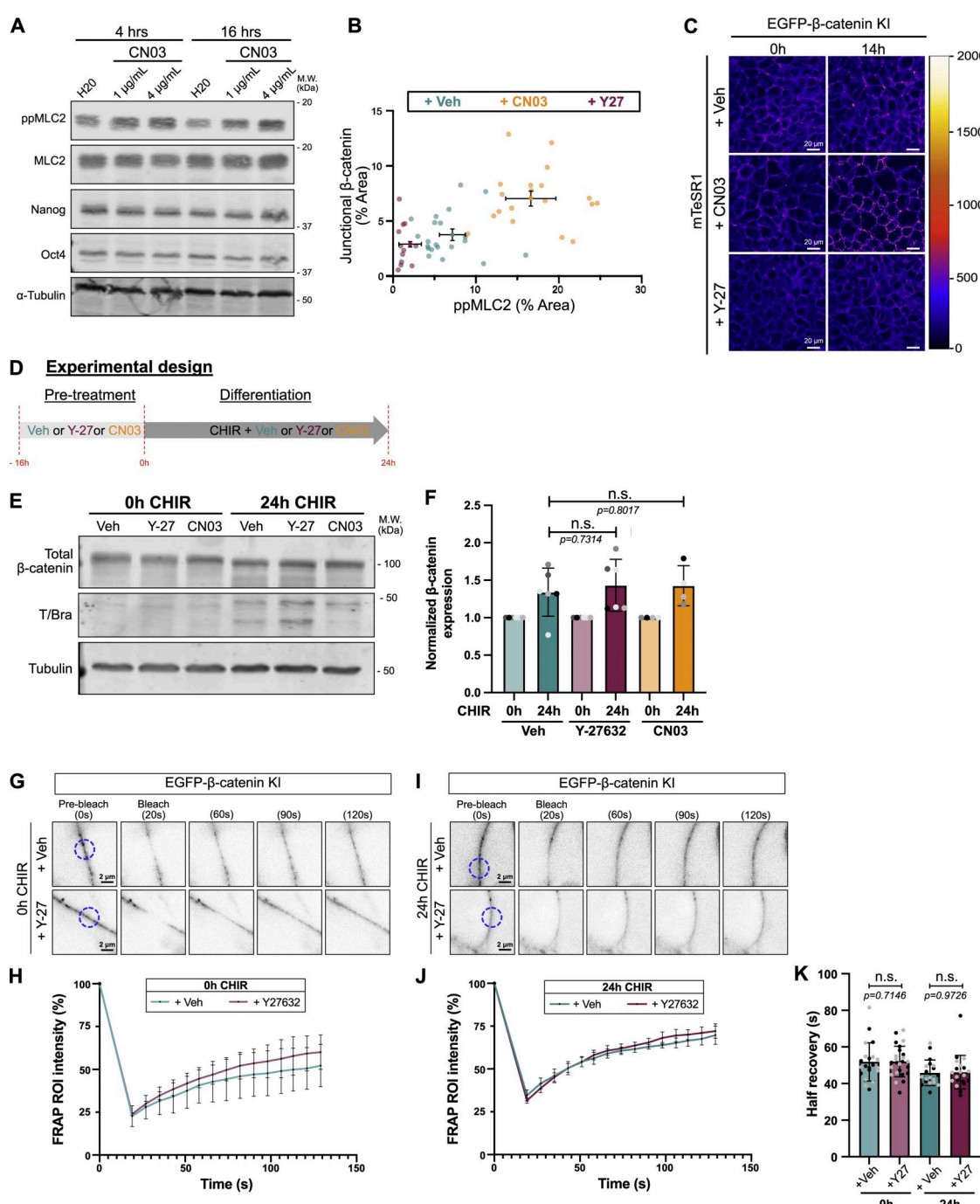

Figure S6.  **Contractility affects β-catenin localization at** AJs **(related to Fig. 5). (A)** Immunoblot assessing the effects of CN03 treatment at 4 and 16 h. Membrane was probed for total and phospho-MLC2 (MLC2 and ppMLC2, respectively), Nanog and Oct-4, and α-tubulin as a loading control. M.W. are displayed on the right side. **(B)** Independent technical repeats from Fig. 5, A and B. Repeats across the 3 biological repeats were averaged and shown as a dot with error bars (SD). **(C)** Still picture from overnight confocal imaging of mEGFP-β-catenin knock-in hiPSCs treated with Veh, 4 μg/ml CN03, and 10 μM Y-27632 at the basal state. Pixels are color-coded by intensity using the Red Fire LUT. Scale bar = 20 μm (related to Video 2). **(D)** Experimental design. Cells were pretreated overnight with Veh or 10 μM Y-27632 or 4 μg/ml CN03 in mTeSR1 and differentiated using CHIR media complemented with Veh or 10 μM Y-27632 or 4 μg/ml CN03 for 24 h before collecting protein lysates. **(E and F)** Representative immunoblot for total β-catenin. The membrane was also probed for Brachyury to show effect of each drug and α-tubulin as a loading control. M.W. are displayed on the right side (E). β-Catenin expression was quantified across $N$ = 6 (Veh and Y-27632) and $N$ = 4 (CN03) independent biological repeats. Mean and SD are displayed. One-way ANOVA with Šidák's multiple comparisons posttest was performed (F). **(G–K)** FRAP experiment was performed on EGFP-β-catenin knock-in hiPSCs pretreated with Veh or 10 μM Y-27632 for 16 h in mTeSR1 (0-h CHIR) or pretreated and differentiated with CHIR supplemented with Veh or 10 μM Y-27632 for 24 h (24-h CHIR). Representative still images at 0 h (G) and 24 h after CHIR (I), before and after bleaching. FRAP area (ROI) is marked as a dotted circle. Scale bar = 2 μm. Fluorescence within the ROI was recorded prior and following bleaching at 0 h (H) and 24 h (J) after CHIR. Half-recovery time was calculated and reported as column graphs. Mean and SD are displayed. A Mann–Whitney test was performed (K). $n$ = 19–25 cells across $N$ = 2 independent biological repeats. M.W., molecular weights; Veh, vehicle. Source data are available for this figure: SourceDataFS6.

Figure S7. **Reduced cell contractility promotes nuclear accumulation of β-catenin (related to Fig. 6). (A)** Experimental design. **(B)** Schematic representation of the quantification pipeline for nuclear β-catenin. **(C)** Quantification of active non–phospho-S45 β-catenin mean intensity in MYPT1$^{CA}$ cells as depicted in C. Veh and Dox were added 16 h prior to CHIR, and cells were fixed before (0 h) or after 24-h CHIR treatment supplemented with Veh or Dox. $n$ = 10–12 technical repeats across $N$ = 3 independent biological repeats. Mean and SD are displayed. One-way ANOVA with Šidák's multiple comparisons posttest was performed. **(D)** Expression of TCF genes (tpm) from parental WTC hiPSC obtained from the Allen Institute transcriptomic data. $N$ = 6 independent biological repeats. **(E)** Positive control for the CUT&RUN assay using IgG and β-catenin binding to WNT target genes (*AXIN2* and *SP5*) between 0 and 24 h of CHIR treatment. Mean and SD are displayed. $N$ = 3 independent biological repeats. One-way ANOVA with Dunnett's multiple comparisons posttest was performed. **(F)** Relative expression of WNT target genes (*LEF1* and *AXIN2*) with increasing concentrations of CHIR in the presence (magenta) or absence (cyan) of 10 µM of ROCK inhibitor Y-27632. $N$ = 7 independent biological repeats. Mean and SEM are displayed. One-way ANOVA with Šidák's multiple comparisons posttest was performed. Veh, vehicle; tpm, transcripts per million.

Video 1. **Bright-field movie of hiPSC colony treated with CHIR supplemented with** Veh **(DMSO) or Q-VD-OPH (cell death inhibitor).** Time in hh:mm. Scale bar = 20 µm. Veh, vehicle.

Video 2.   **Confocal imaging of mEGFP-β-catenin knock-in hiPSCs treated with** Veh **(water), 4 µg/ml CN03, or 10 µM Y-27632 in mTeSR1.** Red Fire LUT reflects the pixel intensity. Time in hh:mm:ss. Scale bar = 20 µm. Veh, vehicle.

Video 3.   **Confocal imaging of mEGFP-β-catenin knock-in hiPSCs treated with CHIR supplemented with** Veh **(water) or 10 µM Y-27632.** Time in hh:mm. Scale bar = 10 µm. Veh, vehicle.

**Provided online are Table S1, Table S2, and Table S3. Table S1 lists common lab reagents. Table S2 lists all antibodies and dyes used in this study, with their respective application and dilution factor. Table S3 lists all qPCR primers, CUT&RUN assay primers, and CRISPR guide oligos.**

