## [Peer Review File · The Journal of Cell Biology]

Actomyosin Contractility is a Potent Suppressor of Mesoderm induction by Human Pluripotent Stem Cells

Loic Fort, Vaishna Vamadevan, Wenjun Wang, and Ian Macara

Corresponding Author(s): Loic Fort, The University of Texas at Arlington

Review Timeline:

Submission Date:	2025-07-15
Editorial Decision:	2025-08-29
Revision Received:	2026-01-07
Editorial Decision:	2026-02-18
Revision Received:	2026-02-23

Monitoring Editor: Elaine Fuchs

Scientific Editor: Gabriele Stephan

Transaction Report:

DOI: <https://doi.org/10.1083/jcb.202507103>

August 29, 2025

Re: JCB manuscript #202507103

Loic Fort
Vanderbilt University

Dear Dr. Fort,

Thank you for submitting your manuscript entitled "Actomyosin Contractility is a Potent Suppressor of Mesoderm Induction by Human Pluripotent Stem Cells". The manuscript was assessed by expert reviewers, whose comments are appended to this letter. We invite you to submit a revision if you can address the reviewers' key concerns, as outlined here.

You will see that reviewers #1 and #2 are supportive thinking your paper provides interesting findings regarding mesoderm specification. We think their proposed alterations and mechanistic extensions, such as those related to mesoderm specification in hESCs under mechanical stress or BMP-4 treatment, as suggested by reviewer #2, would strengthen your paper and should all be addressed in a revised manuscript. Reviewer #3 voices more substantial critiques and we suggest addressing their concerns experimentally, if feasible. Alternatively, we recommend being clear about the limitations of the study in the discussion section of your paper.

GENERAL GUIDELINES:

Text limits: Character count for an Article is < 40,000, not including spaces. Count includes title page, abstract, introduction, results, discussion, and acknowledgments. Count does not include materials and methods, figure legends, references, tables, or supplemental legends.

Figures: Articles may have up to 10 main text figures. Figures must be prepared according to the policies outlined in our Instructions to Authors, under Data Presentation, <https://jcb.rupress.org/site/misc/ifora.xhtml>. All figures in accepted manuscripts will be screened prior to publication.

Supplemental information: There are strict limits on the allowable amount of supplemental data. Articles may have up to 5 supplemental figures. Up to 10 supplemental videos or flash animations are allowed. A summary of all supplemental material should appear at the end of the Materials and methods section.

Please note that JCB now requires authors to submit Source Data used to generate figures containing gels and Western blots with all revised manuscripts. This Source Data consists of fully uncropped and unprocessed images for each gel/blot displayed in the main and supplemental figures. For assays performed using capillary electrophoresis and/or immunoassay-based detection, authors should instead provide the electropherogram graph(s) for each experiment, plotting fluorescence/chemiluminescence intensity vs. molecular weight/size. Please be sure to provide one Source Data file for each figure gels, blots, and/or capillary electrophoresis assays along with your revised manuscript files. File names for Source Data figures should be alphanumeric without any spaces or special characters (i.e., SourceDataF#, where F# refers to the associated main figure number or SourceDataFS# for those associated with Supplementary figures). For traditional gels and blots, the lanes of the gels/blots should be labeled as they are in the associated figure, the place where cropping was applied should be marked (with a box), and molecular weight/size standards should be labeled wherever possible. For capillary electrophoresis assays, each trace in the graph should be color-coded and labeled to indicate which protein, gene, or sample is being measured (please try to avoid red/green combinations to accommodate our color-blind readers).

The typical timeframe for revisions is three to four months. If you anticipate any difficulties in meeting this aforementioned

revision time limit, please contact us and we can work with you to find an appropriate time frame for resubmission. Please note that papers are generally considered through only one revision cycle, so any revised manuscript will likely be either accepted or rejected.

Thank you for this interesting contribution to Journal of Cell Biology. You can contact us at the journal office with any questions at cellbio@rockefeller.edu.

Sincerely,

Elaine Fuchs, Ph.D.
Editor
The Journal of Cell Biology

Gabriele Stephan, Ph.D.
Scientific Editor
Journal of Cell Biology

Reviewer #1:

Review comments:

In this manuscript, Fort et al. investigated the role of actomyosin contractility in mesoderm specification of human pluripotent stem cells. Interestingly, the authors show that genetic or pharmacological increase in contractility suppresses mesoderm induction, while decrease in contractility promotes mesoderm induction. Furthermore, the authors demonstrate that changes in actomyosin contractility prior to WNT activation are important for the contractility-dependent response in mesoderm induction. As a molecular mechanism, they propose that the accumulation of β -catenin at adherens junctions serves as a pool that regulates the levels of active β -catenin in the cytoplasm and nucleus. While this idea is not new to this paper, it may be the first (to my knowledge) to explore this in a stem cell model of early development. Overall, this study nicely bridges the fields of epithelial cell biology, cell mechanobiology and stem cell biology in the mechanobiology of early development and fate development. This work should be of interest to the readership of Journal of Cell Biology and be a solid contribution to our understanding. I do have comments to be addressed before further consideration.

Major comments:

1. While the authors nicely show that exogenous perturbation to cellular contractility impacts mesoderm fate, it is less clear the origins of CHIR-induced changes to cell contractility. Can the authors discuss the mechanism of how CHIR impacts myosin activity and cell dynamics?
2. The experiments to explore the coordination of changes in cell contractility and cell fate priming (e.g. Fig. 3Q-S, lines 210) are really interesting, but only to limited extent. Further development of these might help resolve some of the discrepancies with prior work and broaden the impact of this work to understand how evolving cell mechanical state (in space and time) impacts its transcriptional state. The stem cell model provides an exceptional opportunity to explore this beyond prior studies in developmental models. However, the quantitative analysis of this (both from the RNA profile state analysis and cell dynamics) in this current study is limited. For instance, in the abstract the authors state that "reduced tension primes the pluripotent state". What is precisely meant by "priming" - is there a change in cell signaling or transcriptional identity of these? This should be more precisely characterized, given how central it is to the conclusions of this manuscript.
3. The notion of adherens junctions serving as "dynamic reservoirs" of beta-catenin signaling has been a leading model of fate patterning. The stem cell models provide an opportunity to explore this mechanistically, but I question how much new understanding the authors have contributed beyond previous work. Can the authors please clarify this further?
4. I am also confused how to consider the effects of CHIR on conclusions, rather than the endogenous mechanobiology of Wnt signaling, especially since the authors conclude that the CHIR-driven differentiation is sensitive to mechanobiology in the primed state rather than the induced state. Communicating how their conclusions may be impacted by this differentiation protocol, and how feasible it is to learn about Wnt mechanobiology while also impacting it directly, is important to correct.
5. In Fig. 1 and Fig. S1, the authors co-cultured CA MLCK- or CA ROCK-expressing cells detectable by mVenus with WT cells. However, in many panels, even the "Negative" cells appear to show weak mVenus signals. Could this be due to Dox-independent leaky expression in CA cells, or to heterogeneity in the responses among CA cells? In addition, how did the authors define mVenus-positive and -negative cells?
6. Under the "no dox" condition in Fig. S1F and S1H, mVenus expression is not expected to be induced. I wonder why the

proportion of mVenus-positive cells appears similar to that in the "+dox" condition.

7. In Fig. 2, the authors used a dox-inducible CA MYPT1. The N-terminus 300 amino acids of MYPT1 contain the MLC-binding domain, which is likely to increase the specificity of PP1c for myosin. As this construct appears to be newly designed, the authors should add this information to Fig. 2B with appropriate citations for better understanding.

8. In Fig. 5C, how did the authors quantify the % area? In particular, the β -catenin intensity appears to be similar between the Veh and Y27 conditions, and the junctions seem to be maintained under all conditions.

9. In Fig. 3H, S2M, and S5B, the expression level of T/Bra appears to be higher at the colony edges. Could this be caused by differences in accessibility of CHIR within a colony, as reported for BMP4? If so, what would be the difference between the effects of CHIR and BMP4 on mesoderm induction?

10. In Fig. S4A, the authors estimated the pYAP/total YAP level by Western blotting; however, total YAP expression appears to increase upon CHIR addition. I suggest the authors discuss the involvement of the Hippo pathway based on quantification of nuclear non-phosphorylated YAP levels from immunostaining.

Minor comments:

1. Line 82: Since CalyA is not used throughout this manuscript, I think this explanation may not be necessary.

2. Line 166: Since Y-27632 is not used in Fig. S2L, this sentence should be revised accordingly.

3. In Fig. 3k, the cell sorting images are low resolution and the text is difficult to read. I recommend replacing it with a higher-resolution image.

4. In Fig. 4F, "KO" may be an error and should be "KD".

5. Since Fig. 5E presents images taken after 3 hours of treatment, the authors should remove "or O/N" from Fig. 5D.

6. Line 336: "TCF4L2" should be corrected to "TCF7L2".

7. Could the authors specify the meanings of the abbreviations "NE" and "LSB" in the legend of Fig. 6F?

Reviewer #2:

Summary:

The manuscript from Fort et al investigates how actomyosin contractility regulates WNT-mediated mesoderm induction in human pluripotent stem cells. Using both pharmacological and genetic tools to enhance or suppress actomyosin contractility in hiPSCs undergoing epithelial-mesenchymal transition (EMT) induced by Chiron, they make the unexpected finding that enhancing contractility blocks mesoderm induction. Conversely, suppressing contractility accelerates EMT and mesodermal differentiation. These results were initially surprising to the authors, since Chiron treated hiPSC clusters contract at the edges, which would suggest increased (and not decreased) contractility is part of the differentiation response. However, they conclude that actomyosin contractility must be reduced prior to mesoderm induction through Chiron to prime the pluripotent state for mesoderm commitment. After ruling out alternative explanations as the Hippo pathway and WNT ligand secretion, the authors propose a model in which actomyosin contractility controls the localization of β -catenin. They propose a β -catenin titration model, where actomyosin-mediated junctional binding or release of β -catenin regulates its availability for nuclear translocation and WNT target gene activation.

Overall, the manuscript provides some interesting insights on how contractility impacts fate decisions in pluripotent cells, and is a good fit for the readership of JCB. This is a hot topic in developmental and stem cell biology at the moment, and I think the findings here will be a nice contribution to the ongoing debate of how mechanical aspects shape cell fate and behavior during gastrulation. However, especially because their observations are somewhat divergent from prior reports similar systems, there are several significant points (see especially point 1 below) that should be addressed to better place their key findings within the context of what has already been published.

Major points:

1- The results presented in this paper appear to contradict previous findings reported in the studies of Muncie et al. (2020) and Przybyla et al. (2016), which demonstrated mesoderm specification in hESCs under mechanical tension and BMP-4 treatment. Given this discrepancy, it is critical for the authors to try to recapitulate or better reconcile these earlier findings within their own system. Three specific avenues/variables to consider come to mind that should be addressed:

a) hiPSCs versus hESCs: hiPSCs were used in most experiments (a notable exception being the FACS data in Fig 3k). hESCs as were used in the prior studies from the prior studies. Since hESCs are generally thought to be more naive or less primed than hiPSCs - could this perhaps explain some of the contradiction with Muncie et al? The discussion on this point should be expanded.

b) BMP-4: although the caveats of using BMP-4 to induce mesoderm differentiation are discussed, it would nevertheless be of interest to directly test the impact of BMP-4 treatment together with contractility manipulations in their hPSC system.

c) Matrigel/substrate: the role of Matrigel is briefly alluded to and how it may play a role in the differences observed. The authors claim in the discussion that their system enables them to focus purely on intrinsic contractility, and not a variety of other things

that were examined in the Muncie paper (ie. substrate mechanics). But if the substrate is different here in the present study, this suggests otherwise. Could the be fundamentally different if hiPSCs were cultured on a different substrate/without matrigel? The authors should directly test whether the substrate, particularly the use of Matrigel, is driving the divergent outcomes.

2- The rationale behind using co-cultures in Fig. 1 should be explicitly stated, especially since the results in Fig. 2 are derived from monocultures. This inconsistency raises questions about the impact of cellular interactions on lineage specification. To address this, the authors should consider presenting Fig. 1 using monocultures as well or alternatively provide a clear explanation for the use of co-cultures in Fig. 1. Without this clarification, the co-culture approach may raise unnecessary follow-up questions about the influence of co-culture conditions on the observed outcomes.

3- The paper would benefit from better controls to assess the impact of MYPT1CA on contractility. Specifically, the authors should include analyses of F-actin organization or cell shape, similar to those used in Supplemental Fig. 1, to validate contractility enhancing manipulations. This would help confirm the effectiveness of MYPT1CA in reducing contractility.

4- There are inconsistencies in the timing of experiments that need to be addressed, particularly given the findings in Fig. 3r. For example, why is Fig. 3 based on a 48 h Chiron treatment, while other experiments use 72 h? Similarly, why aren't the cells in Fig. 2 cultured for 72 h to allow for better comparison with Fig. 1? These discrepancies make it difficult to interpret the temporal dynamics of the observed effects. In Fig. 3r, the persistent effect after 24 h of co-treatment with Chiron and Y-27632 (0 h condition) suggests that reducing contractility might be accelerating, rather than simply priming, lineage specification. To clarify this, the authors should include data at 48 h and 72 h following Chiron and contractility inhibition.

5- Experiments to further support the notion that localization of beta-catenin at the junction is controlled by contractility would significantly strengthen the manuscript. One idea would be to FRAP junctional beta-catenin with Chiron under high and low contractility. This seems feasible given the EGFP-beta-catenin cell line already exists (and the data in Figure 5 is a highlight).

6- Drug treatments: more information should be provided with respect to the concentrations that were used for all drug treatments (CN03, ML7, Y27632, H1152). Although the precise concentration is given for some drugs, it is completely missing for a few (Y-26632 for example). Maybe the selected concentrations used are standard in the field, but given this information is not provided, and moreover, that CN03 treatment is lethal after 2 days begs the question of why a lower concentration wasn't used. Dose-response curves could yield interesting insights, but I understand doing this for all datasets is beyond the scope of what is reasonable here.

7- The discussion would benefit from a stronger conclusion and broader outlook. A key unanswered question is whether the observed relaxation before differentiation reflects what happens in vivo, especially since many studies suggest increased tension drives morphogenetic events like primitive streak formation. Additionally, given the early mention of the cardiomyocyte lineage, it would be useful to discuss whether certain lineages, like cardiomyocytes, are more influenced by actomyosin dynamics than others. Expanding on these points would help place the findings in a broader developmental context.

Minor points:

- There is no label for the time of imaging/time after adding Chiron in Fig. 1 and 2d (Fig. 2j has one for example). I think it could be helpful to add one, this goes along with the comments about timing of experiments.
- The data that deals with apoptosis is a bit confusing and distracts from the major findings of the paper. Since these are largely negative results, they should be shown only in supplemental figures and discussed together in one place.
- Why did the authors choose to switch from pure IF analysis in Fig. 1 to WB analysis in Fig. 2? Why is there no EOMES quantification in Fig. 2? A short explanation would be helpful.
- It is impossible to see ZO-1 in the images in Fig. 2d and 2j. Better quality images should be provided.
- It is in general a bit difficult for the reader to follow why the authors switch between IF, WB and then also qPCR analysis in Fig. 3d-g. Maybe this could be briefly clarified.
- TCF7L2: there is a typo on line 336 (TCF4L2 is written instead)

Reviewer #3:

This manuscript presents a study on hPSCs and the potential cross-talk between myosin-dependent cell contractility and induction of mesoderm fate. It is known that hPSC colonies can produce mesoderm upon activation of the Wnt pathway. The

authors made here the observation that when the hPSCs was treated with a RhoGTPase activator, Wnt-dependent induction of TBXT was drastically inhibited. They validated this result through expression of myosin activators caRock and caMLCK, and, reciprocally, they showed that myosin inhibition significantly speeds up mesoderm induction (and EMT, using Slug as marker). The use of Rock chemical inhibitor allowed to test temporality, revealing that contractility plays an antagonistic role at early stages of induction (as opposed to the later effect of Wnt signalling that increases contractility). The author further looked into the role of cadherin adhesion, and showed that reducing cell adhesion also increased mesoderm induction, probably by loosening mechanical coupling and thus global contractility. Finally, they found that experimental cell relaxation led to a decrease in junctional β -catenin and a concomitant increase in nuclear β -catenin, in β -catenin interaction with a member of the TCF family, and in occupancy of the TBXT promoter. The authors provide further validating data, which I will not summarize here. The initial observation of the dramatic block of TBXT expression upon Rock inhibition is indeed spectacular, and definitely deserved to build a project to understand this effect. The study shows clean, convincing data supporting a model where contractility is antagonist to mesoderm induction, and where contractility and β -catenin, thus presumably Wnt signalling, influence each other.

This is altogether a nice piece of work on a nifty system where the contractility- β -catenin-Wnt system yield a clear-cut output in terms of cell fate.

However, I am not quite as enthusiastic about the originality and general impact of these findings. I explain here below why the major findings are not quite up to what one would expect in JCB:

1) Adhesion, tension and Wnt signalling: There have been over more than two decades multiple observations indicating that β -catenin can be released from junctions to activate signalling, or reciprocally that sequestration by cadherins buffers β -catenin and augments the threshold required for Wnt signalling. Similarly, the fact that decreasing adhesion impacts on cell contractility and vice-versa is similarly well established. The actual key questions relate to the mechanisms that may couple this crosstalk with regulation of nuclear β -catenin levels/activity. Unfortunately, the data presented here remain "descriptive" and do not provide new clues on this topic.

2) Significance for developmental biology: The other potential interest of this work is the strong permissive role of low contractility on mesoderm fate induction: This would be exciting if it would reflect an actual developmentally relevant process: In such scenario, contractility act as a gatekeeper, and timely cell softening would contribute to mesoderm induction. Does this happen physiologically? Not being an expert in mammalian/human mesoderm induction and gastrulation, I am not quite sure, but as far as I know TBXT induction occurs before EMT, at a stage where cells are still within the epiblast, thus an epithelial monolayer. Is there any evidence that this layer softens BEFORE TBXT induction?

In any case, the cellular system used in this study is highly artificial, and it is impossible to infer a physiological relevance. Let me explain in a few words: hPSC colonies are grown in an environment without any resemblance to a gastrula embryo. Most relevant here is tension, which is likely to be orders of magnitude lower in the embryo. The "standard" hPSC growing conditions are probably then completely off. These "poor cells" are under abnormally high tension and respond by reinforcing adhesions and cytoskeleton coupling, and sequestering more β -catenin. The resulting threshold for Wnt signalling gets artificially high, and induction works (suboptimally) because the pathway is overactivated by high doses of chemical activator (GSK3 inhibitor, to be specific). If this is the case, then Rock inhibition, or inhibition of adhesion, merely restore conditions more in line within the range of tension that cells are supposed to experience during gastrulation, and mesoderm induction is getting more optimal. In summary: The results are potentially exciting if there is indeed a developmental program that involves tension as a true "regulator". Alternatively, the authors may have found a parameter that improves hPSC induction, which is a technical advance in the field of iPSCs. To test the first (and obviously most exciting) possibility, the authors would need to compare their in vitro settings with in vivo conditions (gastruloids could do). One would minimally determine whether the identified mechanical control of induction works within ranges of contractility/softness expected during gastrulation, and similarly compare the levels of Wnt- β -catenin signalling activation in both systems.

Other comment

Fig6A. a) The cell fractionation protocol is unlikely to truly separate "nuclei" from "cytoplasm". The "nuclear" fraction is most certainly a nuclear insoluble fraction (chromatin + lamina), which is completely fine but should be called as such. A true nuclear fraction should be validated using a soluble nucleoplasmic marker.

b) Also missing is a blot for cadherins to determine where is the plasma membrane pool of β -catenin ("cytoplasm", "nuclei"?). Without these ESSENTIAL controls, it is impossible to interpret these experiments.

Manuscript #202507103

Reviewer #1:

Review comments:

In this manuscript, Fort et al. investigated the role of actomyosin contractility in mesoderm specification of human pluripotent stem cells. Interestingly, the authors show that genetic or pharmacological increase in contractility suppresses mesoderm induction, while decrease in contractility promotes mesoderm induction. Furthermore, the authors demonstrate that changes in actomyosin contractility prior to WNT activation are important for the contractility-dependent response in mesoderm induction. As a molecular mechanism, they propose that the accumulation of β -catenin at adherens junctions serves as a pool that regulates the levels of active β -catenin in the cytoplasm and nucleus. While this idea is not new to this paper, it may be the first (to my knowledge) to explore this in a stem cell model of early development.

Overall, this study nicely bridges the fields of epithelial cell biology, cell mechanobiology and stem cell biology in the mechanobiology of early development and fate development. This work should be of interest to the readership of Journal of Cell Biology and be a solid contribution to our understanding. I do have comments to be addressed before further consideration.

We would like to thank the reviewer for their accurate summary and positive feedback on our manuscript.

Major comments:

1. While the authors nicely show that exogenous perturbation to cellular contractility impacts mesoderm fate, it is less clear the origins of CHIR-induced changes to cell contractility. Can the authors discuss the mechanism of how CHIR impacts myosin activity and cell dynamics?

We thank the reviewer for this thoughtful comment. CHIR99021 is a potent and selective GSK3 inhibitor that activates the WNT/ \$\beta\$ -catenin pathway by blocking \$\beta\$ -catenin destruction (as does canonical WNT signaling). Although non-canonical WNT signaling can modulate RhoA/ROCK activity and, therefore, myosin II-mediated contractility, we are not aware of any direct evidence that would link the canonical WNT pathway to contractility. Addition of the canonical ligand WNT3A to murine osteoblasts activated increased RhoA-GTP and ROCK phosphorylation (Shi, Xu et al Cell. Regen. 2021) ¹ but the mechanism is not known. Additionally, GSK3 inhibition has been reported to alter focal adhesion dynamics (Dobson et al, Cell Report 2023) ², which might secondarily impact actomyosin tension.

Importantly, we see no increase in nuclear \$\beta\$ -catenin when actomyosin contractility is reduced 16 hrs prior to CHIR addition (Figure 6A and Supplementary Figure 7C), probably because the destruction complex removes any free \$\beta\$ -catenin.

We now mention this point in the revised manuscript [lines 323-325 and 404-411]. We have also emphasized a citation to a BioRxiv preprint from the Allen Institute that reports a similar early contraction phase prior to EMT (Hookway et al, BioRxiv 2024) ³.

2. The experiments to explore the coordination of changes in cell contractility and cell fate priming (e.g. Fig. 3Q-S, lines 210) are really interesting, but only to limited extent. Further development of these

might help resolve some of the discrepancies with prior work and broaden the impact of this work to understand how evolving cell mechanical state (in space and time) impacts its transcriptional state. The stem cell model provides an exceptional opportunity to explore this beyond prior studies in developmental models. However, the quantitative analysis of this (both from the RNA profile state analysis and cell dynamics) in this current study is limited. For instance, in the abstract the authors state that "reduced tension primes the pluripotent state". What is precisely meant by "priming" - is there a change in cell signaling or transcriptional identity of these? This should be more precisely characterized, given how central it is to the conclusions of this manuscript.

We agree that the term "*priming*" was misleading in this context, since in the stem cell field it specifically refers to pluripotent state transitions (naïve vs. primed). To avoid confusion, we now use the term "*sensitization*" to describe the effect of reduced tension on subsequent mesodermal differentiation.

As to the underlying mechanism of this sensitization, our manuscript discussed the effects of cell contractility on cis-regulatory elements (line 429-431).

3. The notion of adherens junctions serving as "dynamic reservoirs" of beta-catenin signaling has been a leading model of fate patterning. The stem cell models provide an opportunity to explore this mechanistically, but I question how much new understanding the authors have contributed beyond previous work. Can the authors please clarify this further?

We agree with the reviewer that the relationship between adherens junction dynamics and WNT signaling has been widely explored (Heasman et al, Cell 1994, Fagotto et al, J. Cell Biol 1996, Sanson et al, Nature 1996, Gottardi and Gumbiner, Curr. Biol 2001) ⁴⁻⁷ but the extent to which WNT signaling is controlled physiologically by b-catenin exchange at AJs is still unclear (see recent review from Wal and Amerongen 2020) ⁸. Moreover, as we point out in the manuscript, there are several studies that propose

the opposite, that differentiation to mesoderm involves increased β -catenin at AJs and less in the nucleus (Muncie et al, Przybyla et al) ^{9,10}.

Our work advances this field in several ways:

- a) To our knowledge, this is the first study to directly investigate how actomyosin contractility regulates mesoderm commitment in human pluripotent stem cells, independent of changes to ECM composition, substrate stiffness, cell density, or exogenous physical perturbations (e.g., compression or stretch).
- b) We identify a previously unreported *sensitization* mechanism, whereby changes in contractility in pluripotent cells before differentiation influence subsequent lineage commitment.
- c) We provide evidence for a force-dependent titration of available β -catenin in mammalian cells. The only other study to our knowledge that examined WNT signaling in response to direct manipulation of actomyosin contractility was in *D. melanogaster* imaginal disks (Hall et al, MBoC 2018) ¹¹.
- d) We demonstrate that elevated contractility inhibits mesoderm commitment in human stem cells, which contrasts with prior studies in other systems (e.g., Muncie et al., Przybyla et al., Pukhlyakova et al., Brunet et al.) ^{9,10,12,13} that reported high tension promotes mesoderm specification (using BMP4, which does not directly induce mesoderm).

We have expanded the discussion to more clearly highlight our contributions.

4. I am also confused how to consider the effects of CHIR on conclusions, rather than the endogenous mechanobiology of Wnt signaling, especially since the authors conclude that the CHIR-driven differentiation is sensitive to mechanobiology in the primed state rather than the induced state. Communicating how their conclusions may be impacted by this differentiation protocol, and how feasible it is to learn about Wnt mechanobiology while also impacting it directly, is important to correct. We appreciate the opportunity to clarify this important point and would like to note that addition of Wnt3A does not induce mesoderm commitment, likely because it cannot access receptors on the basolateral membrane.

Our conclusion is not that contractility replaces WNT activation, but rather that contractility modulates the efficiency and kinetics of the CHIR response. In this sense, contractility acts more like a licensing mechanism, sensitizing cells to respond more robustly and rapidly once WNT signaling is induced.

With respect to the pluripotent state, we showed that: (1) altering contractility does not change pluripotency marker expression (Supplementary Fig. 2N, Q), ensuring that cells remain in a comparable baseline state; and (2) reduced contractility in the pluripotent state sensitizes cells to CHIR, by a mechanism we are still investigating. As shown in the additional data provided (**Figure 1 for reviewers**) our results suggest that chromatin accessibility is influenced by the level of contractility experienced during pluripotency. This raises the possibility that force-dependent enhancer/repressor rearrangements at mesoderm-associated loci underlie the enhanced CHIR responsiveness. We are actively pursuing this hypothesis in ongoing work.

5. In Fig. 1 and Fig. S1, the authors co-cultured CA MLCK- or CA ROCK-expressing cells detectable by

mVenus with WT cells. However, in many panels, even the "Negative" cells appear to show weak mVenus signals. Could this be due to Dox-independent leaky expression in CA cells, or to heterogeneity in the responses among CA cells? In addition, how did the authors define mVenus-positive and -negative cells? The negative cells do not contain the lentivector that expresses mVenus so we conclude that the weak background fluorescence is nonspecific. The "negative" cells are WT stem cells that were co-cultured with mVenus positive cells (that can be induced to express ROCK or MLCK).

The way the cells were plated for this co-culture experiment results in a mix of large (10-15 cells) and smaller (2-5 cells) clusters. The dotted lines added on the representative pictures were only highlighting large clusters to avoid making the image too busy. The quantification provided in Supp. 1F and Supp 1H reports our way of distinguishing mVenus status. We set up a binary threshold for green pixels and measured the normalized area covered by the mask. Clusters with a threshold area <15% were considered as negative. In addition, we have clarified our experimental design in the material and method by adding the following section (lines 596-604):

"Generation of ROCK2^{CA} and MLCK^{CA} cell lines and co-culture setup

ROCK^{CA}, MLCK^{CA} constructs, and the empty vector were obtained from Addgene (#84649, #84647, #25734 respectively) and used to generate lentiviral particles. hiPSCs were transduced and clonal populations were obtained. mVenus acts as a transduction reporter and is constitutively expressed. Only ROCK^{CA} and MLCK^{CA} are inducible. mVenus-positive cells were mixed with WT (mVenus negative WT hiPSC) at a 1:1 ratio. Cells were treated with doxycycline or Vehicle overnight prior to CHIR treatment in the presence or absence of doxycycline. To account for a nonspecific background in the WT population, mVenus-positive cells were defined as cells with a green signal covering >15% of their area."

6. Under the "no dox" condition in Fig. S1F and S1H, mVenus expression is not expected to be induced. I wonder why the proportion of mVenus-positive cells appears similar to that in the "+dox" condition. We are sorry for this confusion. The vectors used for these experiments (Addgene 84649 and 84647) contain a mVenus cassette under a Ubiquitin C promoter, so it is constitutively expressed. Only MLCK and ROCK2 are Dox-inducible, under a separate, Tet-on promoter. We tried to represent this system in Figure 1A by having the cells marked green following transduction and not following Dox addition. We have now explicitly explained this point in both the figure legend and material and methods as followed:

"Note that mVenus expression is constitutive and used as a marker for transduction, while expression of ROCK2/MLCK is Dox-inducible."

7. In Fig. 2, the authors used a dox-inducible CA MYPT1. The N-terminus 300 amino acids of MYPT1 contain the MLC-binding domain, which is likely to increase the specificity of PP1c for myosin. As this construct appears to be newly designed, the authors should add this information to Fig. 2B with appropriate citations for better understanding.

We thank the reviewer for the suggestion. We already cited the primary research paper used to design this construct (Yamamoto et al, Nat. Comm 2021)¹⁴. Our initial figure legend provided a brief description of the truncation, which we have now extended. It now reads as followed:

“Schematic representation of full length human MYPT1 (1030 aa), interacting with PP1c phosphatase and a protein of unknown function M20. Truncated MYPT1 Nterm (1-300) (containing the MLC-binding domain) leads to constitutive recruitment and activation of PP1C. Truncated MYPT1 was fused with NES-mNeonGreen as a marker, and the construct is referred to as MYPT1^{CA}-NES-mNG.”

8. In Fig. 5C, how did the authors quantify the % area? In particular, the β -catenin intensity appears to be similar between the Veh and Y27 conditions, and the junctions seem to be maintained under all conditions.

This question has two components:

Quantification in Figure 5C: The goal of this experiment was to test whether contractility correlates with junctional β -catenin levels. The % area measurement was performed in ImageJ by applying a fixed threshold defined from the control condition (Veh) for each channel (β -catenin and pp-MLC). This threshold was then applied uniformly across all treatment groups to ensure consistency. We apologize for not providing these important technical data in our earlier version, which can now be found in the Materials & Methods section (lines 578-580). While we agree that in fixed immunostaining samples the difference between Veh and Y-27632 is subtle, quantification consistently showed a lower junctional β -catenin signal under Y-27632 treatment. We acknowledge that fixation and antibody staining may introduce variability, so to address this, we complemented these data with live imaging of endogenous β -catenin using our EGFP-CTNNB1 knock-in line (Figure 5E–F and Supplementary Video 2), which confirmed significant differences.

Appearance of junctions: The reviewer is correct that junctions appear intact in Figure 5B. This is because the cells were cultured in mTeSR1 and fixed prior to the onset of differentiation. In contrast, the accelerated disassembly of junctions upon Y-27632 treatment (reported in Figure 3) is only observed under differentiation conditions.

9. In Fig. 3H, S2M, and S5B, the expression level of T/Bra appears to be higher at the colony edges. Could this be caused by differences in accessibility of CHIR within a colony, as reported for BMP4? If so, what would be the difference between the effects of CHIR and BMP4 on mesoderm induction?

We thank the reviewer for this observation. Consistent with prior reports, Brachyury-positive cells preferentially emerge at colony edges, which are characterized by lower mechanical tension (Kim et al., Stem Cell Reports 2021) ¹⁵. We do not believe this pattern reflects differences in drug accessibility: CHIR is a cell-permeable small molecule that readily and uniformly penetrates cell membranes, whereas BMP4 requires receptor engagement on the basolateral surface (Vasić et al., Dev. Cell 2023) ¹⁶, which can lead to edge-specific responses. Thus, the observed edge bias in T/Bra expression under CHIR treatment is more likely linked to mechanical context than to differential ligand accessibility. In addition, we have now performed western blot on BMP4/bFGF-treated cells and reported a similar increase of Brachyury expression in low contractile conditions (**Supplementary Figure 2R**).

10. In Fig. S4A, the authors estimated the pYAP/total YAP level by Western blotting; however, total YAP expression appears to increase upon CHIR addition. I suggest the authors discuss the involvement of the Hippo pathway based on quantification of nuclear non-phosphorylated YAP levels from immunostaining.

We thank the reviewer for their suggestion. Because of the slight increase of total YAP, we felt western blots and qPCR for YAP-target genes were more appropriate and sensitive quantification methods than a N/C ratio. N/C ratio is an easy and visually informative proxy but is limited by its indirect readout and does not guarantee transcription activation, compared to target gene expression.

However, we now provide representative immunofluorescence images for YAP in the MYPT1 lines that were induced (pink) or not (blue) with doxycycline overnight. N/C ratio does not show any difference in YAP localization, suggesting that the Hippo pathway is not differentially regulated by cell contractility in hiPSC, confirming both our western blot and qPCR data. This panel is now part of **Supplementary Figure 4A-B** and mentioned lines 234-237)

Legend:

A) Immunofluorescence of MYPT1-CA-NES-mNG cells treated with Doxycycline to promote cell relaxation. Cells were stained for Nuclei and YAP. Scale bar = 50um

B) Nuclear-to-cytosolic (N/C) YAP ratio was quantified from n=106 cells (Veh) and n=80 cells (Dox) across n=3 independent biological repeats. Mean (red plain line), first and fourth quartiles (black dotted lines) are represented. Mann-Whitney test was performed.

Minor comments:

1. Line 82: Since CalyA is not used throughout this manuscript, I think this explanation may not be necessary. We have removed this sentence
2. Line 166: Since Y-27632 is not used in Fig. S2L, this sentence should be revised accordingly. We revised this sentence, which now reads: Apoptosis was not affected following Y-27632 (Supplementary Figure 2N) or H-1152 treatment (Supplementary Figure 2L)
3. In Fig. 3k, the cell sorting images are low resolution and the text is difficult to read. I recommend replacing it with a higher-resolution image. We apologize for the low resolution of the gate. We have reprocessed each panel to increase font size.
4. In Fig. 4F, "KO" may be an error and should be "KD". We have corrected this error.
5. Since Fig. 5E presents images taken after 3 hours of treatment, the authors should remove "or O/N"

from Fig. 5D. We have corrected this error.

6. Line 336: "TCF4L2" should be corrected to "TCF7L2". Thank you. The typo has been fixed.

7. Could the authors specify the meanings of the abbreviations "NE" and "LSB" in the legend of Fig. 6F? We have added the meaning of NE (Nuclei Extraction buffer) and LSB (Lysate Sample Buffer).

Reviewer #2:

Summary:

The manuscript from Fort et al investigates how actomyosin contractility regulates WNT-mediated mesoderm induction in human pluripotent stem cells. Using both pharmacological and genetic tools to enhance or suppress actomyosin contractility in hiPSCs undergoing epithelial-mesenchymal transition (EMT) induced by Chiron, they make the unexpected finding that enhancing contractility blocks mesoderm induction. Conversely, suppressing contractility accelerates EMT and mesodermal differentiation. These results were initially surprising to the authors, since Chiron treated hiPSC clusters contract at the edges, which would suggest increased (and not decreased) contractility is part of the differentiation response. However, they conclude that actomyosin contractility must be reduced prior to mesoderm induction through Chiron to prime the pluripotent state for mesoderm commitment. After ruling out alternative explanations as the Hippo pathway and WNT ligand secretion, the authors propose a model in which actomyosin contractility controls the localization of β -catenin. They propose a β -catenin titration model, where actomyosin-mediated junctional binding or release of β -catenin regulates its availability for nuclear translocation and WNT target gene activation.

Overall, the manuscript provides some interesting insights on how contractility impacts fate decisions in pluripotent cells, and is a good fit for the readership of JCB. This is a hot topic in developmental and stem cell biology at the moment, and I think the findings here will be a nice contribution to the ongoing debate of how mechanical aspects shape cell fate and behavior during gastrulation. However, especially because their observations are somewhat divergent from prior reports similar systems, there are several significant points (see especially point 1 below) that should be addressed to better place their key findings within the context of what has already been published.

We appreciate the positive feedback from the reviewer.

Major points:

1- The results presented in this paper appear to contradict previous findings reported in the studies of Muncie et al. (2020) and Przybyla et al. (2016), which demonstrated mesoderm specification in hESCs under mechanical tension and BMP-4 treatment. Given this discrepancy, it is critical for the authors to try to recapitulate or better reconcile these earlier findings within their own system. Three specific avenues/variables to consider come to mind that should be addressed:

a) hiPSCs versus hESCs: hiPSCs were used in most experiments (a notable exception being the FACS data in Fig 3k). hESCs as were used in the prior studies from the prior studies. Since hESCs are generally thought to be more naive or less primed than hiPSCs - could this perhaps explain some of the contradiction with Muncie et al? The discussion on this point should be expanded.

We thank the reviewer for this comment. While most of our experiments were performed with hiPSCs, we shared the concern about potential cell line-specific effects and therefore repeated key experiments in the hESC line H9. The results were highly consistent across both cell types (see **Supplementary Fig. 2N–Q**). Moreover, we obtained the exact H9 Brachyury-reporter line previously used by the Weaver lab⁹ and performed unbiased flow cytometry analysis. Strikingly, our results in this line were opposite to those previously reported, despite the use of the same cells (but on stiff/soft matrices, +/- BMP4, rather than +/-CHIR).

Therefore, we do not think the discrepancy of results is based on the cell line used. In addition, by using hiPSCs and hESCs, we ruled out an effect of cell reprogramming on our phenotype.

b) BMP-4: although the caveats of using BMP-4 to induce mesoderm differentiation are discussed, it would nevertheless be of interest to directly test the impact of BMP-4 treatment together with contractility manipulations in their hPSC system.

We thank the reviewer for this helpful suggestion. In response, we have performed BMP4-based differentiation experiments combined with contractility manipulation in our hiPSC system. Initially, we used the BMP4 and bFGF concentrations reported by Muncie et al., but under these conditions we did not observe induction of mesoderm markers. After optimization, we found that increasing bFGF from 10 ng/mL to 100 ng/mL was necessary to achieve robust mesoderm induction in hiPSCs, which may reflect differences in signaling requirements between hiPSCs and hESCs. Under these optimized conditions, we observed a similar enhancement of Brachyury expression in Y-27632-treated samples. A notable difference from the CHIR-based protocol, however, was the timing: in the BMP4-based system, Brachyury expression was not detectable at 24 h following ROCK inhibition. (Now **Supplementary Figure 2R**).

Legend:

Expression of T/Bra was analyzed by western blot following treatment with a combination of 50 ng/mL BMP4 and 100 ng/mL bFGF as described by Muncie et al, in the presence or absence of 10 μ M Y-27632. pERK and total ERK were used as a positive control for FGF pathway activity. Molecular weights are shown on the right hand side

c) Matrigel/substrate: the role of Matrigel is briefly alluded to and how it may play a role in the differences observed. The authors claim in the discussion that their system enables them to focus purely

on intrinsic contractility, and not a variety of other things that were examined in the Muncie paper (ie. substrate mechanics). But if the substrate is different here in the present study, this suggests otherwise. Could the be fundamentally different if hiPSCs were cultured on a different substrate/without matrigel? The authors should directly test whether the substrate, particularly the use of Matrigel, is driving the divergent outcomes.

We thank the reviewer for raising this important point. Matrigel remains the gold standard for both maintenance and differentiation of human pluripotent stem cells, and it is well established that switching between substrates can substantially alter stem cell behavior, often requiring extensive protocol re-optimization. We acknowledge the limitations of Matrigel; however, we would like to note that in both their 2016 *Cell Stem Cell* and 2020 *Developmental Cell* studies, Przybyla et al, and Muncie et al used hydrogels coated with either Matrigel or recombinant basement membrane (which they describe as Matrigel-equivalent). This suggests that the use of Matrigel alone is unlikely to account for the divergent outcomes.

To directly address the reviewer’s concern, we repeated key experiments using vitronectin-XF as an alternative substrate, which shows a similar trend in Brachyury expression for cells plated on Matrigel or Vitronectin, following Y-27632 (**Figure 2 for reviewers**).

We expanded our discussion to reflect these points (Lines 389-400).

Figure 2 for reviewers

2- The rationale behind using co-cultures in Fig. 1 should be explicitly stated, especially since the results in Fig. 2 are derived from monocultures. This inconsistency raises questions about the impact of cellular interactions on lineage specification. To address this, the authors should consider presenting Fig. 1 using monocultures as well or alternatively provide a clear explanation for the use of co-cultures in Fig. 1. Without this clarification, the co-culture approach may raise unnecessary follow-up questions about the influence of co-culture conditions on the observed outcomes.

We apologize for the confusion regarding the co-culture setup. Our rationale was twofold: (1) to provide a genetic means of increasing actomyosin contractility, and (2) to include an internal control in the same environment, thereby ensuring that experimental and control cells were exposed to identical cues, a strategy we believe provides the most rigorous comparison. We now emphasize the later in the result section (line 94-95).

While we did not repeat the ROCK2- or MLCK-overexpression experiments in monoculture, we note that pharmacological induction of high contractility using CN03 in wild-type monolayers produced similar

outcomes (Supplementary Figure 1C-D), supporting the conclusion that elevated contractility alone is sufficient to alter lineage specification.

3- The paper would benefit from better controls to assess the impact of MYPT1CA on contractility. Specifically, the authors should include analyses of F-actin organization or cell shape, similar to those used in Supplemental Fig. 1, to validate contractility enhancing manipulations. This would help confirm the effectiveness of MYPT1CA in reducing contractility.

We thank the reviewer for this helpful suggestion. While F-actin organization can provide qualitative information, it is an indirect readout of contractility and is most informative when interpreted alongside ppMLC2 staining (as shown in Supplemental Fig. 1). For the MYPT1^{CA} system, we have already included both western blot and immunofluorescence analyses of ppMLC2 levels \pm doxycycline, which provide a more direct measure of contractility (Figure 2C-E).

To further strengthen this point, we now include an additional panel (new Figure 2F) showing F-actin organization in MYPT1^{CA} cells as suggested. We also include a side-by-side panel for the reviewer with Y-27632-treated cells as comparison.

Legend:

Immunofluorescence of F-actin organization in MYPT1-cells treated with Dox (A) and WT cells treated with Y-27632 (B). As controls, mNG expression (A) and ppMLC2 staining are shown (B). Scale bar = 50um.

4- There are inconsistencies in the timing of experiments that need to be addressed, particularly given the findings in Fig. 3r. For example, why is Fig. 3 based on a 48 h Chiron treatment, while other experiments use 72 h? Similarly, why aren't the cells in Fig. 2 cultured for 72 h to allow for better comparison with Fig. 1? These discrepancies make it difficult to interpret the temporal dynamics of the observed effects. In Fig. 3r, the persistent effect after 24 h of co-treatment with Chiron and Y-27632 (0 h condition) suggests that reducing contractility might be accelerating, rather than simply priming, lineage specification. To clarify this, the authors should include data at 48 h and 72 h following Chiron and contractility inhibition.

We appreciate the reviewer's concern regarding timing. The duration of each experiment was chosen based on the expected dynamics of mesoderm induction under control conditions.

In **Figure 2** and **Figure 3**, our goal was to test whether reduced contractility increases mesoderm commitment. Since >50% of cells in control conditions are already Brachyury-positive by 48 h, we focused on the 24–48 h window, where differences between treatments can still be clearly distinguished. Beyond 48 h, nearly all cells express Brachyury, making direct comparison between conditions less informative (e.g., Y-27632–treated cells already reach >80% positivity by 48 h).

However, to address the reviewer's point, we now provide an extended timecourse experiment from 0h–72h of differentiation +/- Y-27632 (**Figure 3 for reviewers**). Cells were fixed and stained for Slug and ZO-1, as proxy for EMT. While Slug expression emerges earlier in Y-27632–treated cells, consistent with a potential acceleration effect, we also observe a sustained increase in Slug levels at all time points, including at 72 h. Importantly, this difference persists at a stage when EMT is complete in both conditions, as indicated by the loss of ZO-1–positive tight junctions. These findings are therefore inconsistent with a simple temporal acceleration and instead support a sensitization effect of reduced contractility on the differentiation response. This interpretation is further supported by preliminary ATAC-seq data indicating early chromatin opening at WNT-responsive regions following Y-27632 treatment.

By contrast, in **Figure 1** and **Supplementary Figure 1**, we examined the effect of increased contractility. Here, mesoderm commitment is fully blocked at both 24 h and 72 h, and therefore inclusion of a 48 h intermediate point would not provide additional insight.

Figure 3 for Reviewers

Legend:

Immunofluorescence for Slug and ZO-1 of hiPSC treated for up to 72h with CHIR +/- Y-27632. Slug channel is displayed as "green fire blue" LUT to reflect pixel intensity, (Blue = low, Yellow = High). ZO-1 channel is shown as inverted LUT. Scale bar = 50 μ m

5- Experiments to further support the notion that localization of beta-catenin at the junction is controlled by contractility would significantly strengthen the manuscript. One idea would be to FRAP junctional beta-catenin with Chiron under high and low contractility. This seems feasible given the EGFP-beta-catenin cell line already exists (and the data in Figure 5 is a highlight)

We have now performed FRAP of junctional EGFP- β -catenin under baseline and low-contraction conditions (pre-differentiation) and after 24 h CHIR exposure. Across conditions, β -catenin recovery kinetics were indistinguishable, indicating that contractility does not measurably alter the exchange rate of the junction-engaged β -catenin pool in this system.

These data are consistent with published work from Hall et al (MBoC 2019)¹¹, showing that contractility does not affect Armadillo (Drosophila β -catenin) dynamics. These data are now part of a new **Supplementary Figure 6G-K**.

Legend:

Stills from FRAP experiments using EGFP-CTNNB1 hiPSCs treated with Vehicle or Y-27632 prior to differentiation (G) or after 24h +CHIR (I). Normalized EGFP intensity was measured over time (H and J) and half-recovery times were calculated (K). Scale bar = 2 μ m.

n=19-25 cells across N=2 independent biological repeats

Importantly, junctional β -catenin is predominantly bound within the cadherin–catenin adhesion complex, where its turnover is constrained by E-cadherin stability and junctional anchoring rather than by freely diffusing β -catenin dynamics. Thus, FRAP of junctional β -catenin largely reports the dynamics of cadherin-associated complexes rather than signaling-competent β -catenin. Consistent with this view, changes in actomyosin contractility have been shown to regulate adherens junction organization, reinforcement, and β -catenin sequestration without necessarily altering molecular turnover rates of the junctional pool (Mège & Ishiyama, 2017, Cold Spring Harb. Perspect. Biol 2017, Hall MBoC 2019, Röper et al eLife 2018, Yonemura et al, Nat. Cell Biol. 2010)^{11,17–19}.

Finally, this result is consistent with published FRAP datasets showing a relatively stable junctional β -catenin population in other systems (Kafri et al, eLife 2016)²⁰, with kinetic shifts arising primarily when the force-transmission architecture of the cadherin complex is altered (α -catenin/vinculin coupling), rather than by contractility modulation alone (Morales-Camilo et al., Nat. Comm 2024)²¹. Together with our localization/perturbation data, these results support a model in which contractility regulates steady-state junctional enrichment/organization of adherens junctions (and thereby β -catenin availability), rather than β -catenin binding-unbinding kinetics at junctions.

We have extended our discussion to reflect the addition of this new data and our interpretation (lines 445-448).

6- Drug treatments: more information should be provided with respect to the concentrations that were used for all drug treatments (CN03, ML7, Y27632, H1152). Although the precise concentration is given for some drugs, it is completely missing for a few (Y-26632 for example). Maybe the selected concentrations used are standard in the field, but given this information is not provided, and moreover, that CN03 treatment is lethal after 2 days begs the question of why a lower concentration wasn't used. Dose-response curves could yield interesting insights, but I understand doing this for all datasets is beyond the scope of what is reasonable here.

We thank the reviewer for pointing this out and apologize for not clearly stating drug concentrations in the main text and figure legends. While all concentrations were provided in Supplementary Table 1, we agree it is important for readers to see this information at a glance. We have now revised the figure legends to include the concentration of each drug (CN03, ML7, Y-27632, H1152).

Regarding CN03, we acknowledge that this compound is cytotoxic with prolonged exposure. Lower concentrations were less toxic but had little effect on actomyosin contractility and a ten-fold lower concentration of CNO3 did not impact contractility. In our study, these data were intended as proof-of-concept to demonstrate that increased contractility can block mesoderm commitment, which motivated the design of our more specific genetic tool (MYPT1^{CA}).

7- The discussion would benefit from a stronger conclusion and broader outlook. A key unanswered question is whether the observed relaxation before differentiation reflects what happens in vivo, especially since many studies suggest increased tension drives morphogenetic events like primitive streak formation. Additionally, given the early mention of the cardiomyocyte lineage, it would be useful to discuss whether certain lineages, like cardiomyocytes, are more influenced by actomyosin dynamics than others. Expanding on these points would help place the findings in a broader developmental context.

Our manuscript is, to our knowledge, the first to report this sensitization mechanism.

We want to clarify that we do not claim here that the role of actomyosin contractility is specific to mesoderm/cardiomyocyte specification, although the majority of papers on mechanobiology in pluripotent stem cells are specifically focused on cardiogenesis.

In adult mesenchymal stem cells, it is well established that matrix stiffness biases differentiation trajectory and that non-muscle myosin II activity is required for this directed specification (Engler et al, Cell 2006) ²². Additionally, Jian et al (Stem Cell reports, 2024) ²³ found that endoderm specification by embryonic stem cells is regulated through the actomyosin-dependent Hippo pathway and Toyoda et al (Stem Cell report 2017) ²⁴ reported that ROCK/non-muscle 2 inhibition facilitates differentiation to pancreatic endoderm. However, for many lineages no information is available.

With respect to in vivo analysis, such studies are not possible for human embryos. However, in chick embryos, prior to primitive streak formation and gastrulation, tissue dynamics is spatially organized with contracting cells restricted at the location of the future streak and surrounded by expanding cells (Rozbicki et al, Nat. Cell Biol 2015) ²⁵. Prior to mouse gastrulation, actomyosin dynamics is crucial for cell sorting in the blastomere and for cell specification (Maitre et al, Nature 2016) ²⁶ and ICM cells have a lower tension than the outer cells in the early embryo.

We have expanded the Discussion to include some of these studies, plus two reviews summarizing recent advances (Lim & Plachta Nat. Rev. Mol Cell Biol 2021 and Nelson (Annu. Rev. Biomed. Eng 2022)^{27,28} (lines 371-378)

Minor points:

- There is no label for the time of imaging/time after adding Chiron in Fig. 1 and 2d (Fig. 2j has one for example). I think it could be helpful to add one, this goes along with the comments about timing of experiments.

We have added a time for **Figure 2D**. For **Figure 1**, the time (72h) is already shown in the schematic in panel 1A. We now also emphasized the timing in the figure legend as followed:

“(B-H) Representative MaxIP immunofluorescence for Brachyury (B), EOMES (D), Slug (F) and ZO-1 (H) in Vehicle and Doxycycline-induced ROCK2^{CA} co-culture, treated for 72h with CHIR.”

- The data that deals with apoptosis is a bit confusing and distracts from the major findings of the paper. Since these are largely negative results, they should be shown only in supplemental figures and discussed together in one place.

We thank the reviewer for this comment. The apoptosis data were included as an important negative control, since apoptotic signaling is essential for mesoderm specification. Given the well-established overlap between apoptosis and contractility in epithelial monolayers, it was critical to verify that actomyosin perturbation did not alter apoptosis levels in our system. For this reason, we feel it is important to retain these data in the main figures.

- Why did the authors choose to switch from pure IF analysis in Fig. 1 to WB analysis in Fig. 2? Why is there no EOMES quantification in Fig. 2? A short explanation would be helpful.

Throughout the manuscript, our goal has been to provide orthogonal approaches to test each hypothesis. Western blotting offers a less biased readout than immunofluorescence so when possible, we tried to include this method. However, in the specific case of the co-culture experiment in **Figure 1**, western blot analysis is not feasible because experimental and wild-type cells were intermixed at a 50:50 ratio.

We are also a bit uncertain about the reference to **Figure 2**, as EOMES staining is not included in that figure. If the reviewer was instead referring to **Figure 1**, EOMES quantification is provided in panel 1E. Instead, if the reviewer was wondering why we did not stain for EOMES, we and others have previously shown that EOMES and Brachyury are co-expressed in response to WNT signaling (Fort et al, Nat. Cell Biol 2022, Tasic et al, Nat. Cell Biol 2019, Schüle et al, Dev Cell 2023)²⁹⁻³¹

- It is impossible to see ZO-1 in the images in Fig. 2d and 2j. Better quality images should be provided. We agree with the review and apologize for the dim/low resolution signal. We have reprocessed the figure to remove the ZO-1 channel, as it is not needed for the conclusion of those experiments.

- It is in general a bit difficult for the reader to follow why the authors switch between IF, WB and then also qPCR analysis in Fig. 3d-g. Maybe this could be briefly clarified.

We thank the reviewer for this suggestion. As noted above, we intentionally used orthogonal approaches (immunofluorescence, western blot, and qPCR) to strengthen the conclusions. In addition, qPCR was employed for *MESP1* and *TBX6* due to the lack of sufficiently specific antibodies. This approach demonstrates that contractility directly affects mesoderm gene expression, independent of effects on protein stability. We have now clarified this rationale in the manuscript (line 155-156).

- TCF7L2: there is a typo on line 336 (TCF4L2 is written instead).

Thank you. The typo has been fixed.

Reviewer #3:

This manuscript presents a study on hPSCs and the potential cross-talk between myosin-dependent cell contractility and induction of mesoderm fate. It is known that hPSC colonies can produce mesoderm upon activation of the Wnt pathway. The authors made here the observation that when the hPSCs was treated with a RhoGTPase activator, Wnt-dependent induction of TBXT was drastically inhibited. They validated this result through expression of myosin activators caRock and caMLCK, and, reciprocally, they showed that myosin inhibition significantly speeds up mesoderm induction (and EMT, using Slug as marker). The use of Rock chemical inhibitor allowed to test temporality, revealing that contractility plays an antagonistic role at early stages of induction (as opposed to the later effect of Wnt signalling that increases contractility). The author further looked into the role of cadherin adhesion, and showed that reducing cell adhesion also increased mesoderm induction, probably by loosening mechanical coupling and thus global contractility. Finally, they found that experimental cell relaxation led to a decrease in junctional β -catenin and a concomitant increase in nuclear β -catenin, in β -catenin interaction with a member of the TCF family, and in occupancy of the TBXT promoter. The authors provide further validating data, which I will not summarize here.

The initial observation of the dramatic block of TBXT expression upon Rock inhibition is indeed spectacular, and definitely deserved to build a project to understand this effect. The study shows clean, convincing data supporting a model where contractility is antagonist to mesoderm induction, and where contractility and β -catenin, thus presumably Wnt signalling, influence each other.

This is altogether a nice piece of work on a nifty system where the contractility- β -catenin-Wnt system yield a clear-cut output in terms of cell fate.

However, I am not quite as enthusiastic about the originality and general impact of these findings. I explain here below why the major findings are not quite up to what one would expect in JCB:

We thank the reviewer for their thorough summary of our manuscript and their positive comments regarding the strength and quality of the data. We address the issue of novelty below:

1) Adhesion, tension and Wnt signalling: There have been over more than two decades multiple observations indicating that β -catenin can be released from junctions to activate signalling, or reciprocally that sequestration by cadherins buffers β -catenin and augments the threshold required for

Wnt signalling. Similarly, the fact that decreasing adhesion impacts on cell contractility and vice-versa is similarly well established. The actual key questions relate to the mechanisms that may couple this crosstalk with regulation of nuclear β -catenin levels/activity. Unfortunately, the data presented here remain "descriptive" and do not provide new clues on this topic.

We respectfully disagree with the reviewer's statement that the mechanisms linking adhesion, tension, and β -catenin signaling are essentially resolved. While numerous studies over the past two decades have documented correlations between cadherin-based adhesion, junctional β -catenin sequestration, and WNT pathway output, the mechanisms that couple these processes to nuclear β -catenin levels and transcriptional activity remain incompletely understood.

Importantly, a recent synthesis of the field highlights substantial conceptual and experimental gaps rather than consensus (van der Wal & van Amerongen, *Open Biology*, 2020)⁸. This review explicitly emphasizes three unresolved issues that are directly relevant to our study:

1. Subcellular localization does not equate to functional pool identity. Membrane-associated β -catenin is not necessarily part of a stable, force-bearing adherens junction complex, and cytoplasmic β -catenin comprises multiple biochemically distinct species (e.g., freely diffusible, destruction-complex-associated, or vesicular). As a result, inferring signaling competence from localization alone remains problematic.
2. Membrane-associated regulation of β -catenin is poorly defined. Although components of the destruction complex and WNT signalosome can localize to the plasma membrane, it remains unclear whether (and how) β -catenin turnover, stabilization, or release is regulated at this site.
3. There is no consensus on whether, how, or when junctional and signaling pools exchange. Despite long-standing models proposing cadherin-mediated "buffering" of β -catenin, direct experimental evidence demonstrating pool-to-pool conversion under physiological conditions is limited, and the field lacks agreement on the mechanisms involved.

Our key finding, that reduced actomyosin contractility increases nuclear β -catenin levels in mammalian cells, is, to our knowledge, novel, particularly in the absence of genetic manipulation of WNT pathway components. More importantly, we provide mechanistic insight by directly linking junctional β -catenin to transcriptional output. Using CUT&RUN, we show that β -catenin originating from the junctional pool is associated with direct occupancy of the *TBXT* promoter. To our knowledge, there is no prior study that directly connects junctional β -catenin dynamics to promoter binding at endogenous WNT target loci.

Thus, rather than reiterating established correlations between adhesion and signaling, our data provide new mechanistic evidence that mechanical regulation of junctional β -catenin can directly impact nuclear β -catenin function and gene regulation, addressing precisely the gap identified by the reviewer.

2) Significance for developmental biology: The other potential interest of this work is the strong permissive role of low contractility on mesoderm fate induction: This would be exciting if it would reflect an actual developmentally relevant process: In such scenario, contractility act as a gatekeeper, and

timely cell softening would contribute to mesoderm induction. Does this happen physiologically? Not being an expert in mammalian/human mesoderm induction and gastrulation, I am not quite sure, but as far as I know TBXT induction occurs before EMT, at a stage where cells are still within the epiblast, thus an epithelial monolayer. Is there any evidence that this layer softens BEFORE TBXT induction?

We thank the reviewer for raising this important point regarding the developmental relevance of the permissive effect of low contractility on mesoderm induction. We agree that TBXT induction occurs while cells remain within the epiblast epithelium and precedes EMT. However, we would like to clarify that changes in mechanical state and actomyosin tension can occur within an intact epithelium and do not require EMT or junction dissolution.

Indeed, multiple *in vivo* studies have demonstrated that the epiblast is mechanically heterogeneous and dynamically regulated prior to and during primitive streak formation, at the stage when *TBXT* is first induced. In particular, posterior epiblast cells exhibit spatially regulated actomyosin organization and reduced cortical tension while maintaining epithelial integrity, indicating that mechanical softening or reduced contractility can precede, and potentially license, mesodermal gene expression. For example, Rozbicki et al. (Nat. Cell Biol 2015) ²⁵ showed that patterned actomyosin contractility and force anisotropy emerge in the mouse epiblast before and during streak initiation, while cells remain epithelial. Similarly, Chuai et al., (Dev Biol 2006) ³² demonstrated that actomyosin-dependent mechanical asymmetries are established in the epiblast prior to large-scale cell ingression.

Importantly, modulation of actomyosin contractility has been shown to gate primitive streak formation and mesodermal programs without requiring prior EMT (Martin et al, Nature 2009 and Shindo & Wallingford Science 2014) ^{33,34}, both in embryonic and tissue-scale contexts. Perturbations that reduce myosin II-dependent tension can induce streak-associated behaviors, whereas excessive cortical tension suppresses them, supporting the idea that contractility functions as a permissive regulator of mesoderm induction.

Together, these studies indicate that mechanical tuning of the epiblast, specifically reduced or redistributed actomyosin tension, occurs at or before the onset of *TBXT* expression, independently of EMT. While it is challenging to expand these findings to human development, our *in vitro* data support a model in which low contractility acts as a permissive mechanical state that lowers the threshold for mesoderm induction, rather than as an instructive EMT-like program.

In any case, the cellular system used in this study is highly artificial, and it is impossible to infer a physiological relevance. Let me explain in a few words: hPSC colonies are grown in an environment without any resemblance to a gastrula embryo. Most relevant here is tension, which is likely to be orders of magnitude lower in the embryo. The "standard" hPSC growing conditions are probably then completely off. These "poor cells" are under abnormally high tension and respond by reinforcing adhesions and cytoskeleton coupling, and sequestering more β -catenin. The resulting threshold for Wnt signalling gets artificially high, and induction works (suboptimally) because the pathway is overactivated by high doses of chemical activator (GSK3 inhibitor, to be specific). If this is the case, then Rock inhibition, or inhibition of adhesion, merely restore conditions more in line within the range of tension

that cells are supposed to experience during gastrulation, and mesoderm induction is getting more optimal. In summary: The results are potentially exciting if there is indeed a developmental program that involves tension as a true "regulator". Alternatively, the authors may have found a parameter that improves hPSC induction, which is a technical advance in the field of iPSCs. To test the first (and obviously most exciting) possibility, the authors would need to compare their in vitro settings with in vivo conditions (gastruloids could do). One would minimally determine whether the identified mechanical control of induction works within ranges of contractility/softness expected during gastrulation, and similarly compare the levels of Wnt- β -catenin signalling activation in both systems.

We agree with the reviewer that human pluripotent stem cell-based systems are inherently reductionist and cannot fully recapitulate the complex geometry, force distributions, and morphogenetic dynamics of a gastrulating embryo. While in principle gastruloids would be a more "in vivo-like" system, the standard method for gastruloid formation we have used, unfortunately requires Y-27632 + CHIR treatment during the initial aggregation phase and they do not develop in the absence of this ROCK inhibitor (Moris et al, Nature 2020)³⁵. Murine embryo development is very different from that of human embryos with, for instance, opposite effects of Wnt activation, so these are not a useful model.

Our goal in this study was not to recreate a gastrula-like mechanical environment, but rather to interrogate how cell-intrinsic contractility and adhesion state modulate responsiveness to mesoderm-inducing signals under controlled conditions. Note that our protocol achieves almost 100% mesoderm specification and (see Fort et al Nature Cel Biol 2022)³¹ cardiomyocyte differentiation, so is highly efficient, and recapitulates all of the embryonic stages from primitive streak gene expression through cardiac mesoderm specification as is believed to take place in the human embryo.

Importantly, while it is reasonable to speculate that standard hPSC culture conditions may impose non-physiological mechanical constraints, there is currently no quantitative information on the magnitude or spatial distribution of mechanical tension or stiffness in the human epiblast during gastrulation. Existing "embryo-like" compliant substrates are proxies derived largely from avian or non-human systems, and their application to human stem cells does not resolve this fundamental gap.

Finally, we note that comparable effects were obtained when cells were cultured on vitronectin-XF instead of Matrigel, indicating that the permissive role of reduced contractility on mesoderm induction is not specific to a single matrix formulation and different mechanical properties of the substrate (**Figure 2 for reviewers**)

Other comment

Fig6A. a) The cell fractionation protocol is unlikely to truly separate "nuclei" from "cytoplasm". The "nuclear" fraction is most certainly a nuclear insoluble fraction (chromatin + lamina), which is completely fine but should be called as such. A true nuclear fraction should be validated using a soluble nucleoplasmic marker. b) Also missing is a blot for cadherins to determine where is the plasma membrane pool of β -catenin ("cytoplasm", "nuclei"?). Without these ESSENTIAL controls, it is impossible to interpret these experiments.

We appreciate this important comment from the reviewer. Despite multiple attempts, we were not able to cleanly isolate E-Cadherin from these fractions. Therefore, we have decided to remove the cell fractionation data. This alteration does not alter our conclusion as we show a clear accumulation of non-phosphorylated beta-catenin in the nucleus in low contractile cells using immunofluorescence.

References

1. Shi, W. *et al.* RhoA/Rock activation represents a new mechanism for inactivating Wnt/ β -catenin signaling in the aging-associated bone loss. *Cell Regeneration* 10, 8 (2021).
2. Dobson, L. *et al.* GSK3 and lamellipodin balance lamellipodial protrusions and focal adhesion maturation in mouse neural crest migration. *Cell Rep* 42, 113030 (2023).
3. Hookway, C. *et al.* A human induced pluripotent stem (hiPS) cell model for the holistic study of epithelial to mesenchymal transitions (EMTs). *BioRxiv* (2024).
4. HEASMAN, J. Overexpression of cadherins and underexpression of β -catenin inhibit dorsal mesoderm induction in early *Xenopus* embryos. *Cell* 79, 791–803 (1994).
5. Fagotto, F., Funayama, N., Gluck, U. & Gumbiner, B. M. Binding to cadherins antagonizes the signaling activity of beta-catenin during axis formation in *Xenopus*. *J Cell Biol* 132, 1105–1114 (1996).
6. Sanson, B., White, P. & Vincent, J.-P. Uncoupling cadherin-based adhesion from wingless signalling in *Drosophila*. *Nature* 383, 627–630 (1996).
7. Gottardi, C. J. & Gumbiner, B. M. Adhesion signaling: How β -catenin interacts with its partners. *Current Biology* 11, R792–R794 (2001).
8. van der Wal, T. & van Amerongen, R. Walking the tight wire between cell adhesion and WNT signalling: a balancing act for β -catenin. *Open Biol* 10, (2020).
9. Muncie, J. M. *et al.* Mechanical Tension Promotes Formation of Gastrulation-like Nodes and Patterns Mesoderm Specification in Human Embryonic Stem Cells. *Dev Cell* 55, 679-694.e11 (2020).
10. Przybyla, L., Lakins, J. N. & Weaver, V. M. Tissue Mechanics Orchestrate Wnt-Dependent Human Embryonic Stem Cell Differentiation. *Cell Stem Cell* 19, 462–475 (2016).
11. Hall, E. T., Hoesing, E., Sinkovics, E. & Verheyen, E. M. Actomyosin contractility modulates Wnt signaling through adherens junction stability. *Mol Biol Cell* 30, 411–426 (2019).
12. Brunet, T. *et al.* Evolutionary conservation of early mesoderm specification by mechanotransduction in Bilateria. *Nat Commun* 4, 2821 (2013).

13. Pukhlyakova, E., Aman, A. J., Elsayad, K. & Technau, U. β -Catenin-dependent mechanotransduction dates back to the common ancestor of Cnidaria and Bilateria. *Proceedings of the National Academy of Sciences* 115, 6231–6236 (2018).
14. Yamamoto, K. *et al.* Optogenetic relaxation of actomyosin contractility uncovers mechanistic roles of cortical tension during cytokinesis. *Nat Commun* 12, 7145 (2021).
15. Kim, Y. *et al.* Cell position within human pluripotent stem cell colonies determines apical specialization via an actin cytoskeleton-based mechanism. *Stem Cell Reports* 17, 68–81 (2022).
16. Vasic, I. *et al.* Loss of TJP1 disrupts gastrulation patterning and increases differentiation toward the germ cell lineage in human pluripotent stem cells. *Dev Cell* 58, 1477–1488.e5 (2023).
17. Mège, R. M. & Ishiyama, N. Integration of Cadherin Adhesion and Cytoskeleton at *Adherens* Junctions. *Cold Spring Harb Perspect Biol* 9, a028738 (2017).
18. Yonemura, S., Wada, Y., Watanabe, T., Nagafuchi, A. & Shibata, M. α -Catenin as a tension transducer that induces adherens junction development. *Nat Cell Biol* 12, 533–542 (2010).
19. Röper, J.-C. *et al.* The major β -catenin/E-cadherin junctional binding site is a primary molecular mechano-transducer of differentiation in vivo. *Elife* 7, (2018).
20. Kafri, P. *et al.* Quantifying β -catenin subcellular dynamics and cyclin D1 mRNA transcription during Wnt signaling in single living cells. *Elife* 5, (2016).
21. Morales-Camilo, N. *et al.* Alternative molecular mechanisms for force transmission at adherens junctions via β -catenin-vinculin interaction. *Nat Commun* 15, 5608 (2024).
22. Engler, A. J., Sen, S., Sweeney, H. L. & Discher, D. E. Matrix Elasticity Directs Stem Cell Lineage Specification. *Cell* 126, 677–689 (2006).
23. Jiang, L. *et al.* Cell size regulates human endoderm specification through actomyosin-dependent AMOT-YAP signaling. *Stem Cell Reports* 19, 1137–1155 (2024).
24. Toyoda, T. *et al.* Rho-Associated Kinases and Non-muscle Myosin IIs Inhibit the Differentiation of Human iPSCs to Pancreatic Endoderm. *Stem Cell Reports* 9, 419–428 (2017).
25. Rozbicki, E. *et al.* Myosin-II-mediated cell shape changes and cell intercalation contribute to primitive streak formation. *Nat Cell Biol* 17, 397–408 (2015).
26. Maître, J.-L. *et al.* Asymmetric division of contractile domains couples cell positioning and fate specification. *Nature* 536, 344–348 (2016).
27. Lim, H. Y. G. & Plachta, N. Cytoskeletal control of early mammalian development. *Nat Rev Mol Cell Biol* 22, 548–562 (2021).
28. Nelson, C. M. Mechanical Control of Cell Differentiation: Insights from the Early Embryo. *Annu Rev Biomed Eng* 24, 307–322 (2022).

29. Tasic, J. *et al.* Eomes and Brachyury control pluripotency exit and germ-layer segregation by changing the chromatin state. *Nat Cell Biol* 21, 1518–1531 (2019).
30. Schüle, K. M. *et al.* Eomes restricts Brachyury functions at the onset of mouse gastrulation. *Dev Cell* 58, 1627-1642.e7 (2023).
31. Fort, L., Gama, V. & Macara, I. G. Stem cell conversion to the cardiac lineage requires nucleotide signalling from apoptosing cells. *Nat Cell Biol* 24, 434–447 (2022).
32. Chuai, M. *et al.* Cell movement during chick primitive streak formation. *Dev Biol* 296, 137–149 (2006).
33. Martin, A. C., Kaschube, M. & Wieschaus, E. F. Pulsed contractions of an actin–myosin network drive apical constriction. *Nature* 457, 495–499 (2009).
34. Shindo, A. & Wallingford, J. B. PCP and Septins Compartmentalize Cortical Actomyosin to Direct Collective Cell Movement. *Science* (1979) 343, 649–652 (2014).
35. Moris, N. *et al.* An in vitro model of early anteroposterior organization during human development. *Nature* 582, 410–415 (2020).

February 18, 2026

RE: JCB Manuscript #202507103R

Loic Fort
The University of Texas at Arlington

Dear Dr. Fort:

Thank you for submitting your revised manuscript entitled "Actomyosin Contractility is a Potent Suppressor of Mesoderm Induction by Human Pluripotent Stem Cells". The reviewers all now support publication so we would be happy to publish your paper in JCB pending final revisions necessary to meet our formatting guidelines (see details below).

In your final revision, please be sure to address reviewer #1's final minor concerns in the text.

A. MANUSCRIPT ORGANIZATION AND FORMATTING:

Full guidelines are available on our Instructions for Authors page, <http://jcb.rupress.org/submission-guidelines#revised>.

- 1) Text limits: Character count for Articles is < 40,000, not including spaces. Count includes abstract, introduction, results, discussion, and acknowledgments. Count does not include title page, figure legends, materials and methods, references, tables, or supplemental legends.
- 2) Figures limits: Articles may have up to 10 main text figures.
- 3) Figure formatting: Scale bars must be present on all microscopy images, including inset magnifications. Molecular weight or nucleic acid size markers must be included on all gel electrophoresis. Aspect ratios of images may not be altered.
- 4) Statistical analysis: Error bars on graphic representations of numerical data must be clearly described in the figure legend. The number of independent data points (n) represented in a graph must be indicated in the legend. Statistical methods should be explained in full in the materials and methods. For figures presenting pooled data the statistical measure should be defined in the figure legends. Please also be sure to indicate the statistical tests used in each of your experiments (either in the figure legend itself or in a separate methods section) as well as the parameters of the test (for example, if you ran a t-test, please indicate if it was one- or two-sided, etc.). Also, if you used parametric tests, please indicate if the data distribution was tested for normality (and if so, how). If not, you must state something to the effect that "Data distribution was assumed to be normal but this was not formally tested."
- 5) Abstract and title: The abstract should be no longer than 160 words and should communicate the significance of the paper for a general audience. The title should be less than 100 characters including spaces. Make the title concise but accessible to a general readership.
- 6) Materials and methods: Should be comprehensive and not simply reference a previous publication for details on how an experiment was performed. Please provide full descriptions in the text for readers who may not have access to referenced manuscripts.
- 7) All antibodies, cell lines, animals, and tools used in the manuscript should be described in full, including accession numbers for materials available in a public repository such as the Resource Identification Portal. Please be sure to provide the sequences for all of your primers/oligos and RNAi constructs in the materials and methods. You must also indicate in the methods the source, species, and catalog numbers (where appropriate) for all of your antibodies. Please also indicate the acquisition and quantification methods for immunoblotting/western blots.
- 8) Microscope image acquisition: The following information must be provided about the acquisition and processing of images:
 - a. Make and model of microscope
 - b. Type, magnification, and numerical aperture of the objective lenses
 - c. Temperature
 - d. Imaging medium
 - e. Fluorochromes
 - f. Camera make and model
 - g. Acquisition software

h. Any software used for image processing subsequent to data acquisition. Please include details and types of operations involved (e.g., type of deconvolution, 3D reconstitutions, surface or volume rendering, gamma adjustments, etc.).

10) Supplemental materials: There are strict limits on the allowable amount of supplemental data. Articles may have up to 5 supplemental figures. You currently have 7 supplemental figures, and we will be able to give you a bit more space in this case. Please also note that tables, like figures, should be provided as individual, editable files. A summary of all supplemental material should appear at the end of the Materials and methods section.

**13) ORCID IDs: ORCID IDs are unique identifiers allowing researchers to create a record of their various scholarly contributions in a single place. Please note that ORCID IDs are now *required* for all authors. At resubmission of your final files, please be sure to provide your ORCID ID and those of all co-authors.

Please note that JCB now requires authors to submit Source Data used to generate figures containing gels and Western blots with all revised manuscripts. This Source Data consists of fully uncropped and unprocessed images for each gel/blot displayed in the main and supplemental figures. For assays performed using capillary electrophoresis and/or immunoassay-based detection, authors should instead provide the electropherogram graph(s) for each experiment, plotting fluorescence/chemiluminescence intensity vs. molecular weight/size. Please be sure to provide one Source Data file for each figure gels, blots, and/or capillary electrophoresis assays along with your revised manuscript files. File names for Source Data figures should be alphanumeric without any spaces or special characters (i.e., SourceDataF#, where F# refers to the associated main figure number or SourceDataFS# for those associated with Supplementary figures). For traditional gels and blots, the lanes of the gels/blots should be labeled as they are in the associated figure, the place where cropping was applied should be marked (with a box), and molecular weight/size standards should be labeled wherever possible. For capillary electrophoresis assays, each trace in the graph should be color-coded and labeled to indicate which protein, gene, or sample is being measured (please try to avoid red/green combinations to accommodate our color-blind readers).

Journal of Cell Biology now requires a data availability statement for all research article submissions. These statements will be published in the article directly above the Acknowledgments. The statement should address all data underlying the research presented in the manuscript. Please visit the JCB instructions for authors for guidelines and examples of statements at (<https://rupress.org/jcb/pages/editorial-policies#data-availability-statement>).

B. FINAL FILES:

-- Cover images: If you have any striking images related to this story, we would be happy to consider them for inclusion on the

journal cover. Submitted images may also be chosen for highlighting on the journal table of contents or JCB homepage carousel. Images should be uploaded as TIFF or EPS files and must be at least 300 dpi resolution.

****It is JCB policy that if requested, original data images must be made available to the editors. Failure to provide original images upon request will result in unavoidable delays in publication. Please ensure that you have access to all original data images prior to final submission.****

****The license to publish form must be signed before your manuscript can be sent to production. A link to the electronic license to publish form will be sent to the corresponding author only. Please take a moment to check your funder requirements before choosing the appropriate license.****

Thank you for your attention to these final processing requirements. Please revise and format the manuscript and upload materials within 7 days. If you need an extension for whatever reason, please let us know and we can work with you to determine a suitable revision period.

Thank you for this interesting contribution, we look forward to publishing your paper in Journal of Cell Biology.

Sincerely,

Elaine Fuchs, Ph.D.
Editor
The Journal of Cell Biology

Gabriele Stephan, Ph.D.
Scientific Editor
Journal of Cell Biology

Reviewer #1:

In the revised manuscript, the authors have addressed many of the concerns raised in the initial version. However, a few clarifications may strengthen the manuscript before publication.

1. Nuclear translocation dynamics of β -catenin.

In Fig. 6A-C, β -catenin appears to show transient nuclear localization under Veh conditions, whereas its nuclear localization seems stronger and more sustained in the presence of Y-27632. This observation may be consistent with the result shown in Fig. S7C, where a higher level of nuclear active β -catenin is detected at 24 hrs under reduced contractility. Do the authors consider that the actomyosin contractility-dependent differences in TBX-T expression observed 24 hrs after CHIR treatment could be regulated not only by the release of β -catenin from AJs, but also by the dynamics of its nuclear translocation (sustained vs. transient) and the extent of its nuclear accumulation? In other words, could actomyosin contractility modulate β -catenin dynamics after its release from AJs? The additional ATAC-seq data on chromatin accessibility may be relevant to this. If this is the case, to what extent do junctional actomyosin contractility and the amount of junctional β -catenin contribute to mesoderm specification? As this point lies at the core of the model proposed in this manuscript, the authors should discuss the relative importance of the junction-based titration mechanism.

2. I.348: The "licensing model" is not defined at this point and is therefore unclear. I recommend the authors to remove this term or to provide a brief explanation, ideally in conjunction with Fig. 7.

Reviewer #2:

It is clear that the authors have gone to considerable effort to thoughtfully address all my prior concerns. It is now suitable for publication in JCB.

Reviewer #3:

The authors have satisfactorily dealt with most of the reviewers' concerns and comments.

I do still have strong reservations about the physiological significance of the mechanical-dependent effect, and it will remain of utter importance to validate this mechanism under conditions where physical constrains are closer to the situation in an actual embryo.

But such reservations are applicable to the broader community, and the current manuscript present a very nice piece of work, on a compelling case of b-catenin regulation at the interplay between Wnt signalling and adhesion/mechanics.